# ARB-LLM: ALTERNATING REFINED BINARIZATIONS FOR LARGE LANGUAGE MODELS

**Zhiteng Li**[1*], **Xianglong Yan**[1*], **Tianao Zhang**[1], **Haotong Qin**[2], **Dong Xie**[3],
**Jiang Tian**[3], **Zhongchao Shi**[3], **Linghe Kong**[1†], **Yulun Zhang**[1†], **Xiaokang Yang**[1]
[1]Shanghai Jiao Tong University, [2]ETH Zürich, [3]Lenovo Research

## ABSTRACT

Large Language Models (LLMs) have greatly pushed forward advancements in natural language processing, yet their high memory and computational demands hinder practical deployment. Binarization, as an effective compression technique, can shrink model weights to just 1 bit, significantly reducing the high demands on computation and memory. However, current binarization methods struggle to narrow the distribution gap between binarized and full-precision weights, while also overlooking the column deviation in LLM weight distribution. To tackle these issues, we propose ARB-LLM, a novel 1-bit post-training quantization (PTQ) technique tailored for LLMs. To narrow the distribution shift between binarized and full-precision weights, we first design an alternating refined binarization (ARB) algorithm to progressively update the binarization parameters, which significantly reduces the quantization error. Moreover, considering the pivot role of calibration data and the column deviation in LLM weights, we further extend ARB to ARB-X and ARB-RC. In addition, we refine the weight partition strategy with column-group bitmap (CGB), which further enhance performance. Equipping ARB-X and ARB-RC with CGB, we obtain ARB-LLM$_X$ and ARB-LLM$_{RC}$ respectively, which significantly outperform state-of-the-art (SOTA) binarization methods for LLMs. As a binary PTQ method, our ARB-LLM$_{RC}$ is the first to surpass FP16 models of the same size. Code: `https://github.com/ZHITENGLI/ARB-LLM`.

## 1 INTRODUCTION

Recently, Transformer-based (Vaswani, 2017) large language models have shown impressive performance across various natural language processing tasks. However, this unprecedented capability is largely attributed to the sheer scale of these models, which often encompass billions of parameters. For instance, open pre-trained Transformer (OPT) series (Zhang et al., 2022) includes various models, with the largest boasting 66B parameters. Similarly, the LLaMA family (Touvron et al., 2023) features variants such as LLaMA3-70B, showcasing even larger architectures. The substantial memory requirements for inference in such large models (*e.g.,* 150 GB memory for a 70B model) pose significant challenges for their deployment on mobile devices.

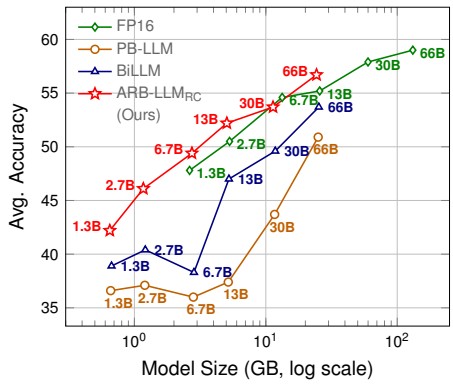

Figure 1: OPT performance on 7 zero-shot Question Answering (QA) datasets. Our ARB-LLM$_{RC}$ outperforms the same-size FP16 models.

The study of compressing LLMs can be categorized into weight quantization (Lin et al., 2024; Frantar et al., 2023), low-rank factorization (Zhang et al., 2024; Yuan et al., 2023), network pruning (Sun et al., 2024; Frantar & Alistarh, 2023), and knowledge distillation (Zhong et al., 2024; Gu et al., 2024). Among these, binarization, a specific technique within the realm of quantization, is particularly distinguished for its ability to achieve extreme memory compression, reducing storage requirements to as low as 1 bit. Given the substantial size of LLMs, some binarization methods adopt the post-training quantization (PTQ) framework to enable a rapid transition from full-precision models to compact binarized versions, requiring minimal resources (*e.g.,* binarizing a 70B model in one 80 GB GPU).

---
*Equal contribution
†Corresponding authors: Linghe Kong, linghe.kong@sjtu.edu.cn, Yulun Zhang, yulun100@gmail.com

Recent binary PTQ methods, such as PB-LLM (Shang et al., 2024) and BiLLM (Huang et al., 2024), emphasize the identification of salient weights, which are crucial for model performance (Lin et al., 2024). The higher-bit representation and refined searching strategy for salient weights help to achieve a better trade-off between performance and storage. Despite their success, the refinement of the binarization process itself remains largely unaddressed, resulting in a significant difference between the binarized weights and their full-precision counterparts. This gap presents a considerable obstacle to further enhance the performance of binary LLMs.

To minimize quantization error during the binarization process, we revisit the solutions for the binarization objective. Our analyses reveal that: **(i)** The current approach is suboptimal due to the distribution shift between binarized and full-precision weights after binarization. As shown in Figure 2, the mean of the binarized weights is not aligned with the full-precision mean. Consequently, refining the binarization parameters based on the initial distribution of the binarized weights can yield a more accurate estimation of the original weights. Furthermore, this refinement can be alternately applied to different binarization parameters, ultimately leading to a significantly improved estimation. **(ii)** While the calibration set is small, it plays a crucial role in the quantization of LLMs. However, the integration of calibration data for updating binarization parameters, which reflects a more realistic scenario, remains underexplored. **(iii)** The weight distribution in LLMs exhibits noticeable column-wise deviations (see Figure 3), suggesting that the standard row-wise binarization method is inflexible and potentially unsuitable. Thus, incorporating both row and column scaling factors can produce more representative binarization results.

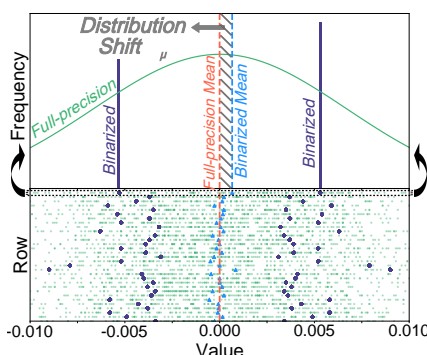

Figure 2: Distribution shift between the mean of binarized and full-precision weights. **Top**: distribution shift of one row. **Bottom**: distribution shifts of multiple rows. Each row represents a top view of the corresponding upper image.

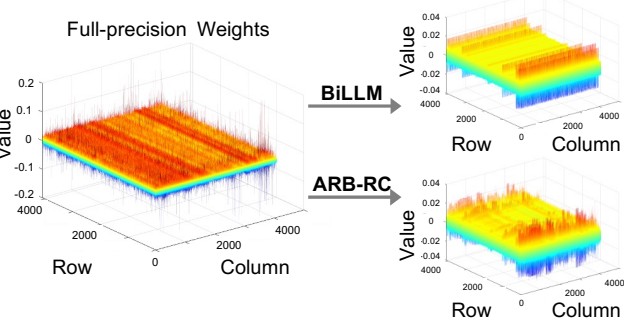

Figure 3: **Left**: Full-precision weights exhibit column-wise deviations. **Right**: BiLLM (Huang et al., 2024) with row-wise binarization smooths the deviations. Our ARB-RC with row-column-wise binarization effectively preserves them.

With the above observations and analyses, we first propose Alternating Refined Binarization (ARB) to align the distribution between binarized and full-precision weights in standard binarization process. Then, we extend this approach by incorporating the calibration data and row-column-wise scaling factors, leading to two advanced extensions: ARB-X and ARB-RC. Additionally, based on previous methods, which divide salient and non-salient weights and group weights by magnitude, we refine the integration of these two divisions by a column-group bitmap (CGB).

Our key contributions can be summarized as follows:

- We propose a novel binarization framework **ARB**, designed to progressively align the distribution between binarized and full-precision weights. In addition, we provide rigorous theoretical analyses of the quantization error throughout the progressive updates.

- Building on the basic **ARB** framework, we develop two advanced extensions: ARB with calibration data (**ARB-X**), and ARB along row-column axes (**ARB-RC**). They are tailored to address specific challenges in binarized large language models.

- We propose a refined strategy to combine the salient column bitmap and group bitmap (**CGB**), which improves the bitmap utilization and further enhances the performance.

- Extensive experiments demonstrate that our **ARB-LLM$_{RC}$ (ARB-RC + CGB)** significantly outperforms SOTA binary PTQ methods while requiring less memory. Furthermore, **ARB-LLM$_{RC}$**, for the first time, surpasses same-size FP16 models on zero-shot QA datasets.

## 2 RELATED WORKS

### 2.1 NETWORK BINARIZATION

Network binarization compresses the parameters to only 1 bit ($\pm 1$) by using the sign function. Then, the straight through estimator (STE) (Bengio et al., 2013) is used to tackle the gradient vanishing during back-propagation if training a binary network. Binary weight network (BWN) (Rastegari et al., 2016) implemented binarization on weights while maintaining full-precision activations. XNOR-Net (Rastegari et al., 2016) extended this by binarizing weights and activations. They both focus on standard first-order binarization and employ a scaling factor $\alpha$ to reduce quantization error. Network sketching (Guo et al., 2017) extended the first-order binarization by proposing a binary coding quantization (BCQ) to approximate the full-precision weights with multiple binary matrices. Xu et al. (2018) improved BCQ by using a binary search tree to determine the optimal code. However, both methods are tailored for scenarios involving multiple binarizations and are not applicable to standard first-order binarization processes. In another direction, OneBit (Xu et al., 2024) extended the scaling factor to both weights and activations. BinaryMoS (Jo et al., 2024) introduced several scaling experts to improve the performance. Nevertheless, training these models requires substantial resources. For example, training OneBit on LLaMA-7B takes 7 days using 8 A100-80GB GPUs.

### 2.2 LARGE LANGUAGE MODEL QUANTIZATION

Current quantization techniques for large language models mainly fall into quantization-aware training (QAT) and post-training quantization (PTQ) frameworks.

**Quantization-Aware Training (QAT).** QAT integrates quantization into the training process, enabling the model to adapt to low-bit representations. Recent works have successfully applied QAT to LLMs. LLM-QAT (Liu et al., 2024) addressed data barrier issues in QAT training by adding data-free distillation. EfficientQAT (Chen et al., 2024) proposed an optimized QAT framework with two stages (*i.e.,* Block-AP and E2E-QP) to reduce QAT's memory and computational overhead for LLMs. However, QAT still requires considerable computational resources, including significant GPU memory and more training time. Therefore, LLM quantization techniques such as QLoRA (Dettmers et al., 2024a) focused on parameter-efficient fine-tuning methods, which enhanced the efficiency of QAT. Nevertheless, the efficiency of the LLM quantization methods remained unsatisfactory.

**Post-Training Quantization (PTQ).** PTQ applied quantization directly to the existing model weights instead of retraining it. Therefore, it is significantly faster and more resource-efficient than QAT. Recent studies have effectively deployed PTQ in LLMs. RTN rounds weights to the nearest quantization level in order to ensure efficient runtimes when quantizing LLMs. Works like ZeroQuant (Yao et al., 2022) and BRECQ (Li et al., 2021) enhanced quantization accuracy by adding additional grouping labels for custom quantization blocks. GPTQ (Frantar et al., 2023) utilized layer-wise quantization and reduced the quantization error by second-order error compensation. Moreover, PB-LLM (Shang et al., 2024), SpQR (Dettmers et al., 2024b), and BiLLM (Huang et al., 2024) implemented a hybrid approach by selectively quantizing salient weights with low bits while binarizing non-salient weights. In addition, Smoothquant (Xiao et al., 2023) proposed a strategy of scaling weight and activation outliers, which simplified quantization. Thereafter, AWQ (Lin et al., 2024) and OWQ (Lee et al., 2024) also proposed scale transformations of salient weights for activation features to preserve their model capacity. Our work belongs to the category of binary PTQ, achieving a significant improvement over the SOTA method BiLLM.

## 3 METHOD

**Overview.** As shown in Figure 4, to progressively align the distribution between binarized and full-precision weights in LLMs, we first propose a framework Alternating Refined Binarization (**ARB**) in Section 3.1. Based on **ARB** framework, we propose the Alternating Refined Binarization with calibration data (**ARB-X**) to enhance the usage of the calibration set, which is crucial for binary LLMs. Additionally, we introduce the Alternating Refined Binarization along row-column axes (**ARB-RC**) to address the column deviation challenge in LLM weights. These methods are detailed in Sections 3.2 and 3.3, respectively. Finally, we discuss our refined strategy to combine salient column bitmap and group bitmap (**CGB**) in Section 3.4. Our final models, **ARB-LLM$_X$** and **ARB-LLM$_{RC}$**, are obtained by equipping **ARB-X** and **ARB-RC** with **CGB** respectively.

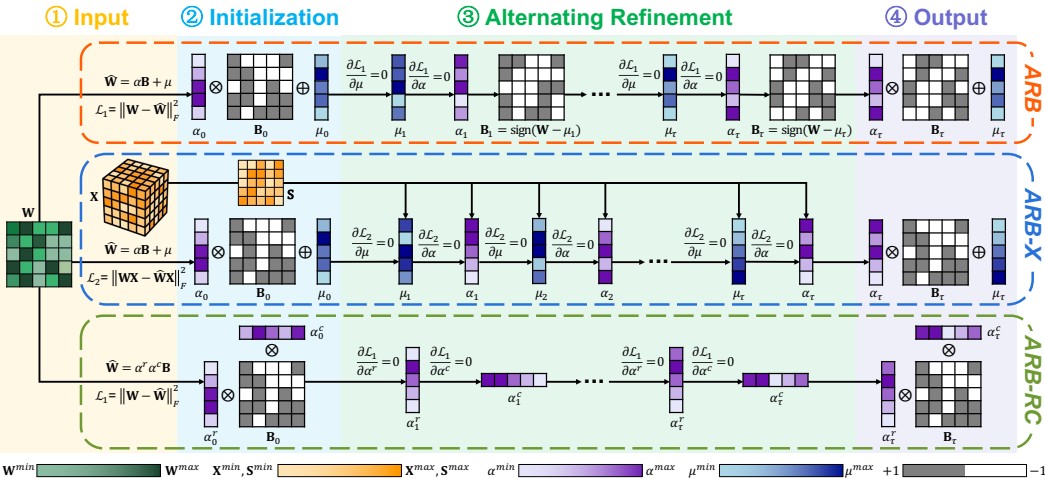

Figure 4: Overview of our ARB series. **ARB**: alternating refine mean, row scaling factor, and binarized matrix. **ARB-X**: introducing calibration data into the update of binarization parameters. **ARB-RC**: alternating refine row and column scaling factors.

## 3.1 ALTERNATING REFINED BINARIZATION (ARB)

We begin by discussing standard weight binarization in LLMs. For a full-precision weight $\mathbf{W} \in \mathbb{R}^{n \times m}$, we define the objective of binarization as (with dimension broadcasting omitted for simplicity)

$$\arg\min_{\alpha, \mathbf{B}} ||\widetilde{\mathbf{W}} - \alpha\mathbf{B}||_F^2, \quad \text{where } \widetilde{\mathbf{W}} = \mathbf{W} - \mu, \ \mu = \frac{1}{m}\sum_{j=1}^{m}\mathbf{W}_{\cdot j}, \tag{1}$$

where $\alpha \in \mathbb{R}^n$ denotes the row-wise scaling factor, and $\mathbf{B} \in \{+1, -1\}^{n \times m}$ is a binary matrix.

Since the mean of $\mathbf{W}$ is not necessarily zero, a common practice is to apply a row-wise redistribution before binarization. After redistribution, the weights achieve a row-wise zero-mean distribution, which facilitates the binarization process. Under the objective of binarization (Equation (1)), the optimal solutions for $\alpha$ and $\mathbf{B}$ can be solved with $\alpha = \frac{1}{m}\sum_{j=1}^{m}|\widetilde{\mathbf{W}}_{\cdot j}|$ and $\mathbf{B} = \text{sign}(\widetilde{\mathbf{W}})$ respectively. Then we can define the quantization error $\mathcal{L}_1$ after binarization as

$$\mathcal{L}_1 = ||\mathbf{W} - \widehat{\mathbf{W}}||_F^2, \quad \text{where } \widehat{\mathbf{W}} = \alpha\mathbf{B} + \mu. \tag{2}$$

Moving forward, we aim to investigate how to reduce the quantization error $\mathcal{L}_1$. We first define the residual matrix as $\mathbf{R} = \mathbf{W} - \widehat{\mathbf{W}}$. In analyzing the residual matrix $\mathbf{R}$, we observe a distribution shift in $\mathbf{R}$, where the mean of $\mathbf{R}$ is not always zero due to inevitable errors during the binarization process (see Figure 2). To address this, we introduce a correction term $\delta_\mu$ to the original mean $\mu$, effectively mitigating the distribution shift. The refined mean is defined as follows:

$$\mu_{\text{refine}} = \mu + \delta_\mu, \quad \text{where } \delta_\mu = \frac{1}{m}\sum_{j=1}^{m}\mathbf{R}_{\cdot j}. \tag{3}$$

This is equivalent to taking the partial derivative of $\mathcal{L}_1$ with respect to $\mu$ and setting it to 0, as shown in Figure 4. Since $\mu$ has been updated to $\mu_{\text{refine}}$, the original $\alpha$ and $\mathbf{B}$ are no longer optimal solutions for quantization error $\mathcal{L}_1$. To further minimize the quantization error, the optimal solutions for $\alpha_{\text{refine}}$ and $\mathbf{B}_{\text{refine}}$ can be obtained by setting $\partial\mathcal{L}_1/\partial\alpha = 0$, leading to the following expressions:

$$\alpha_{\text{refine}} = \frac{1}{m}\text{diag}(\mathbf{B}^\top(\mathbf{W} - \mu_{\text{refine}})), \quad \mathbf{B}_{\text{refine}} = \text{sign}(\mathbf{W} - \mu_{\text{refine}}). \tag{4}$$

After refining $\mu$, $\alpha$, and $\mathbf{B}$, we can obtain the $\widehat{\mathbf{W}}_{\text{refine}}$ as $\widehat{\mathbf{W}}_{\text{refine}} = \alpha_{\text{refine}} \cdot \mathbf{B}_{\text{refine}} + \mu_{\text{refine}}$. We find that this parameter update strategy can be extended to an iterative algorithm.

In each iteration, we sequentially update $\mu$, $\alpha$, and $\mathbf{B}$ to ensure they are the optimal solutions under the current quantization error $\mathcal{L}_1$. The pseudocode is shown in Algorithm 1, which extends **ARB** with group mask (a bitmap detailed in Section 3.4). Moreover, we theoretically analyze the quantization error during the **ARB** process and derive a specific value for the reduced quantization error after $\tau$ iterations, as stated in Theorem 1. The proof is provided in supplementary file.

---

**Algorithm 1** First-Order **A**lternating **R**efined **B**inarization

func $\text{ARB}^1(\mathbf{W}, \mathbf{M}, T)$

**Input:** $\mathbf{W} \in \mathbb{R}^{n \times m}$ - full-precision weight
$\qquad \mathbf{M} \in \mathbb{R}^{n \times m}$ - group mask
$\qquad T$ - total iterations

**Output:** $\widehat{\mathbf{W}} \in \mathbb{R}^{n \times m}$

1: $\widehat{\mathbf{W}}, \alpha, \mathbf{B}, \mu := \text{binary}(\mathbf{W}, \mathbf{M})$
2: **for** iter $= 1, 2, ..., T$ **do**
3: $\quad \mathbf{R} := \mathbf{W} - \widehat{\mathbf{W}}$ ▷ residual matrix
4: $\quad \delta_\mu := \sum_j (\mathbf{R} \odot \mathbf{M})_{.j}$
5: $\quad \mu \leftarrow \mu + \delta_\mu$ ▷ refine mean
6: $\quad \alpha \leftarrow \text{refine\_alpha}(\mathbf{B}, \mathbf{W}, \mathbf{M}, \mu)$
7: $\quad \mathbf{B} \leftarrow \text{sign}(\mathbf{W} - \mu)$ ▷ refine B
8: $\quad \widehat{\mathbf{W}} \leftarrow \alpha \mathbf{B} + \mu$
9: **end for**
10: **return** $\widehat{\mathbf{W}}$

func binary $(\mathbf{W}, \mathbf{M})$

1: $\mu := \frac{1}{m} \sum_{j=1}^m (\mathbf{W} \odot \mathbf{M})_{.j}$
2: $\widetilde{\mathbf{W}} := \mathbf{W} - \mu$
3: $\alpha := \frac{1}{m} \sum_{j=1}^m |(\widetilde{\mathbf{W}} \odot \mathbf{M})_{.j}|$
4: $\mathbf{B} := \text{sign}(\widetilde{\mathbf{W}} \odot \mathbf{M})$
5: $\widehat{\mathbf{W}} := \alpha \cdot \mathbf{B} + \mu$
6: **return** $\widehat{\mathbf{W}}, \alpha, \mathbf{B}, \mu$

func refine\_alpha $(\mathbf{B}, \mathbf{W}, \mathbf{M}, \mu)$

1: $num := \sum_{j=1}^m (\mathbf{B}_{.j} \odot \mathbf{M}_{.j}) \cdot (\mathbf{W}_{.j} - \mu)$
2: $den := \sum_{j=1}^m (\mathbf{B}_{.j} \odot \mathbf{M}_{.j})^2 + \epsilon$ ▷ avoid
zero-division
3: $\alpha := \frac{num}{den}$
4: **return** $\alpha$

---

**Theorem 1.** *For any $\tau \geq 0$, Algorithm 1 achieves a quantization error $\mathcal{L}_1^\tau$ satisfying*

$$\mathcal{L}_1^\tau = \mathcal{L}_1^0 - m((\alpha^\tau)^2 - (\alpha^0)^2 - (\mu^\tau - \mu^0)^2) \leq \mathcal{L}_1^0, \tag{5}$$

*where $\alpha^0$ and $\mu^0$ denote the initial scaling factor and mean respectively, $\alpha^\tau$, $\mu^\tau$, and $\mathcal{L}_1^\tau$ represent the scaling factor, mean, and quantization error after the $\tau$-th iteration respectively.*

To achieve better quantization precision, we extend **ARB** to second-order binarization and apply it to salient weights, following BiLLM (Huang et al., 2024). The second-order binarized matrix $\widehat{\mathbf{W}}$ is

$$\widehat{\mathbf{W}} = \alpha_1 \mathbf{B}_1 + \alpha_2 \mathbf{B}_2 + \mu, \tag{6}$$

where $\mathbf{B}_1, \mathbf{B}_2 \in \{+1, -1\}^{n \times m}$ are binary matrices, $\alpha_1, \alpha_2 \in \mathbb{R}^n$ are corresponding row-wise scaling factors, and $\mu \in \mathbb{R}^n$ is the row-wise shifting factor. Based on the first-order **ARB**, we use Equation 3 to update $\mu$, then sequentially update $\alpha_1$ and $\alpha_2$ by setting $\partial \mathcal{L}_1 / \partial \alpha_1 = 0$ and $\partial \mathcal{L}_1 / \partial \alpha_2 = 0$ respectively, leading to the following formulas:

$$\tilde{\alpha}_1 = \frac{1}{m} \text{diag}(\mathbf{B}_1^\top (\mathbf{W} - \mu_{\text{refine}} - \alpha_2 \mathbf{B}_2)), \quad \tilde{\alpha}_2 = \frac{1}{m} \text{diag}(\mathbf{B}_2^\top (\mathbf{W} - \mu_{\text{refine}} - \tilde{\alpha}_1 \mathbf{B}_1)). \tag{7}$$

The final step is to update the binary matrices $\mathbf{B}_1$ and $\mathbf{B}_2$. The objective of refining $\mathbf{B}_1$ and $\mathbf{B}_2$ is:

$$\widetilde{\mathbf{B}}_1, \widetilde{\mathbf{B}}_2 = \underset{\mathbf{B}_1, \mathbf{B}_2}{\arg\min} ||\mathbf{W} - \mu_{\text{refine}} - \tilde{\alpha}_1 \mathbf{B}_1 - \tilde{\alpha}_2 \mathbf{B}_2||_{\ell_1}. \tag{8}$$

Since $\mathbf{B}_1, \mathbf{B}_2 \in \{+1, -1\}^{n \times m}$, there are only four possible combinations for $(\tilde{\alpha}_1 \mathbf{B}_1 + \tilde{\alpha}_2 \mathbf{B}_2)$. Thus, we construct a candidate vector $\mathbf{V} = \{-\tilde{\alpha}_1 - \tilde{\alpha}_2, -\tilde{\alpha}_1 + \tilde{\alpha}_2, +\tilde{\alpha}_1 - \tilde{\alpha}_2, +\tilde{\alpha}_1 + \tilde{\alpha}_2\} \in \mathbb{R}^4$, then use binary search to find the combination that is closest to $(\mathbf{W} - \mu_{\text{refine}})$. The corresponding elements of $\widehat{\mathbf{B}}_1$ and $\widehat{\mathbf{B}}_2$ are then determined accordingly. Detailed pseudocode is provided in supplementary file.

## 3.2 ARB WITH CALIBRATION DATA (ARB-X)

Although the **ARB** algorithm can effectively reduce the quantization error $\mathcal{L}_1$, we observe that the weight matrix $\mathbf{W}$ operates in conjunction with the input data to produce the output. It means that $\mathcal{L}_1$ alone does not fully capture the true impact of quantization. To address this issue, we introduce calibration data $\mathbf{X}$ and define a new quantization error $\mathcal{L}_2$ as $\mathcal{L}_2 = ||\mathbf{W}\mathbf{X} - \widehat{\mathbf{W}}\mathbf{X}||_F^2$. Based on $\mathcal{L}_2$ and the **ARB** algorithm, we propose an extended algorithm, naming **ARB-X**.

**Reformulation.** However, incorporating calibration data necessitates a large number of matrix multiplications when computing $\mathcal{L}_2$, substantially increasing computational overhead, and often making the combination of calibration data impractical. To address this issue, we reformulate the error computation by decoupling the calibration data and weight matrix as:

$$\mathcal{L}_2 = \langle \mathbf{S}, \mathbf{R}^\top \mathbf{R} \rangle_F = \text{Tr}(\mathbf{R}\mathbf{S}\mathbf{R}^\top), \quad \text{where } \mathbf{S} = \sum_b \mathbf{X}_b^T \mathbf{X}_b, \ \mathbf{R} = \mathbf{W} - \mu - \alpha \mathbf{B}. \tag{9}$$

$\mathbf{X} \in \mathbb{R}^{B \times L \times m}$ denotes the calibration data with batch size $B$, sequence length $L$, and embedding dimension $m$. By compressing the high-dimensional tensor $\mathbf{X}$ into a 2D matrix $\mathbf{S} \in \mathbb{R}^{m \times m}$ and precomputing it, we can significantly reduce the computational overhead. To quantify the efficiency

improvement of our reformulation, we define the speedup ratio $\eta$, which denotes the ratio between the time complexity of the original error computation and that of the revised method. We present the theoretical result in Theorem 2, with the proof provided in the supplementary file.

> **Theorem 2.** *The speedup ratio $\eta$ of the reformulation compared to the original method is*
> $$\eta \propto \frac{1}{k \cdot \left(\frac{1}{n \cdot T} + \frac{1}{B \cdot L}\right)}, \tag{10}$$
> *where $n$ is the hidden dimension of $\mathbf{W}$, $k$ is the block size, and $T$ is the number of iterations.*

Typically, we set $n$ to 4,096, $B$ to 128, $L$ to 2,048, $T$ to 15, and $k$ to 128. Under these circumstances, $\eta$ is proportional to 389, meaning that the reformulated method is approximately 389× faster than the original one. Further details are provided in supplementary file.

**Parameter Update.** By combining the parameter updating strategy with the reformulated $\mathcal{L}_2$, we can derive the parameter update formulas for **ARB-X** by setting $\partial\mathcal{L}_2/\partial\mu = 0$ and $\partial\mathcal{L}_2/\partial\alpha = 0$, which results in the sequential updates of $\mu$ and $\alpha$ respectively:

$$\mu = \frac{\mathbf{1}^\top \mathbf{S}(\mathbf{W} - \alpha\mathbf{B})^\top}{\mathbf{1}^\top \mathbf{S}\mathbf{1}}, \quad \alpha = \frac{\text{diag}(\mathbf{BS}(\mathbf{W} - \mu)^\top)}{\text{diag}(\mathbf{BSB}^\top)}. \tag{11}$$

More details are provided in supplementary file. It is worth noting that, during this process, the matrix $\mathbf{B}$ is not updated. Since $\mathbf{B}$ consists of discrete values (*i.e.*, +1 and -1), it is not possible to update $\mathbf{B}$ directly by setting the partial derivative of $\mathcal{L}_2$ with respect to $\mathbf{B}$ to zero. The pseudocodes for the first-order and second-order **ARB-X** are provided in supplementary file.

### 3.3 ARB ALONG ROW-COLUMN AXES (ARB-RC)

Previous binarization methods use a row-wise scaling factor $\alpha^r$ for weight binarization. However, our analyses of the numerical distribution of the weight matrix $\mathbf{W}$ in LLMs reveal significant deviations across columns, with some columns exhibiting notably larger values (Figure 3). As a result, using a single row-wise scaling factor may not effectively capture the distribution characteristics of LLM parameters. Additionally, the weight distribution shows a mean close to zero, making the redistribution to zero-mean less effective in LLM binarization.

To address this, we propose the **ARB-RC** algorithm, which introduces a column-wise scaling factor $\alpha^c$ to better handle parameter variations across columns, while eliminating the redistribution parameter $\mu$ to enhance compression in LLMs. The row-column binarization process is performed as follows:

$$\alpha^r = \frac{1}{m}\sum_{j=1}^{m}|\mathbf{W}_{\cdot j}|, \quad \alpha^c = \frac{1}{n}\sum_{j=1}^{n}|\frac{\mathbf{W}_{j\cdot}}{\alpha^r_j}|, \quad \mathbf{B} = \text{sign}(\mathbf{W}). \tag{12}$$

Then, we can obtain the binarized matrix as $\widehat{\mathbf{W}} = \alpha^r\alpha^c\mathbf{B}$, where removing $\mu$ while introducing $\alpha^c$ reduces parameters but improves model performance. However, introducing $\alpha^c$ without adopting an alternating parameter update strategy fails to improve performance and can even increase quantization error. Thus, it is necessary to combine $\alpha^c$ with the discussed **ARB** algorithm. In this approach, we optimize the parameters using the quantization error $\mathcal{L}_1$. Although the quantization error $\mathcal{L}_2$ is more aligned with real-world conditions, our analysis shows that incorporating $\mathbf{X}$ in the **ARB-RC** method results in parameter coupling, making optimization difficult (detailed in supplementary file). Thus, based on $\mathcal{L}_1$, we can update $\alpha^r$ and $\alpha^c$ by setting $\partial\mathcal{L}_1/\partial\alpha^r = 0$ and $\partial\mathcal{L}_1/\partial\alpha^c = 0$ respectively:

$$\alpha^r = \frac{\text{diag}(\mathbf{W}(\alpha^c\mathbf{B})^\top)}{\text{diag}((\alpha^c\mathbf{B})(\alpha^c\mathbf{B})^\top)}, \quad \alpha^c = \frac{\text{diag}(\mathbf{W}^\top(\alpha^r\mathbf{B}))}{\text{diag}((\alpha^r\mathbf{B})^\top(\alpha^r\mathbf{B}))}. \tag{13}$$

The first-order and second-order pseudocodes of the **ARB-RC** are provided in supplementary file.

### 3.4 COLUMN-GROUP BITMAP (CGB)

Inspired by BiLLM (Huang et al., 2024), we partition the entire set of weights into salient and non-salient columns, and apply higher-bit representation, *i.e.,* second-order binarization, to the salient weights. However, different from BiLLM, we not only divide the non-salient weights into sparse and concentrated groups but also divide salient weights in a similar manner. This approach allows for more efficient use of both column bitmap and group bitmap, as shown in Figure 5.

To identify the sensitivity of weights, *i.e.,* salient weights, we follow well-established PTQ methods by utilizing the Hessian matrix as a standard criterion. The sensitivity is computed as $s_i = w_i^2/[\mathbf{H}^{-1}]_{ii}^2$,

Table 1: Perplexity of RTN, GPTQ, PB-LLM, BiLLM, and our methods on **OPT** family. The columns represent the perplexity results on **WikiText2** datasets with different model sizes.

| Method | Block Size | Weight Bits | 1.3B | 2.7B | 6.7B | 13B | 30B | 66B |
|---|---|---|---|---|---|---|---|---|
| Full Precision | - | 16.00 | 14.62 | 12.47 | 10.86 | 10.13 | 9.56 | 9.34 |
| RTN | - | 3.00 | 13,337.38 | 15,594.72 | 5,797.32 | 3,357.01 | 1,566.00 | 6,126.09 |
| GPTQ | 128 | 3.00 | 16.45 | 13.61 | 11.31 | 10.47 | 9.71 | 10.55 |
| RTN | - | 2.00 | 11,272.65 | 9,505.76 | 28,363.14 | 194,086.78 | 169,616.47 | 1,165,864.25 |
| GPTQ | 128 | 2.00 | 121.64 | 59.53 | 20.81 | 20.05 | 13.04 | 46.38 |
| RTN | - | 1.00 | 17,165.72 | 36,516.69 | 11,550.91 | 69,863,488.00 | 6,485.99 | 184,796.30 |
| GPTQ | 128 | 1.00 | 8,719.58 | 11,700.13 | 6,633.13 | 1,743,929.88 | 14,083.15 | 11,045.36 |
| PB-LLM | 128 | 1.70 | 239.81 | 278.27 | 144.25 | 74.59 | 28.30 | 27.66 |
| BiLLM | 128 | 1.11 | 69.05 | 48.61 | 47.65 | 18.75 | 13.86 | 12.05 |
| **ARB-LLM$_X$** | **128** | **1.11** | **45.40** | **34.37** | **20.07** | **15.47** | **12.36** | **11.23** |
| **ARB-LLM$_{RC}$** | **128** | **1.11** | **26.63** | **19.84** | **14.92** | **12.92** | **11.12** | **10.30** |

where $\mathbf{H}$ represents the Hessian matrix for each layer, and $w_i$ represents the original value of each weight element. Weight columns with higher $s_i$ are selected as salient columns, which are then marked using the salient **column bitmap**. For more details, please refer to Huang et al. (2024).

For non-salient columns, BiLLM further divides them into sparse and concentrated groups based on their magnitude, marking them using a **group bitmap**. Although this grouping strategy significantly reduces the quantization error, it can be further refined since some regions of the group bitmap are underutilized. As shown on the left side of Figure 5, the salient columns of the group bitmap remain unused. Thus, to better utilize the space of the group bitmap, we optimize the combination of the column bitmap and group bitmap. Specifically, we further categorize the salient weights into sparse and concentrated groups, which improve the quantization accuracy of salient weights without increasing bitmap storage. Our combination format is defined as follows:

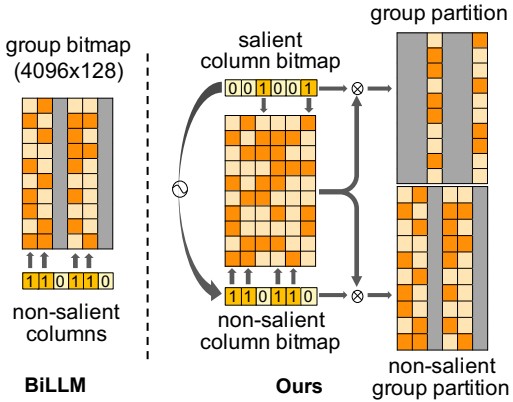

Figure 5: Comparison between BiLLM and our combination of column and group bitmaps.

$$\mathbf{G_s} = \mathbf{1}_n \mathbf{C_s}^\top \odot \mathbf{G}, \quad \mathbf{G_{ns}} = \mathbf{1}_n \mathbf{C_{ns}}^\top \odot \mathbf{G}, \tag{14}$$

where $\mathbf{G_s}$ and $\mathbf{G_{ns}}$ represent group bitmaps for salient and non-salient weights, respectively. $\mathbf{C}_s$ indicates the salient columns, while $\mathbf{C_{ns}} = \neg \mathbf{C_s}$ indicates the non-salient columns. We extend the column bitmap along the row axis and then perform element-wise multiplication with the group bitmap to obtain the final partitions. Experiments demonstrate that our **C**olumn-**G**roup **B**itmap (**CGB**) further enhances the quantization performance when applied to ARB algorithms. Additionally, following BiLLM, we adopt the block-wise compensation (Frantar et al., 2023; Frantar & Alistarh, 2022) to mitigate quantization errors. For further details, please refer to their papers.

## 4 EXPERIMENTS

### 4.1 SETUP

All the experiments are conducted with PyTorch (Paszke et al., 2019b) and Huggingface (Paszke et al., 2019a) on a single NVIDIA A800-80GB GPU. We implement 15 iterations for ARB-LLM$_X$ and ARB-LLM$_{RC}$ to ensure the convergence of binarization parameters. Following Frantar et al. (2023) and Huang et al. (2024), we use 128 samples from C4 (Raffel et al., 2020) dataset as calibration data.

**Models and Datasets.** We conduct extensive experiments on the LLaMA, LLaMA-2, and LLaMA-3 families (Touvron et al., 2023), the OPT family (Zhang et al., 2022), and *instruction-tuned* LLMs Vicuna (Chiang et al., 2023). To evaluate the effectiveness of our proposed ARB-LLM$_X$ (ARB-X + CGB) and ARB-LLM$_{RC}$ (ARB-RC + CGB), we measure the perplexity of LLM's outputs on WikiText2 (Merity et al., 2017), PTB (Marcus et al., 1994), as well as a part of the C4 (Raffel et al., 2020) data. Moreover, we also evaluate the accuracy for 7 zero-shot QA datasets: ARC-c (Clark et al., 2018), ARC-e (Clark et al., 2018), BoolQ (Clark et al., 2019), Hellaswag (Zellers et al., 2019), OBQA (Mihaylov et al., 2018), PIQA (Bisk et al., 2020), and Winogrande (Sakaguchi et al., 2020).

Table 2: Perplexity of RTN, GPTQ, PB-LLM, BiLLM, and our methods on LLaMA family. The columns represent the perplexity results on **WikiText2** dataset with different model sizes. N/A: LLaMA-2 lacks a 30B version, and LLaMA-3 lacks both 13B and 30B versions. *: LLaMA has a 65B version, while both LLaMA-2 and LLaMA-3 have 70B versions.

| Model | Method | Block Size | Weight Bits | 7B/8B* | 13B | 30B | 65B/70B* |
|---|---|---|---|---|---|---|---|
| **LLaMA** | Full Precision | - | 16.00 | 5.68 | 5.09 | 4.10 | 3.53 |
| | RTN | - | 3.00 | 25.54 | 11.40 | 14.89 | 10.59 |
| | GPTQ | 128 | 3.00 | 8.63 | 5.67 | 4.87 | 4.17 |
| | RTN | - | 2.00 | 106,767.34 | 57,409.93 | 26,704.36 | 19,832.87 |
| | GPTQ | 128 | 2.00 | 129.19 | 20.46 | 15.29 | 8.66 |
| | RTN | - | 1.00 | 168,388.00 | 1,412,020.25 | 14,681.76 | 65,253.24 |
| | GPTQ | 128 | 1.00 | 164,471.78 | 131,505.41 | 10,339.15 | 20,986.16 |
| | PB-LLM | 128 | 1.70 | 82.76 | 44.93 | 23.72 | 12.81 |
| | BiLLM | 128 | 1.09 | 49.79 | 14.58 | 9.90 | 8.37 |
| | **ARB-LLM$_X$** | **128** | **1.09** | **21.81** | **11.20** | **8.66** | **7.27** |
| | **ARB-LLM$_{RC}$** | **128** | **1.09** | **14.03** | **10.18** | **7.75** | **6.56** |
| **LLaMA-2** | Full Precision | - | 16.00 | 5.47 | 4.88 | N/A | 3.32 |
| | RTN | - | 3.00 | 542.80 | 10.68 | N/A | 7.53 |
| | GPTQ | 128 | 3.00 | 6.44 | 5.46 | N/A | 3.88 |
| | RTN | - | 2.00 | 17,788.94 | 51,145.61 | N/A | 26,066.13 |
| | GPTQ | 128 | 2.00 | 52.22 | 23.63 | N/A | 8.18 |
| | RTN | - | 1.00 | 157,058.34 | 47,902.32 | N/A | 160,389.91 |
| | GPTQ | 128 | 1.00 | 59,758.69 | 22,926.54 | N/A | 14,219.35 |
| | PB-LLM | 128 | 1.70 | 66.41 | 236.40 | N/A | 28.37 |
| | BiLLM | 128 | 1.08 | 32.31 | 21.35 | N/A | 13.32 |
| | **ARB-LLM$_X$** | **128** | **1.08** | **21.61** | **14.86** | **N/A** | **7.88** |
| | **ARB-LLM$_{RC}$** | **128** | **1.08** | **16.44** | **11.85** | **N/A** | **6.16** |
| **LLaMA-3** | Full Precision | - | 16.00 | 6.14 | N/A | N/A | 2.86 |
| | RTN | - | 3.00 | 2,194.98 | N/A | N/A | 13,592.69 |
| | GPTQ | 128 | 3.00 | 18.68 | N/A | N/A | 6.65 |
| | RTN | - | 2.00 | 1,335,816.13 | N/A | N/A | 481,927.66 |
| | GPTQ | 128 | 2.00 | 1,480.43 | N/A | N/A | 82.23 |
| | RTN | - | 1.00 | 1,353,698.38 | N/A | N/A | 375,658.34 |
| | GPTQ | 128 | 1.00 | 1,121,260.50 | N/A | N/A | 130,516.50 |
| | PB-LLM | 128 | 1.70 | 73.08 | N/A | N/A | 22.96 |
| | BiLLM | 128 | 1.06 | 55.80 | N/A | N/A | 66.30 |
| | **ARB-LLM$_X$** | **128** | **1.06** | **31.98** | **N/A** | **N/A** | **14.15** |
| | **ARB-LLM$_{RC}$** | **128** | **1.06** | **27.42** | **N/A** | **N/A** | **11.10** |

**Baselines.** We mainly compare our ARB series with BiLLM (Huang et al., 2024), the SOTA PTQ approach on binary LLMs. Other recent PTQ algorithms, such as RTN (round-to-nearest), GPTQ (Frantar et al., 2023), and PB-LLM (Shang et al., 2024) are also selected.

## 4.2 MAIN RESULTS

We follow BiLLM to report the average bit-width of all methods, where our methods have the same bit-width as BiLLM. Table 1 presents the perplexity comparison of the OPT family across different model sizes. It can be observed

Table 3: Perplexity of GPTQ, PB-LLM, BiLLM, and our methods on **Vicuna** family. The columns represent the perplexity results on **WikiText2** datasets with different model sizes.

| Method | Block Size | Weight Bits | 7B | 13B |
|---|---|---|---|---|
| Full Precision | - | 16.00 | 6.34 | 5.57 |
| GPTQ | 128 | 2.00 | 688.08 | 37.97 |
| PB-LLM | 128 | 1.70 | 58.68 | 2,506.44 |
| BiLLM | 128 | 1.08 | 39.36 | 43.39 |
| **ARB-LLM$_X$** | **128** | **1.08** | **22.79** | **13.76** |
| **ARB-LLM$_{RC}$** | **128** | **1.08** | **17.60** | **13.38** |

that both ARB-LLM$_X$ and ARB-LLM$_{RC}$ significantly outperform SOTA BiLLM, and reduce the perplexity by up to **68.7%** without increasing weight bit-width. Table 2 presents the perplexity comparison on LLaMA1&2&3 families, which also suggests the superior performance of our ARB-LLM. It is noteworthy that ARB-LLM$_{RC}$ outperforms RTN with 3-bit quantization on some models, such as the LLaMA1&3 families, LlaMA2-70B model, as well as OPT family. Similarly, ARB-LLM$_{RC}$

Table 4: Ablation study on LLaMA-7B, where all ARB methods are equipped with CGB except for ablation (b). Results are measured by perplexity, with final results highlighted in **bold**.

(a) Effectiveness of two advanced variants

| Method | Calibration update | Row-column update | WikiText2 ↓ | C4 ↓ |
|---|---|---|---|---|
| BiLLM | - | - | 49.79 | 46.96 |
| ARB | ✗ | ✗ | 22.67 | 26.44 |
| ARB-LLM$_X$ | ✓ | ✗ | **21.81** | **22.73** |
| ARB-LLM$_{RC}$ | ✗ | ✓ | **14.03** | **17.92** |

(b) Effectiveness of CGB

| Method | CGB | WikiText2 ↓ | C4 ↓ |
|---|---|---|---|
| BiLLM | - | 49.79 | 46.96 |
| ARB-LLM$_X$ | ✗ | 26.29 | 27.11 |
| ARB-LLM$_X$ | ✓ | **21.81** | **22.73** |
| ARB-LLM$_{RC}$ | ✗ | 15.85 | 19.42 |
| ARB-LLM$_{RC}$ | ✓ | **14.03** | **17.92** |

(c) Study of decoupling column and group bitmaps

| Method | Column bitmap | Group bitmap | WikiText2 ↓ | C4 ↓ |
|---|---|---|---|---|
| ARB-LLM$_{RC}$ | ✗ | ✗ | 10,942.45 | 11,032.93 |
| ARB-LLM$_{RC}$ | ✓ | ✗ | 369.20 | 205.56 |
| ARB-LLM$_{RC}$ | ✗ | ✓ | 920.42 | 572.69 |
| ARB-LLM$_{RC}$ | ✓ | ✓ | **14.03** | **17.92** |

(d) Study of ARB-LLM$_X$ calibration set size

| Method | Calibration set size | WikiText2 ↓ | C4 ↓ |
|---|---|---|---|
| BiLLM | 128 | 49.79 | 46.96 |
| ARB-LLM$_X$ | 64 | 24.79 | 25.11 |
| ARB-LLM$_X$ | 128 | **21.81** | **22.73** |
| ARB-LLM$_X$ | 256 | 21.88 | 24.28 |

(e) Study of ARB-LLM iteration number

| Method | #Iteration | WikiText2 ↓ |
|---|---|---|
| BiLLM | 0 | 49.79 |
| ARB-LLM$_X$ | 1 / 3 / 15 | 22.59 / 21.12 / **21.81** |
| ARB-LLM$_{RC}$ | 1 / 3 / 15 | 15.23 / 14.34 / **14.03** |

(f) Study of ARB-LLM group number

| Method | #Group | WikiText2 ↓ | C4 ↓ |
|---|---|---|---|
| BiLLM | 2 | 49.79 | 46.96 |
| ARB-LLM$_X$ | 2 / 4 | **21.81** / 6.55 | **22.73** / 8.56 |
| ARB-LLM$_{RC}$ | 2 / 4 | **14.03** / 12.77 | **17.92** / 16.06 |

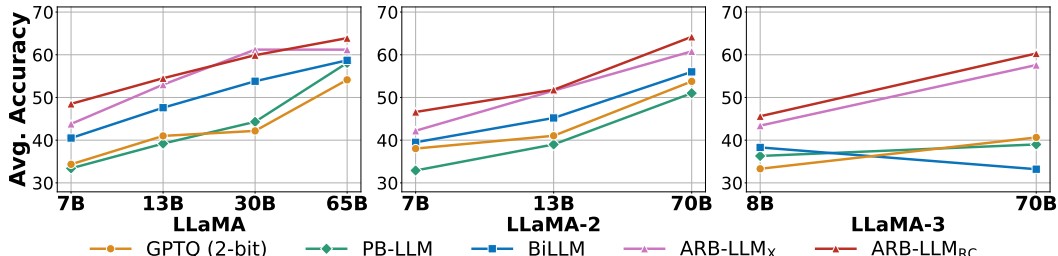

Figure 6: Average accuracy of 7 zero-shot QA datasets on LLaMA1&2&3 families.

also surpasses GPTQ with 3-bit quantization on OPT-66B model. For *instruction-tuned* Vicuna comparison shown in Table 3, ARB-LLM$_X$ and ARB-LLM$_{RC}$ also show superior performance, surpassing SOTA binary PTQ method BiLLM for a large margin. Regarding average accuracy on QA datasets, ARB-LLM$_X$ and ARB-LLM$_{RC}$ both significantly outperform previous methods, as shown in Figure 6. More results are provided in the supplementary file.

## 4.3 ABLATION STUDY

**Effectiveness of Advanced Variants.** To validate the effectiveness of our advanced variants ARB-LLM$_X$ and ARB-LLM$_{RC}$, we compare them with the vanilla ARB algorithm in Table 4a. First, we observe that the vanilla ARB already significantly outperforms BiLLM. Furthermore, by introducing either the calibration update or the row-column update to the binarization process, performance is further improved. This demonstrates that our advanced variants, ARB-LLM$_X$ and ARB-LLM$_{RC}$, can further enhance the performance of binary LLMs based on ARB.

**Effectiveness of CGB.** To demonstrate the effectiveness of our column-group bitmap (CGB), we conduct an ablation study in Table 4b. In this study, the absence of CGB does not imply the exclusion of partitioning but rather the use of the partitioning strategy used by BiLLM. The results show that CGB further enhances the performance of both ARB-LLM$_X$ and ARB-LLM$_{RC}$. Notably, even when using BiLLM's partitioning strategy, our methods significantly outperform BiLLM.

**Column Bitmap and Group Bitmap.** We use a column bitmap to differentiate between salient and non-salient weights, and a group bitmap to separate weights based on their magnitude. The combination of column and group bitmaps creates four distinct zones. As shown in Table 4c, we explore the effect of decoupling this combination by using either the column bitmap or the group bitmap individually. It is evident that using the column bitmap or group bitmap only will result in a significant performance drop. Omitting both column bitmap and group bitmap entirely (*i.e.,* #group=1), which reduces the method to naive binarization, leads to complete failure.

**Calibration Set Size.** Similar to other PTQ methods, our ARB-LLM requires a small calibration set of just 128 samples. We further incorporate the calibration data into the update of binarization parameters in ARB-LLM$_X$. To explore the effect of calibration set size on performance, we compare results using different set sizes, as shown in Table 4d. It can be observed that using fewer calibration samples (*e.g.,* 64) results in a performance drop, while increasing the calibration set size from 128 to 256 yields similar results. This indicates that our ARB-LLM$_X$ requires only a small calibration set. Even with just 64 samples, ARB-LLM$_X$ significantly outperforms the baseline BiLLM.

**ARB Iteration Number.** We use 15 iterations for the main results (Table 1, Table 2, Table 3, and Figure 6), as all parameters have fully converged. To explore the impact of different iteration numbers, we compare results using 1, 3, and 15 iterations in Table 4e. As can be seen, regardless of the iteration number, the perplexity of ARB-LLM$_X$ and ARB-LLM$_{RC}$ significantly outperforms the baseline BiLLM. Increasing the iteration number further reduces perplexity, yet they can achieve superior results even with just one iteration. Additionally, we visualize the changes in the scaling factor $\alpha$ throughout the alternating iterations to provide further insights in supplementary file.

**Group Number.** Following BiLLM (Huang et al., 2024), we introduce an additional bitmap for grouping weights, which has been demonstrated to enhance performance. To explore the impact of group size, we expand the group bitmap from a 1-bit to a 2-bit system, increasing the number of groups from 2 to 4. As shown in Table 4f, increasing the number of groups leads to better performance, especially for ARB-LLM$_X$, which outperforms ARB-LLM$_{RC}$ with the same number of groups. Yet, this also results in extra storage (about 0.8 GB for LLaMA-7B). In contrast, using only one group (*i.e.,* the first row of Table 4c) results in total failure. Given the additional storage overhead, the 2-group configuration strikes a good balance between performance and memory efficiency.

## 4.4 TIME AND MEMORY ANALYSES

**Time Comparison.** As a binary PTQ framework, ARB-LLM eliminates the need for fine-tuning. The alternating algorithm requires more computation to align the distribution progressively, yet this overhead is acceptable. In Table 5, ARB-LLM$_{RC}$ with 15 iterations requires only 21 more minutes than BiLLM, while ARB-LLM$_{RC}$ (without CGB) requires only 3 more minutes than BiLLM using just 1 iteration. The combination of CGB results in an increase of time overhead, due to the percentile search for optimal splitting.

Table 5: Time comparison between BiLLM and our ARB-LLM methods on LLaMA-7B.

| Method | CGB | #Iter=1 | #Iter=3 | #Iter=15 |
|---|---|---|---|---|
| BiLLM | - | 45 min (#Iter=0) | | |
| ARB-LLM$_X$ | ✗ | 52 min | 59 min | 70 min |
| ARB-LLM$_X$ | ✓ | 72 min | 78 min | 88 min |
| ARB-LLM$_{RC}$ | ✗ | 48 min | 49 min | 53 min |
| ARB-LLM$_{RC}$ | ✓ | 67 min | 68 min | 76 min |

**Memory Comparison.** Following PB-LLM and BiLLM, we present the memory usage with Raw bitmap / CSR compressed bitmap in Table 6. ARB-LLM$_{RC}$, which replaces the row-wise mean with a column-wise scaling factor, achieves a higher compression ratio along with better performance. Although the refined column-group bitmap (CGB) strategy requires more memory due to more scaling factors, the combination of ARB-RC and CGB still results in lower storage requirements than BiLLM, while delivering outstanding performance. As shown in Table 6, ARB-LLM$_{RC}$ with or without CGB both require less storage than previous methods. The computation formulas can be found in supplementary file.

Table 6: Memory (GB, Raw bitmap / CSR bitmap) comparison between FP16, PB-LLM, BiLLM, and our ARB-LLM methods.

| Method | CGB | LLaMA-7B | LLaMA-13B |
|---|---|---|---|
| FP16 | - | 13.48 | 26.03 |
| PB-LLM | - | 2.91 / 2.21 | 5.33 / 3.96 |
| BiLLM | - | 2.93 / 2.19 | 5.36 / 3.92 |
| ARB-LLM$_X$ | ✗ | 2.93 / 2.19 | 5.36 / 3.92 |
| ARB-LLM$_X$ | ✓ | 3.23 / 2.49 | 5.95 / 4.51 |
| ARB-LLM$_{RC}$ | ✗ | **2.63 / 1.89** | **4.77 / 3.33** |
| ARB-LLM$_{RC}$ | ✓ | 2.83 / 2.09 | 5.17 / 3.73 |

## 5 CONCLUSION

In this work, we propose ARB-LLM, a series of alternating refined binarization (ARB) methods for LLMs. Through the analyses of the distribution shift between binarized and full-precision weights, we propose an alternating refinement of binarization parameters to progressively align the weight distribution. Moreover, we extend the basic ARB by equipping the calibration data and scaling along row-column axes, resulting in ARB-X and ARB-RC respectively. Additionally, we propose a refined strategy to better combine the salient column bitmap and group bitmap. Our experiments on multiple open-source LLM families show that the final models ARB-LLM$_X$ and ARB-LLM$_{RC}$ can further push the performance boundary from the SOTA binary PTQ methods.

## ETHICS STATEMENT

The research conducted in the paper conforms, in every respect, with the ICLR Code of Ethics.

## REPRODUCIBILITY STATEMENT

We have provided implementation details in Sec. 4. We will also release all the code and models.

## ACKNOWLEDGMENTS

This work was supported by Shanghai Municipal Science and Technology Major Project (2021SHZDZX0102) and the Fundamental Research Funds for the Central Universities. It was also supported by Lenovo Research (202411SJTU01-LR22).

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
