# ARB-LLM: ALTERNATING REFINED BINARIZATIONS FOR LARGE LANGUAGE MODELS

**Zhiteng Li**[1*], **Xianglong Yan**[1*], **Tianao Zhang**[1], **Haotong Qin**[2], **Dong Xie**[3],
**Jiang Tian**[3], **Zhongchao Shi**[3], **Linghe Kong**[1†], **Yulun Zhang**[1†], **Xiaokang Yang**[1]
[1]Shanghai Jiao Tong University, [2]ETH Zürich, [3]Lenovo Research

CONTENTS

---

[*]Equal contribution

[†]Corresponding authors: Linghe Kong, linghe.kong@sjtu.edu.cn, Yulun Zhang, yulun100@gmail.com

# A  FIRST-ORDER ARB-X AND ARB-RC

## A.1  FIRST-ORDER ARB-X

First-order Alternating Refined Binarization with Calibration Data (**ARB-X**) is based on the weight-activation quantization error $\mathcal{L} = ||\mathbf{WX} - \widehat{\mathbf{W}}\mathbf{X}||_F^2$.

**Rewritten weight-activation quantization error**  We first rewrite the quantization error $\mathcal{L}$ to decouple $\mathbf{W}$ and $\mathbf{X}$, reducing the computational cost when calculating the quantization error. We define $\widetilde{\mathbf{W}}$ as $\widetilde{\mathbf{W}} = \mathbf{W} - \mu$. Then we rewrite the quantization error as

$$\mathcal{L} = ||\mathbf{WX} - \widehat{\mathbf{W}}\mathbf{X}||_F^2 \tag{1}$$

$$= ||\mathbf{X}(\widetilde{\mathbf{W}} - \alpha\mathbf{B})^\top||_F^2 \tag{2}$$

$$= \sum_i \sum_j (\sum_b \sum_k (\mathbf{X}_b)_{ik}(\widetilde{\mathbf{W}}_{jk} - \alpha_j\mathbf{B}_{jk}))^2. \tag{3}$$

Then, we define the residual matrix as $\mathbf{R} = \mathbf{W} - \mu - \alpha\mathbf{B}$ and further simplify $\mathcal{L}$:

$$\mathcal{L} = \sum_i \sum_j (\sum_b \sum_k (\mathbf{X}_b)_{ik}\mathbf{R}_{jk})^2 \tag{4}$$

$$= \sum_i \sum_j (\sum_b \sum_k \sum_l (\mathbf{X}_b)_{ik}(\mathbf{X}_b)_{il}\mathbf{R}_{jk}\mathbf{R}_{jl}) \tag{5}$$

$$= \sum_k \sum_l (\sum_b \sum_i (\mathbf{X}_b)_{ik}(\mathbf{X}_b)_{il})(\sum_j \mathbf{R}_{jk}\mathbf{R}_{jl}). \tag{6}$$

After that, we define the matrix $\mathbf{S}$ using the following formula:

$$\mathbf{S}_{kl} = \sum_b \sum_i (\mathbf{X}_b)_{ik}(\mathbf{X}_b)_{il}, \tag{7}$$

where $k = 1, 2, \ldots, m$, $l = 1, 2, \ldots, m$. Then we obtain the final simplified $\mathcal{L}$ as

$$\mathcal{L} = \langle \mathbf{S}, \mathbf{R}^\top\mathbf{R} \rangle_F = \text{Tr}(\mathbf{R}\mathbf{S}\mathbf{R}^\top). \tag{8}$$

**Parameter Update**  We use the quantization error $\mathcal{L}$ to update $\mu$:

$$\mathcal{L} = \sum_k \sum_l \mathbf{S}_{kl} \sum_j \mathbf{R}_{jk}\mathbf{R}_{jl} \tag{9}$$

$$= \sum_k \sum_l \mathbf{S}_{kl} \sum_j (\widetilde{\mathbf{W}}_{jk}\widetilde{\mathbf{W}}_{jl} - \alpha_j(\mathbf{B}_{jk}\widetilde{\mathbf{W}}_{jl} + \mathbf{B}_{jl}\widetilde{\mathbf{W}}_{jk}) + \alpha_j^2\mathbf{B}_{jk}\mathbf{B}_{jl}) \tag{10}$$

$$= \sum_k \sum_l \mathbf{S}_{kl} \sum_j ((\mathbf{W}_{jk} - \mu_j)(\mathbf{W}_{jl} - \mu_j) - \alpha_j(\mathbf{B}_{jk}(\mathbf{W}_{jl} - \mu_j) \tag{11}$$

$$+ \mathbf{B}_{jl}(\mathbf{W}_{jk} - \mu_j)) + \alpha_j^2\mathbf{B}_{jk}\mathbf{B}_{jl}). \tag{12}$$

To obtain the optimal solution for $\mu$, we take the partial derivative of $\mathcal{L}$ with respect to $\mu_j$, where $j = 1, 2, \ldots, n$:

$$\frac{\partial \mathcal{L}}{\partial \mu_j} = \sum_k \sum_l \mathbf{S}_{kl}(-\mathbf{W}_{jl} - \mathbf{W}_{jk} + 2\mu_j + \alpha_j\mathbf{B}_{jk} + \alpha_j\mathbf{B}_{jl}). \tag{13}$$

We set $\frac{\partial \mathcal{L}}{\partial \mu_j} = 0$ to get the optimal solution for $\mu_j$:

$$\mu_j = \frac{\sum_k \sum_l \mathbf{S}_{kl}(\mathbf{W}_{jk} - \alpha_j\mathbf{B}_{jk} + \mathbf{W}_{jl} - \alpha_j\mathbf{B}_{jl})}{2\sum_k \sum_l \mathbf{S}_{kl}}, \quad \text{where } j = 1, 2, \ldots, n. \tag{14}$$

Then, we define the matrix $\mathbf{P}$ as:

$$\mathbf{P}_{kl} = \mathbf{W}_{jk} - \alpha_j \mathbf{B}_{jl}, \quad \text{where } k = 1, 2, \ldots, m, l = 1, 2, \ldots, m. \tag{15}$$

After that, we can simplify $\mu_j$ as

$$\mu_j = \frac{\sum_k \sum_l (\mathbf{S} \odot (\mathbf{P} + \mathbf{P}^\top))_{kl}}{2 \sum_k \sum_l \mathbf{S}_{kl}}, \quad \text{where } j = 1, 2, \ldots, n. \tag{16}$$

Since $\mathbf{S}$ is symmetric, we can further simplify the above equation as:

$$\mu_j = \frac{\sum_k \sum_l (\mathbf{S} \odot \mathbf{P})_{kl}}{\sum_k \sum_l \mathbf{S}_{kl}}, \quad \text{where } j = 1, 2, \ldots, n. \tag{17}$$

We can also express $\mu$ in a more compact vector form:

$$\mu = \frac{\mathbf{1}^\top \mathbf{S} (\mathbf{W} - \alpha \mathbf{B})^\top}{\mathbf{1}^\top \mathbf{S} \mathbf{1}}. \tag{18}$$

Similarly, we use the same quantization error to update $\alpha$:

$$\mathcal{L} = \sum_k \sum_l \mathbf{S}_{kl} \sum_j \mathbf{R}_{jk} \mathbf{R}_{jl} \tag{19}$$

$$= \sum_k \sum_l \mathbf{S}_{kl} \sum_j (\widetilde{\mathbf{W}}_{jk} \widetilde{\mathbf{W}}_{jl} - \alpha_j (\mathbf{B}_{jk} \widetilde{\mathbf{W}}_{jl} + \mathbf{B}_{jl} \widetilde{\mathbf{W}}_{jk}) + \alpha_j^2 \mathbf{B}_{jk} \mathbf{B}_{jl}). \tag{20}$$

To obtain the optimal solution for $\alpha$, we take the partial derivative of $\mathcal{L}$ with respect to $\alpha_j$, where $j = 1, 2, \ldots, n$:

$$\frac{\partial \mathcal{L}}{\partial \alpha_j} = \sum_k \sum_l \mathbf{S}_{kl} (2 \mathbf{B}_{jk} \mathbf{B}_{jl} \alpha_j - (\mathbf{B}_{jk} \widetilde{\mathbf{W}}_{jl} + \mathbf{B}_{jl} \widetilde{\mathbf{W}}_{jk})). \tag{21}$$

We set $\frac{\partial \mathcal{L}}{\partial \alpha_j} = 0$ to get the optimal solution for $\alpha_j$:

$$\alpha_j = \frac{\sum_k \sum_l \mathbf{S}_{kl} (\mathbf{B}_{jk} \widetilde{\mathbf{W}}_{jl} + \mathbf{B}_{jl} \widetilde{\mathbf{W}}_{jk})}{2 \sum_k \sum_l \mathbf{S}_{kl} (\mathbf{B}_{jk} \mathbf{B}_{jl})}, \quad \text{where } j = 1, 2, \ldots, n. \tag{22}$$

Then, we define the matrix $\mathbf{U}$ and $\mathbf{V}$ as:

$$\mathbf{U}_{kl} = \mathbf{B}_{jk} \widetilde{\mathbf{W}}_{jl}, \ \mathbf{V}_{kl} = \mathbf{B}_{jk} \mathbf{B}_{jl}, \tag{23}$$

where $k = 1, 2, \ldots, m, l = 1, 2, \ldots, m$. After that, we simplify $\alpha_j$ using $\mathbf{U}$ and $\mathbf{V}$:

$$\alpha_j = \frac{\sum_k \sum_l (\mathbf{S} \odot (\mathbf{U} + \mathbf{U}^\top))_{kl}}{2 \sum_k \sum_l (\mathbf{S} \odot \mathbf{V})_{kl}}. \tag{24}$$

Since $\mathbf{S}$ is symmetric, we can further simplify the above equation as

$$\alpha_j = \frac{\sum_k \sum_l (\mathbf{S} \odot \mathbf{U})_{kl}}{\sum_k \sum_l (\mathbf{S} \odot \mathbf{V})_{kl}}. \tag{25}$$

We can also express $\alpha$ in a more compact vector form:

$$\alpha = \frac{\text{diag}(\mathbf{B} \mathbf{S} (\mathbf{W} - \mu)^\top)}{\text{diag}(\mathbf{B} \mathbf{S} \mathbf{B}^\top)}. \tag{26}$$

For the detailed pseudocode of first-order ARB-X, see Algorithm 1.

---

**Algorithm 1** First-Order **A**lternating **R**efined **B**inarization with Calibration Data

---

func ARB-X[1]$(\mathbf{W}, \mathbf{M}, \mathbf{X}, T)$
**Input:** $\mathbf{W} \in \mathbb{R}^{n \times m}$ - full-precision weight
      $\mathbf{M} \in \mathbb{R}^{n \times m}$ - group mask
      $\mathbf{X} \in \mathbb{R}^{B \times L \times m}$ - calibration data
      $T$ - iteration rounds
**Output:** $\widehat{\mathbf{W}} \in \mathbb{R}^{n \times m}$

1: $\mathbf{S} \coloneqq \text{X2S}(\mathbf{X})$ // $\mathbf{S} \in \mathbb{R}^{m \times m}$
2: $\widehat{\mathbf{W}}, \alpha, \mathbf{B}, \mu \coloneqq \text{binary}(\mathbf{W}, \mathbf{M})$
3: **for** $iter = 1, 2, \ldots, T$ **do**
4:     $\mu \leftarrow \text{refine\_}\mu(\mathbf{S}, \mathbf{W}, \mathbf{B}, \alpha, \mathbf{M})$
5:     $\alpha \leftarrow \text{refine\_}\alpha(\mathbf{S}, \mathbf{W}, \mu, \mathbf{B}, \mathbf{M})$
6:     $\widehat{\mathbf{W}} \leftarrow (\alpha \cdot \mathbf{B} + \mu) \odot \mathbf{M}$
7: **end for**
8: **return** $\widehat{\mathbf{W}}$

func binary $(\mathbf{W}, \mathbf{M})$

1: $\mu \coloneqq \frac{1}{m} \sum_{j=1}^{m} (\mathbf{W} \odot \mathbf{M})_{.j}$
2: $\widetilde{\mathbf{W}} \coloneqq \mathbf{W} - \mu$
3: $\alpha \coloneqq \frac{1}{m} \sum_{j=1}^{m} |(\widetilde{\mathbf{W}} \odot \mathbf{M})_{.j}|$
4: $\mathbf{B} \coloneqq \text{sign}(\widetilde{\mathbf{W}} \odot \mathbf{M})$
5: $\widehat{\mathbf{W}} \coloneqq \alpha \cdot \mathbf{B} + \mu$
6: **return** $\widehat{\mathbf{W}}, \alpha, \mathbf{B}, \mu$

func X2S $(\mathbf{X})$

1: **for** $b = 1, 2, \ldots B$ **do**
2:     **for** $k = 1, 2, \ldots, m; l = 1, 2, \ldots, m$ **do**

3:       $\mathbf{S}_{kl} = \sum_b \sum_i (\mathbf{X}_b)_{ik} (\mathbf{X}_b)_{il}$
4:     **end for**
5: **end for**
6: **return** $\mathbf{S}$

func refine\_$\mu$ $(\mathbf{S}, \mathbf{W}, \mathbf{B}, \alpha, \mathbf{M})$

1: **for** $i = 1, 2, \ldots, n$ **do**
2:     **for** $k = 1, 2, \ldots, m; l = 1, 2, \ldots, m$ **do**
3:       $\mathbf{P}_{kl} \coloneqq \mathbf{W}_{ik} - \alpha_i \mathbf{B}_{il} \cdot \mathbf{M}_{il}$
4:     **end for**
5:     $num \coloneqq \sum_k \sum_l (\mathbf{S} \odot \mathbf{P})_{kl}$
6:     $den \coloneqq \sum_k \sum_l \mathbf{S}_{kl} + \epsilon$
7:     $\mu_i \coloneqq \frac{num}{den}$
8: **end for**
9: **return** $\mu$

func refine\_$\alpha$ $(\mathbf{S}, \mathbf{W}, \mu, \mathbf{B}, \mathbf{M})$

1: $\widetilde{\mathbf{W}} \coloneqq \mathbf{W} - \mu$
2: **for** $i = 1, 2, \ldots, n$ **do**
3:     **for** $k = 1, 2, \ldots, m; l = 1, 2, \ldots, m$ **do**
4:       $\mathbf{U}_{kl} \coloneqq (\mathbf{B}_{ik} \cdot \mathbf{M}_{ik}) \widetilde{\mathbf{W}}_{il}$
5:       $\mathbf{V}_{kl} \coloneqq (\mathbf{B}_{ik} \cdot \mathbf{M}_{ik}) \mathbf{B}_{il}$
6:     **end for**
7:     $num \coloneqq \sum_k \sum_l (\mathbf{S} \odot \mathbf{U})_{kl}$
8:     $den \coloneqq \sum_k \sum_l (\mathbf{S} \odot \mathbf{V})_{kl} + \epsilon$
9:     $\alpha_i \coloneqq \frac{num}{den}$
10: **end for**
11: **return** $\alpha$

---

## A.2 FIRST-ORDER ARB-RC

The first-order ARB-RC is based on the quantization error without calibration data, so in this section, $\mathcal{L}$ is defined as $\mathcal{L} = ||\mathbf{W} - \widehat{\mathbf{W}}||_F^2$. We first perform first-order binarization with the row-wise scaling factor $\alpha^r$ and column-wise scaling factor $\alpha^c$:

$$\alpha^r = \frac{1}{m} \sum_{j=1}^{m} |\mathbf{W}_{.j}|, \quad \alpha^c = \frac{1}{n} \sum_{j=1}^{n} |\frac{\mathbf{W}_{j.}}{\alpha_j^r}|, \quad \mathbf{B} = \text{sign}(\mathbf{W}). \tag{27}$$

where we discard the mean $\mu$ in this process. Then we obtain $\widehat{\mathbf{W}}$:

$$\widehat{\mathbf{W}}_{jk} = \alpha_j^r \alpha_k^c \mathbf{B}_{jk}, \quad \text{where } j = 1, 2, \ldots, n, k = 1, 2, \ldots, m. \tag{28}$$

We use the quantization error to update $\alpha^r$:

$$\mathcal{L} = ||\mathbf{W} - \widehat{\mathbf{W}}||_{\ell 2} \tag{29}$$

$$= \sum_j \sum_k (\mathbf{W}_{jk} - \alpha_j^r \alpha_k^c \mathbf{B}_{jk})^2 \tag{30}$$

$$= \sum_j \sum_k ((\mathbf{W}_{jk})^2 - 2\mathbf{W}_{jk}\alpha_k^c \mathbf{B}_{jk}\alpha_j^r + (\alpha_k^c)^2 \mathbf{B}_{jk}^2 (\alpha_j^r)^2). \tag{31}$$

To obtain the optimal solution for $\alpha^r$, we take the partial derivative of $\mathcal{L}$ with respect to $\alpha_j^r$, where $j = 1, 2, \ldots, n$:

$$\frac{\partial \mathcal{L}}{\partial \alpha_j^r} = \sum_k (-2\mathbf{W}_{jk}\mathbf{B}_{jk}\alpha_k^c + 2(\alpha_k^c)^2 \mathbf{B}_{jk}^2 \alpha_j^r). \tag{32}$$

We set $\frac{\partial \mathcal{L}}{\partial \alpha_j^r} = 0$ to get the optimal solution for $\alpha_j^r$:

$$\alpha_j^r = \frac{\sum_k \mathbf{W}_{jk} \alpha_k^c \mathbf{B}_{jk}}{\sum_k (\alpha_k^c)^2 \mathbf{B}_{jk}^2}, \quad \text{where } j = 1, 2, \ldots, n. \tag{33}$$

Then we use the same quantization error to update $\alpha_c$, we take the partial derivative of $\mathcal{L}$ with respect to $\alpha_k^c$, where $k = 1, 2, \ldots, m$:

$$\frac{\partial \mathcal{L}}{\partial \alpha_k^c} = \sum_j (-2\mathbf{W}_{jk}\mathbf{B}_{jk}\alpha_j^r + 2(\alpha_j^r)^2 \mathbf{B}_{jk}^2 \alpha_k^c). \tag{34}$$

We set $\frac{\partial \mathcal{L}}{\partial \alpha_k^c} = 0$ to get the optimal solution for $\alpha_k^c$:

$$\alpha_k^c = \frac{\sum_j \mathbf{W}_{jk} \alpha_j^r \mathbf{B}_{jk}}{\sum_j (\alpha_j^r)^2 \mathbf{B}_{jk}^2}. \tag{35}$$

We can also express $\alpha^r$ and $\alpha^c$ in a more compact vector form:

$$\alpha^r = \frac{\text{diag}(\mathbf{W}(\alpha^c \mathbf{B})^\top)}{\text{diag}((\alpha^c \mathbf{B})(\alpha^c \mathbf{B})^\top)}, \quad \alpha^c = \frac{\text{diag}(\mathbf{W}^\top (\alpha^r \mathbf{B}))}{\text{diag}((\alpha^r \mathbf{B})^\top (\alpha^r \mathbf{B}))}. \tag{36}$$

For the detailed pseudocode of first-order ARB-RC, see Algorithm 2.

---

**Algorithm 2** First-Order **A**lternating **R**efined **B**inarization along **R**ow-**C**olumn Axes

---

func ARB-RC[1]$(\mathbf{W}, \mathbf{M}, \mathbf{X}, T)$
**Input:** $\mathbf{W} \in \mathbb{R}^{n \times m}$ - full-precision weight
$\quad\quad \mathbf{M} \in \mathbb{R}^{n \times m}$ - group mask
$\quad\quad \mathbf{X} \in \mathbb{R}^{B \times L \times m}$ - calibration data
$\quad\quad T$ - iteration rounds
**Output:** $\widehat{\mathbf{W}} \in \mathbb{R}^{n \times m}$

1: $\widehat{\mathbf{W}}, \alpha^r, \alpha^c, \mathbf{B} := \text{binary\_rc}(\mathbf{W}, \mathbf{M})$
2: **for** $iter = 1, 2, \ldots, T$ **do**
3: $\quad \alpha^r \leftarrow \text{refine\_}\alpha^r(\mathbf{W}, \mathbf{B}, \alpha^c, \mathbf{M})$
4: $\quad \alpha^c \leftarrow \text{refine\_}\alpha^c(\mathbf{W}, \mathbf{B}, \alpha^r, \mathbf{M})$
5: $\quad$ **for** $k = 1, 2, \ldots, n; l = 1, 2, \ldots, m$ **do**
6: $\quad\quad \widehat{\mathbf{W}}_{kl} \leftarrow (\alpha_k^r \cdot \alpha_l^c \cdot \mathbf{B}_{kl}) \cdot \mathbf{M}_{kl}$
7: $\quad$ **end for**
8: **end for**
9: **return** $\widehat{\mathbf{W}}$
func binary\_rc $(\mathbf{W}, \mathbf{M})$
1: $\alpha^r := \frac{1}{m} \sum_{j=1}^m |(\mathbf{W} \odot \mathbf{M})_{\cdot j}|$
2: $\alpha^c := \frac{1}{n} \sum_{j=1}^n |\frac{(\mathbf{W} \odot \mathbf{M})_{j \cdot}}{\alpha_j^r}|$

3: $\mathbf{B} := \text{sign}(\mathbf{W} \odot \mathbf{M})$
4: **for** $k = 1, 2, \ldots, n; l = 1, 2, \ldots, m$ **do**
5: $\quad \widehat{\mathbf{W}}_{kl} := \alpha_k^r \alpha_l^c \mathbf{B}_{kl}$
6: **end for**
7: **return** $\widehat{\mathbf{W}}, \alpha^r, \alpha^c, \mathbf{B}$
func refine\_$\alpha^r$ $(\mathbf{W}, \mathbf{B}, \alpha^c, \mathbf{M})$
1: **for** $j = 1, 2, \ldots, n$ **do**
2: $\quad num := \sum_k \mathbf{W}_{jk} \alpha_k^c \mathbf{B}_{jk} \cdot \mathbf{M}_{jk}$
3: $\quad den := \sum_k (\alpha_k^c)^2 (\mathbf{B}_{jk} \cdot \mathbf{M}_{jk})^2 + \epsilon$
4: $\quad \alpha_j^r := \frac{num}{den}$
5: **end for**
6: **return** $\alpha^r$
func refine\_$\alpha^c$ $(\mathbf{W}, \mathbf{B}, \alpha^r, \mathbf{M})$
1: **for** $k = 1, 2, \ldots, m$ **do**
2: $\quad num := \sum_j \mathbf{W}_{jk} \alpha_j^r \mathbf{B}_{jk} \cdot \mathbf{M}_{jk}$
3: $\quad den := \sum_j (\alpha_j^r)^2 (\mathbf{B}_{jk} \cdot \mathbf{M}_{jk})^2 + \epsilon$
4: $\quad \alpha_k^c := \frac{num}{den}$
5: **end for**
6: **return** $\alpha^c$

---

# B SECOND-ORDER ARB, ARB-X, AND ARB-RC

To achieve higher quantization precision for salient weight, we apply the second-order binarization to them. To begin with, we perform a second-order binarization on the full-precision weight matrix $\mathbf{W}$:

$$\mathbf{W}_1 = \mathbf{W} - \mu_1, \quad \text{where } \mu_1 = \frac{1}{m} \sum_{j=1}^m \mathbf{W}_{\cdot j}. \tag{37}$$

The optimiztion objective for $\alpha_1$ and $\mathbf{B}_1$ is:

$$\alpha_1^*, \mathbf{B}_1^* = \underset{\alpha_1, \mathbf{B}_1}{\arg\min} ||\mathbf{W}_1 - \alpha_1 \mathbf{B}_1||_F^2. \tag{38}$$

We can obtain the optimal solution $\alpha_1^*$ and $\mathbf{B}_1^*$:

$$\alpha_1^* = \frac{1}{m} \sum_{j=1}^{m} |(\mathbf{W}_1)_{\cdot j}|, \quad \mathbf{B}_1^* = \text{sign}(\mathbf{W}_1). \tag{39}$$

Then, we define the residual matrix $\widetilde{\mathbf{W}}_1$ as $\widetilde{\mathbf{W}}_1 = \mathbf{W}_1 - \alpha_1^* \cdot \mathbf{B}_1^*$. Following the previous steps, we perform binarization on the residual matrix $\widetilde{\mathbf{W}}_1$:

$$\mathbf{W}_2 = \widetilde{\mathbf{W}}_1 - \mu_2, \quad \text{where } \mu_2 = \frac{1}{m} \sum_{j=1}^{m} (\widetilde{\mathbf{W}}_1)_{\cdot j}. \tag{40}$$

$$\alpha_2^*, \mathbf{B}_2^* = \underset{\alpha_2, \mathbf{B}_2}{\arg\min} ||\mathbf{W}_2 - \alpha_2 \mathbf{B}_2||_F^2. \tag{41}$$

$$\alpha_2^* = \frac{1}{m} \sum_{j=1}^{m} |(\widetilde{\mathbf{W}}_2)_{\cdot j}|, \quad \mathbf{B}_2^* = \text{sign}(\mathbf{W}_2). \tag{42}$$

$$\mu = \mu_1 + \mu_2. \tag{43}$$

Then we can achieve the binarized matrix $\widehat{\mathbf{W}}$:

$$\widehat{\mathbf{W}} = \alpha_1^* \cdot \mathbf{B}_1^* + \alpha_2^* \cdot \mathbf{B}_2^* + \mu. \tag{44}$$

For a more concise representation of the formula, we adopt the following formula in subsequent calculations:

$$\widehat{\mathbf{W}} = \alpha_1 \cdot \mathbf{B}_1 + \alpha_2 \cdot \mathbf{B}_2 + \mu. \tag{45}$$

## B.1 SECOND-ORDER ARB

For second-order ARB algorithm, we compute the residual matrix $\mathbf{R}$ and its row-wise mean $\delta_\mu$:

$$\mathbf{R} = \mathbf{W} - \widehat{\mathbf{W}}, \quad \delta_\mu = \frac{1}{m} \sum_{j=1}^{m} \mathbf{R}_{\cdot j}. \tag{46}$$

We first use $\delta_\mu$ to refine $\mu$:

$$\tilde{\mu} = \mu + \delta_\mu. \tag{47}$$

Then, we sequentially update $\alpha_1$ and $\alpha_2$:

$$\tilde{\alpha}_1 = \frac{\sum_{j=1}^{m} \left( (\mathbf{B}_1) \odot (\mathbf{W} - (\alpha_2 \mathbf{B}_2) - \tilde{\mu}) \right)_{\cdot j}}{\sum_{j=1}^{m} (\mathbf{B}_1)_{\cdot j}^2}, \tag{48}$$

$$\tilde{\alpha}_2 = \frac{\sum_{j=1}^{m} \left( (\mathbf{B}_2) \odot (\mathbf{W} - (\tilde{\alpha}_1 \mathbf{B}_1) - \tilde{\mu}) \right)_{\cdot j}}{\sum_{j=1}^{m} (\mathbf{B}_2)_{\cdot j}^2}. \tag{49}$$

We can further simplify it into a vectorized form:

$$\tilde{\alpha}_1 = \frac{1}{m} \text{diag}(\mathbf{B}_1^\top (\mathbf{W} - \tilde{\mu} - \alpha_2 \mathbf{B}_2)), \quad \tilde{\alpha}_2 = \frac{1}{m} \text{diag}(\mathbf{B}_2^\top (\mathbf{W} - \tilde{\mu} - \tilde{\alpha}_1 \mathbf{B}_1)). \tag{50}$$

The optimization objectives for $\mathbf{B}_1^*$ and $\mathbf{B}_2^*$ are as follows:

$$\widetilde{\mathbf{B}}_1, \widetilde{\mathbf{B}}_2 = \underset{\mathbf{B}_1, \mathbf{B}_2}{\arg\min} ||\mathbf{W} - \tilde{\mu} - \widetilde{\alpha}_1 \mathbf{B}_1 - \widetilde{\alpha}_2 \mathbf{B}_2||_{\ell_1}. \tag{51}$$

In the implementation, we utilize binary search to optimize them. Then we obtain the refined $\widehat{\mathbf{W}}$:

$$\widehat{\mathbf{W}}_{\text{refine}} = \widetilde{\alpha}_1 \cdot \widetilde{\mathbf{B}}_1 + \widetilde{\alpha}_2 \cdot \widetilde{\mathbf{B}}_2 + \tilde{\mu}. \tag{52}$$

The detailed pseudocode can be found in Algorithm 3.

---

**Algorithm 3** Second-Order **A**lternating **R**efined **B**inarization

func ARB$^2$(**W**, **M**, $T$)
**Input:** $\mathbf{W} \in \mathbb{R}^{n \times m}$ - full-precision weight
$\quad\quad\quad \mathbf{M} \in \mathbb{R}^{n \times m}$ - group mask
$\quad\quad\quad T$ - iteration rounds
**Output:** $\widehat{\mathbf{W}} \in \mathbb{R}^{n \times m}$
1: $\mathbf{W}_1, \alpha_1, \mathbf{B}_1, \mu_1 := \text{binary}(\mathbf{W}, \mathbf{M})$
2: $\mathbf{W}_2, \alpha_2, \mathbf{B}_2, \mu_2 := \text{binary}(\mathbf{W} - \mathbf{W}_1, \mathbf{M})$
3: $\mu := \mu_1 + \mu_2$
4: $\widehat{\mathbf{W}} := \mathbf{W}_1 + \mathbf{W}_2$
5: **for** iter $= 1, 2, ..., T$ **do**
6: $\quad$ $\mathbf{R} := \mathbf{W} - \widehat{\mathbf{W}}$ // residual matrix
7: $\quad$ $\delta_\mu := \sum_j (\mathbf{R} \odot \mathbf{M})_{.j}$
8: $\quad$ $\mu \leftarrow \mu + \delta_\mu$ // refine mean
9: $\quad$ $\widetilde{\mathbf{W}}_1 := \mathbf{W} - \alpha_2 \mathbf{B}_2$
10: $\quad$ $\alpha_1 \leftarrow \text{refine\_alpha}(\mathbf{B}_1, \mathbf{M}, \widetilde{\mathbf{W}}_1, \mu)$
11: $\quad$ $\widetilde{\mathbf{W}}_2 := \mathbf{W} - \alpha_1 \mathbf{B}_1$
12: $\quad$ $\alpha_2 \leftarrow \text{refine\_alpha}(\mathbf{B}_2, \mathbf{M}, \widetilde{\mathbf{W}}_2, \mu)$
13: $\quad$ $\mathbf{B}_1, \mathbf{B}_2 \leftarrow \text{refine\_B}(\mathbf{W}, \alpha_1, \alpha_2, \mu)$
14: $\quad$ $\widehat{\mathbf{W}} \leftarrow \alpha_1 \cdot \mathbf{B}_1 + \alpha_2 \cdot \mathbf{B}_2 + \mu$
15: **end for**
16: **return** $\widehat{\mathbf{W}}$

func binary (**W**, **M**)
1: $\mu := \frac{1}{m} \sum_{j=1}^m (\mathbf{W} \odot \mathbf{M})_{.j}$
2: $\widetilde{\mathbf{W}} := \mathbf{W} - \mu$
3: $\alpha := \frac{1}{m} \sum_{j=1}^m |(\widetilde{\mathbf{W}} \odot \mathbf{M})_{.j}|$
4: $\mathbf{B} := \text{sign}(\widetilde{\mathbf{W}} \odot \mathbf{M})$
5: $\widehat{\mathbf{W}} := \alpha \cdot \mathbf{B} + \mu$
6: **return** $\widehat{\mathbf{W}}, \alpha, \mathbf{B}, \mu$

func refine\_alpha (**B**, **M**, **W**, $\mu$)
1: $num := \sum_j (\mathbf{B} \odot \mathbf{M} \odot (\mathbf{W} - \mu))_{.j}$
2: $den := \sum_{j=1}^m (\mathbf{B} \odot \mathbf{M})_{.j}^2 + \epsilon$ // avoid zero-division
3: $\alpha := \frac{num}{den}$
4: **return** $\alpha$

func refine\_B (**W**, $\alpha_1, \alpha_2, \mu$)
1: $\mathbf{v} := [-\alpha_1 - \alpha_2, -\alpha_1 + \alpha_2, \alpha_1 - \alpha_2, \alpha_1 + \alpha_2]$ // ascending order
2: **for** $i = 1, 2, \ldots, n; j = 1, 2, \ldots, m$ **do**
3: $\quad$ $x := \text{BST}(\mathbf{W}_{ij} - \mu_i, \mathbf{v}_i)$
4: $\quad$ **switch** $(x)$
5: $\quad$ **case** $\mathbf{v}_1$:
6: $\quad\quad$ $(\mathbf{B}_1)_{ij} = -1, (\mathbf{B}_2)_{ij} = -1$
7: $\quad$ **case** $\mathbf{v}_2$:
8: $\quad\quad$ $(\mathbf{B}_1)_{ij} = -1, (\mathbf{B}_2)_{ij} = +1$
9: $\quad$ **case** $\mathbf{v}_3$:
10: $\quad\quad$ $(\mathbf{B}_1)_{ij} = +1, (\mathbf{B}_2)_{ij} = -1$
11: $\quad$ **default:**
12: $\quad\quad$ $(\mathbf{B}_1)_{ij} = +1, (\mathbf{B}_2)_{ij} = +1$
13: $\quad$ **end switch**
14: **end for**
15: **return** $\mathbf{B}_1, \mathbf{B}_2$

func BST $(w, \mathbf{v})$
1: $l := \text{length}(\mathbf{v})$
2: **if** $l == 1$ **then**
3: $\quad$ **return** $\mathbf{v}_1$
4: **else if** $w \geq (v_{m/2} + v_{m/2+1})/2$ **then**
5: $\quad$ **return** $\text{BST}(w, \mathbf{v}_{m/2+1:m})$
6: **else**
7: $\quad$ **return** $\text{BST}(w, \mathbf{v}_{1:m/2})$
8: **end if**

---

## B.2 SECOND-ORDER ARB-X

Based on the previously discussed second-order binarization process, we can obtain the binarized weights as $\widehat{\mathbf{W}} = \alpha_1 \mathbf{B}_1 + \mu_1 + \alpha_2 \mathbf{B}_2 + \mu_2$. The second-order ARB-X is based on the quantization error $\mathcal{L}$ with calibration data:

$$\mathcal{L} = ||\mathbf{W}\mathbf{X} - \widehat{\mathbf{W}}\mathbf{X}||_F^2. \tag{53}$$

**Rewritten weight-activation quantization error** In this section, we rewrite the quantization error to decouple $\mathbf{W}$ and $\mathbf{X}$, reducing the computational cost when calculating the quantization error. We define $\widetilde{\mathbf{W}}$ as $\widetilde{\mathbf{W}} = \mathbf{W} - (\mu_1 + \mu_2)$. Then we rewrite the quantization error:

$$\mathcal{L} = ||\mathbf{W}\mathbf{X} - \widehat{\mathbf{W}}\mathbf{X}||_F^2 \tag{54}$$

$$= ||\mathbf{X}(\widetilde{\mathbf{W}} - \alpha_1 \mathbf{B}_1 - \alpha_2 \mathbf{B}_2)^\top||_F^2 \tag{55}$$

$$= \sum_i \sum_j (\sum_b \sum_k (\mathbf{X}_b)_{ik}(\widetilde{\mathbf{W}}_{jk} - \alpha_j^{(1)}\mathbf{B}_{jk}^{(1)} - \alpha_j^{(2)}\mathbf{B}_{jk}^{(2)}))^2. \tag{56}$$

We define the residual matrix $\mathbf{R}$ as:

$$\mathbf{R}_{jk} = \widetilde{\mathbf{W}}_{jk} - \alpha_j^{(1)}\mathbf{B}_{jk}^{(1)} - \alpha_j^{(2)}\mathbf{B}_{jk}^{(2)}, \quad \text{where } j = 1, 2, \ldots, n, k = 1, 2, \ldots, m. \tag{57}$$

Then we simplify $\mathcal{L}$ with residual matrix $\mathbf{R}$:

$$\mathcal{L} = \sum_i \sum_j (\sum_b \sum_k (\mathbf{X}_b)_{ik} \mathbf{R}_{jk})^2 \tag{58}$$

$$= \sum_i \sum_j (\sum_b \sum_k \sum_l (\mathbf{X}_b)_{ik} (\mathbf{X}_b)_{il} \mathbf{R}_{jk} \mathbf{R}_{jl}) \tag{59}$$

$$= \sum_k \sum_l (\sum_b \sum_i (\mathbf{X}_b)_{ik} (\mathbf{X}_b)_{il})(\sum_j \mathbf{R}_{jk} \mathbf{R}_{jl}). \tag{60}$$

After that, we define the matrix $\mathbf{S}$:

$$\mathbf{S}_{kl} = \sum_b \sum_i (\mathbf{X}_b)_{ik} (\mathbf{X}_b)_{il}, \quad \text{where } k = 1, 2, \ldots, m, l = 1, 2, \ldots, m. \tag{61}$$

Then we obtain the final simplified $\mathcal{L}$:

$$\mathcal{L} = \langle \mathbf{S}, \mathbf{R}^\top \mathbf{R} \rangle_F = \mathrm{Tr}(\mathbf{R}\mathbf{S}\mathbf{R}^\top). \tag{62}$$

**Parameter Update** We use the quantization error $\mathcal{L}$ to update $\mu$:

$$\mathcal{L} = \sum_k \sum_l \mathbf{S}_{kl} \sum_j \mathbf{R}_{jk} \mathbf{R}_{jl} \tag{63}$$

$$= \sum_k \sum_l \mathbf{S}_{kl} \sum_j (\widetilde{\mathbf{W}}_{jk} - \alpha_j^{(1)} \mathbf{B}_{jk}^{(1)} - \alpha_j^{(2)} \mathbf{B}_{jk}^{(2)})(\widetilde{\mathbf{W}}_{jl} - \alpha_j^{(1)} \mathbf{B}_{jl}^{(1)} - \alpha_j^{(2)} \mathbf{B}_{jl}^{(2)}) \tag{64}$$

$$= \sum_k \sum_l \mathbf{S}_{kl} \sum_j (\widetilde{\mathbf{W}}_{jk} \widetilde{\mathbf{W}}_{jl} - \widetilde{\mathbf{W}}_{jk} \alpha_j^{(1)} \mathbf{B}_{jl}^{(1)} - \widetilde{\mathbf{W}}_{jk} \alpha_j^{(2)} \mathbf{B}_{jl}^{(2)} - \widetilde{\mathbf{W}}_{jl} \alpha_j^{(1)} \mathbf{B}_{jk}^{(1)} \tag{65}$$

$$- \widetilde{\mathbf{W}}_{jl} \alpha_j^{(2)} \mathbf{B}_{jk}^{(2)})$$

$$= \sum_k \sum_l \mathbf{S}_{kl} \sum_j ((\mathbf{W}_{jk} - \mu_j)(\mathbf{W}_{jl} - \mu_j) - (\mathbf{W}_{jk} - \mu_j)(\alpha_j^{(1)} \mathbf{B}_{jl}^{(1)} + \alpha_j^{(2)} \mathbf{B}_{jl}^{(2)}) \tag{66}$$

$$- (\mathbf{W}_{jl} - \mu_j)(\alpha_j^{(1)} \mathbf{B}_{jk}^{(1)} + \alpha_j^{(2)} \mathbf{B}_{jk}^{(2)})).$$

To obtain the optimal solution for $\mu$, we take the partial derivative of $\mathcal{L}$ with respect to $\mu_j$, where $j = 1, 2, \ldots, n$:

$$\frac{\partial \mathcal{L}}{\partial \mu_j} = \sum_k \sum_l \mathbf{S}_{kl}(2\mu_j - (\mathbf{W}_{jk} + \mathbf{W}_{jl}) + (\alpha_j^{(1)} \mathbf{B}_{jl}^{(1)} + \alpha_j^{(2)} \mathbf{B}_{jl}^{(2)} + \alpha_j^{(1)} \mathbf{B}_{jk}^{(1)} + \alpha_j^{(2)} \mathbf{B}_{jk}^{(2)})). \tag{67}$$

We set $\frac{\partial \mathcal{L}}{\partial \mu_j} = 0$ to obtain the optimal solution for $\mu_j$:

$$\mu_j = \frac{\sum_k \sum_l \mathbf{S}_{kl}(\mathbf{W}_{jk} - (\alpha_j^{(1)} \mathbf{B}_{jl}^{(1)} + \alpha_j^{(2)} \mathbf{B}_{jl}^{(2)}) + \mathbf{W}_{jl} - (\alpha_j^{(1)} \mathbf{B}_{jk}^{(1)} + \alpha_j^{(2)} \mathbf{B}_{jk}^{(2)}))}{2 \sum_k \sum_l \mathbf{S}_{kl}}, \tag{68}$$

where $j = 1, 2, \ldots, n$. We define the matrix $\mathbf{P}$:

$$\mathbf{P}_{kl} = \mathbf{W}_{jk} - (\alpha_j^{(1)} \mathbf{B}_{jl}^{(1)} + \alpha_j^{(2)} \mathbf{B}_{jl}^{(2)}), \quad \text{where } k = 1, 2, \ldots, m, l \text{ from 1 to m.} \tag{69}$$

We simplify $\mu_j$ using the matrix $\mathbf{P}$:

$$\mu_j = \frac{\sum_k \sum_l (\mathbf{S} \odot (\mathbf{P} + \mathbf{P}^\top))_{kl}}{2 \sum_k \sum_l \mathbf{S}_{kl}}, \quad \text{where } j = 1, 2, \ldots, n. \tag{70}$$

Since $\mathbf{S}$ is symmetric, we can further simplify the above equation as:

$$\mu_j = \frac{\sum_k \sum_l (\mathbf{S} \odot \mathbf{P})_{kl}}{\sum_k \sum_l \mathbf{S}_{kl}}, \quad \text{where } j = 1, 2, \ldots, n. \tag{71}$$

We use the quantization error to sequentially update $\alpha_1$ and $\alpha_2$:

$$\mathcal{L} = \sum_k \sum_l \mathbf{S}_{kl} \sum_j \mathbf{R}_{jk} \mathbf{R}_{jl} \tag{72}$$

$$= \sum_k \sum_l \mathbf{S}_{kl} \sum_j (\widetilde{\mathbf{W}}_{jk} - \alpha_j^{(1)} \mathbf{B}_{jk}^{(1)} - \alpha_j^{(2)} \mathbf{B}_{jk}^{(2)})(\widetilde{\mathbf{W}}_{jl} - \alpha_j^{(1)} \mathbf{B}_{jl}^{(1)} - \alpha_j^{(2)} \mathbf{B}_{jl}^{(2)}) \tag{73}$$

To update $\alpha_1$, we take the partial derivative of $\mathcal{L}$ with respect to $\alpha_j^{(1)}$, where $j = 1, 2, \ldots, n$:

$$\frac{\partial \mathcal{L}}{\partial \alpha_j^{(1)}} = \sum_k \sum_l \mathbf{S}_{kl}(2\mathbf{B}_{jk}^{(1)} \mathbf{B}_{jl}^{(1)} \alpha_j^{(1)} - \mathbf{B}_{jk}^{(1)} \widetilde{\mathbf{W}}_{jl} - \mathbf{B}_{jl}^{(1)} \widetilde{\mathbf{W}}_{jk} + \alpha_j^{(2)} \mathbf{B}_{jk}^{(1)} \mathbf{B}_{jl}^{(2)} + \alpha_j^{(2)} \mathbf{B}_{jl}^{(1)} \mathbf{B}_{jk}^{(2)}) \tag{74}$$

We set $\frac{\partial \mathcal{L}}{\partial \alpha_j^{(1)}} = 0$ to get the optimal solution for $\alpha_j^{(1)}$:

$$\alpha_j^{(1)} = \frac{\sum_k \sum_l \mathbf{S}_{kl}(\mathbf{B}_{jk}^{(1)} \widetilde{\mathbf{W}}_{jl} + \mathbf{B}_{jl}^{(1)} \widetilde{\mathbf{W}}_{jk} - \mathbf{B}_{jl}^{(1)} \alpha_j^{(2)} \mathbf{B}_{jk}^{(2)} - \mathbf{B}_{jk}^{(1)} \alpha_j^{(2)} \mathbf{B}_{jl}^{(2)})}{2 \sum_k \sum_l \mathbf{S}_{kl}(\mathbf{B}_{jk}^{(1)} \mathbf{B}_{jl}^{(1)})}, \tag{75}$$

where $j = 1, 2, \ldots, n$. We define the matrix $\mathbf{U}_1$ and $\mathbf{V}_1$ as:

$$(\mathbf{U}_1)_{kl} = \mathbf{B}_{jk}^{(1)} \widetilde{\mathbf{W}}_{jl} - \mathbf{B}_{jk}^{(1)} \alpha_j^{(2)} \mathbf{B}_{jl}^{(2)}, \quad (\mathbf{V}_1)_{kl} = \mathbf{B}_{jk}^{(1)} \mathbf{B}_{jl}^{(1)}, \tag{76}$$

where $k = 1, 2, \ldots, m$, $l = 1, 2, \ldots, m$. Then we simplify $\alpha_j^{(1)}$:

$$\alpha_j^{(1)} = \frac{\sum_k \sum_l (\mathbf{S} \odot (\mathbf{U}_1 + \mathbf{U}_1^\top))_{kl}}{2 \sum_k \sum_l (\mathbf{S} \odot \mathbf{V}_1)_{kl}}, \quad \text{where } j = 1, 2, \ldots, n. \tag{77}$$

Since $\mathbf{S}$ is symmetric, we can further simplify the above equation as

$$\alpha_j^{(1)} = \frac{\sum_k \sum_l (\mathbf{S} \odot \mathbf{U}_1)_{kl}}{\sum_k \sum_l (\mathbf{S} \odot \mathbf{V}_1)_{kl}}, \quad \text{where } j = 1, 2, \ldots, n. \tag{78}$$

Then we update $\alpha_2$, we take the partial derivative of $\mathcal{L}$ with respect to $\alpha_j^{(2)}$, where $j = 1, 2, \ldots, n$:

$$\frac{\partial \mathcal{L}}{\partial \alpha_j^{(2)}} = \sum_k \sum_l \mathbf{S}_{kl}(2\mathbf{B}_{jk}^{(2)} \mathbf{B}_{jl}^{(2)} \alpha_j^{(2)} - \mathbf{B}_{jk}^{(2)} \widetilde{\mathbf{W}}_{jl} - \mathbf{B}_{jl}^{(2)} \widetilde{\mathbf{W}}_{jk} \tag{79}$$

$$+ \alpha_j^{(1)} \mathbf{B}_{jk}^{(2)} \mathbf{B}_{jl}^{(1)} + \alpha_j^{(1)} \mathbf{B}_{jl}^{(2)} \mathbf{B}_{jk}^{(1)}).$$

We set $\frac{\partial \mathcal{L}}{\partial \alpha_j^{(2)}} = 0$ to get the optimal $\alpha_j^{(2)}$:

$$\alpha_j^{(2)} = \frac{\sum_k \sum_l \mathbf{S}_{kl}(\mathbf{B}_{jk}^{(2)} \widetilde{\mathbf{W}}_{jl} + \mathbf{B}_{jl}^{(2)} \widetilde{\mathbf{W}}_{jk} - \mathbf{B}_{jl}^{(2)} \alpha_j^{(1)} \mathbf{B}_{jk}^{(1)} - \mathbf{B}_{jk}^{(2)} \alpha_j^{(1)} \mathbf{B}_{jl}^{(1)})}{2 \sum_k \sum_l \mathbf{S}_{kl}(\mathbf{B}_{jk}^{(2)} \mathbf{B}_{jl}^{(2)})}, \tag{80}$$

where $j = 1, 2, \ldots, n$. We define the matrix $\mathbf{U}_2$ and $\mathbf{V}_2$ as:

$$(\mathbf{U}_2)_{kl} = \mathbf{B}_{jk}^{(2)} \widetilde{\mathbf{W}}_{jl} - \mathbf{B}_{jk}^{(2)} \alpha_j^{(1)} \mathbf{B}_{jl}^{(1)}, \quad (\mathbf{V}_2)_{kl} = \mathbf{B}_{jk}^{(2)} \mathbf{B}_{jl}^{(2)}, \tag{81}$$

where $k = 1, 2, \ldots, m$, $l = 1, 2, \ldots, m$. We can simplify $\alpha_j^{(2)}$ using $\mathbf{U}_2$ and $\mathbf{V}_2$:

$$\alpha_j^{(2)} = \frac{\sum_k \sum_l (\mathbf{S} \odot (\mathbf{U}_2 + \mathbf{U}_2^\top))_{kl}}{2 \sum_k \sum_l (\mathbf{S} \odot \mathbf{V}_2)_{kl}}, \quad \text{where } j = 1, 2, \ldots, n. \tag{82}$$

Since $\mathbf{S}$ is symmetric, we can further simplify the above equation as:

$$\alpha_j^{(2)} = \frac{\sum_k \sum_l (\mathbf{S} \odot \mathbf{U}_2)_{kl}}{\sum_k \sum_l (\mathbf{S} \odot \mathbf{V}_2)_{kl}}, \quad \text{where } j = 1, 2, \ldots, n. \tag{83}$$

The detailed pseudocode can be found in Algorithm 4.

---

**Algorithm 4** Second-Order **A**lternating **R**efined **B**inarization with calibration data

---

func ARB-X$^2$(**W**, **M**, **X**, $T$)
**Input:** $\mathbf{W} \in \mathbb{R}^{n \times m}$ - full-precision weight
       $\mathbf{M} \in \mathbb{R}^{n \times m}$ - group mask
       $\mathbf{X} \in \mathbb{R}^{B \times L \times m}$ - calibration data
       $T$ - iteration rounds
**Output:** $\widehat{\mathbf{W}} \in \mathbb{R}^{n \times m}$
 1: $\mathbf{S} := \text{X2S}(\mathbf{X})$ // $\mathbf{S} \in \mathbb{R}^{m \times m}$
 2: $\widehat{\mathbf{W}}_1, \alpha_1, \mathbf{B}_1, \mu_1 := \text{binary}(\mathbf{W}, \mathbf{M})$
 3: $\widehat{\mathbf{W}}_2, \alpha_2, \mathbf{B}_2, \mu_2 := \text{binary}(\mathbf{W} - \widehat{\mathbf{W}}_1, \mathbf{M})$
 4: $\mu := \mu_1 + \mu_2$
 5: $\widehat{\mathbf{W}} := \widehat{\mathbf{W}}_1 + \widehat{\mathbf{W}}_2$
 6: $\mathbf{B}_1 \leftarrow \mathbf{B}_1 \odot \mathbf{M}$
 7: $\mathbf{B}_2 \leftarrow \mathbf{B}_2 \odot \mathbf{M}$
 8: **for** $iter = 1, 2, \ldots, T$ **do**
 9:    $\mu \leftarrow \text{refine\_}\mu(\mathbf{S}, \mathbf{W}, \mathbf{B}_1, \mathbf{B}_2, \alpha_1, \alpha_2)$
10:   $\alpha_1 \leftarrow \text{refine\_}\alpha(\mathbf{S}, \mathbf{W}, \mu, \mathbf{B}_1, \mathbf{B}_2, \alpha_2)$
11:   $\alpha_2 \leftarrow \text{refine\_}\alpha(\mathbf{S}, \mathbf{W}, \mu, \mathbf{B}_2, \mathbf{B}_1, \alpha_1)$
12:   $\widehat{\mathbf{W}} \leftarrow (\alpha \cdot \mathbf{B} + \mu) \odot \mathbf{M}$
13: **end for**
14: **return** $\widehat{\mathbf{W}}$

func binary (**W**, **M**)
 1: $\mu := \frac{1}{m} \sum_{j=1}^{m} (\mathbf{W} \odot \mathbf{M})_{.j}$
 2: $\widetilde{\mathbf{W}} := \mathbf{W} - \mu$
 3: $\alpha := \frac{1}{m} \sum_{j=1}^{m} |(\widetilde{\mathbf{W}} \odot \mathbf{M})_{.j}|$
 4: $\mathbf{B} := \text{sign}(\widetilde{\mathbf{W}} \odot \mathbf{M})$
 5: $\widehat{\mathbf{W}} := \alpha \cdot \mathbf{B} + \mu$
 6: **return** $\widehat{\mathbf{W}}, \alpha, \mathbf{B}, \mu$

func X2S (**X**)
 1: **for** $b = 1, 2, \ldots B$ **do**
 2:    **for** $k = 1, 2, \ldots, m; l = 1, 2, \ldots, m$ **do**
 3:      $\mathbf{S}_{kl} = \sum_b \sum_i (\mathbf{X}_b)_{ik} (\mathbf{X}_b)_{il}$
 4:    **end for**
 5: **end for**
 6: **return** $\mathbf{S}$

func refine\_$\mu$(**S**, **W**, $\mathbf{B}_1$, $\mathbf{B}_2$, $\alpha_1$, $\alpha_2$)
 1: **for** $i = 1, 2, \ldots, n$ **do**
 2:    **for** $k = 1, 2, \ldots, m; l = 1, 2, \ldots, m$ **do**
 3:      $\mathbf{P}_{kl} = \mathbf{W}_{ik} - (\alpha_i^{(1)} \mathbf{B}_{il}^{(1)} + \alpha_i^{(2)} \mathbf{B}_{il}^{(2)})$
 4:    **end for**
 5:    $num := \sum_k \sum_l (\mathbf{S} \odot \mathbf{P})_{kl}$
 6:    $den := \sum_k \sum_l \mathbf{S}_{kl} + \epsilon$
 7:    $\mu_i := \frac{num}{den}$
 8: **end for**
 9: **return** $\mu$

func refine\_$\alpha$ (**S**, **W**, $\mu$, **B**, $\widetilde{\mathbf{B}}$, $\widetilde{\alpha}$)
 1: $\widetilde{\mathbf{W}} := \mathbf{W} - \mu$
 2: **for** $i = 1, 2, \ldots, n$ **do**
 3:    **for** $k = 1, 2, \ldots, m; l = 1, 2, \ldots, m$ **do**
 4:      $\mathbf{U}_{kl} := \mathbf{B}_{ik} \widetilde{\mathbf{W}}_{il} - \mathbf{B}_{ik} \widetilde{\alpha}_i \widetilde{\mathbf{B}}_{il}$
 5:      $\mathbf{V}_{kl} := \mathbf{B}_{ik} \mathbf{B}_{il}$
 6:    **end for**
 7:    $num := \sum_k \sum_l (\mathbf{S} \odot \mathbf{U})_{kl}$
 8:    $den := \sum_k \sum_l (\mathbf{S} \odot \mathbf{V})_{kl} + \epsilon$
 9:    $\alpha_i := \frac{num}{den}$
10: **end for**
11: **return** $\alpha$

---

### B.3 Second-order ARB-RC

We perform second-order binarization under the ARB-RC method:

$$\alpha_1^r = \frac{1}{m} \sum_{j=1}^{m} |\mathbf{W}_{.j}|, \quad \alpha_1^c = \frac{1}{n} \sum_{j=1}^{n} |\frac{\mathbf{W}_{j.}}{(\alpha_1^r)_j}|, \quad \mathbf{B}_1 = \text{sign}(\mathbf{W}). \tag{84}$$

We define the residual matrix **R** as:

$$\mathbf{R}_{jk} = \mathbf{W}_{jk} - \alpha_j^r \alpha_k^c \mathbf{B}_{jk}, \quad \text{where } j = 1, 2, \ldots, n, k = 1, 2, \ldots, m. \tag{85}$$

Then we perform binarization on residual matrix **R**:

$$\alpha_2^r = \frac{1}{m} \sum_{j=1}^{m} |\mathbf{R}_{.j}|, \quad \alpha_2^c = \frac{1}{n} \sum_{j=1}^{n} |\frac{\mathbf{R}_{j.}}{(\alpha_2^r)_j}|, \quad \mathbf{B}_2 = \text{sign}(\mathbf{R}). \tag{86}$$

We can obtain the results of the second-order binarization:

$$\widehat{\mathbf{W}}_{jk} = (\alpha_1^r)_j (\alpha_1^c)_k (\mathbf{B}_1)_{jk} + (\alpha_2^r)_j (\alpha_2^c)_k (\mathbf{B}_2)_{jk}, \quad \text{where } j = 1, 2, \ldots, n, k = 1, 2, \ldots, m. \tag{87}$$

Then quantization error without calibration data is given by the following formula:

$$\mathcal{L} = ||\mathbf{W} - \widehat{\mathbf{W}}||_{\ell 2} \tag{88}$$

$$= \sum_j \sum_k (\mathbf{W}_{jk} - ((\alpha_1^r)_j (\alpha_1^c)_k (\mathbf{B}_1)_{jk} + (\alpha_2^r)_j (\alpha_2^c)_k (\mathbf{B}_2)_{jk}))^2. \tag{89}$$

We define $\widetilde{\mathbf{W}}_1$ and $\widetilde{\mathbf{W}}_2$ as:

$$(\widetilde{\mathbf{W}}_1)_{jk} = \mathbf{W}_{jk} - (\alpha_2^r)_j(\alpha_2^c)_k(\mathbf{B}_2)_{jk}, \quad (\widetilde{\mathbf{W}}_2)_{jk} = \mathbf{W}_{jk} - (\alpha_1^r)_j(\alpha_1^c)_k(\mathbf{B}_1)_{jk}. \quad (90)$$

where $j = 1, 2, \ldots, n$, $k = 1, 2, \ldots, m$. Then we update $\alpha_1^r$ using the following formulas:

$$\frac{\partial \mathcal{L}}{\partial (\alpha_1^r)_j} = \sum_k (-2(\widetilde{\mathbf{W}}_1)_{jk}(\mathbf{B}_1)_{jk}(\alpha_1^c)_k + 2(\alpha_1^c)_k^2(\mathbf{B}_1)_{jk}^2(\alpha_1^r)_j). \quad (91)$$

We set $\frac{\partial \mathcal{L}}{\partial (\alpha_1^r)_j} = 0$ to get the optimal solution for $(\alpha_1^r)_j$:

$$(\alpha_1^r)_j = \frac{\sum_k (\widetilde{\mathbf{W}}_1)_{jk}(\alpha_1^c)_k(\mathbf{B}_1)_{jk}}{\sum_k (\alpha_1^c)_k^2(\mathbf{B}_1)_{jk}^2}, \quad \text{where } j = 1, 2, \ldots, n. \quad (92)$$

We update $\alpha_1^c$ using the following formulas:

$$\frac{\partial \mathcal{L}}{\partial (\alpha_1^c)_k} = \sum_j (-2(\widetilde{\mathbf{W}}_1)_{jk}(\mathbf{B}_1)_{jk}(\alpha_1^r)_j + 2(\alpha_1^r)_j^2(\mathbf{B}_1)_{jk}^2(\alpha_1^c)_k). \quad (93)$$

We set $\frac{\partial \mathcal{L}}{\partial (\alpha_1^c)_k} = 0$ to get the optimal solution for $(\alpha_1^c)_k$:

$$(\alpha_1^c)_k = \frac{\sum_j (\widetilde{\mathbf{W}}_1)_{jk}(\alpha_1^r)_j(\mathbf{B}_1)_{jk}}{\sum_j (\alpha_1^r)_j^2(\mathbf{B}_1)_{jk}^2}, \quad \text{where } k = 1, 2, \ldots, m. \quad (94)$$

We update $\alpha_2^r$ using the following formulas:

$$\frac{\partial \mathcal{L}}{\partial (\alpha_2^r)_j} = \sum_k (-2(\widetilde{\mathbf{W}}_2)_{jk}(\mathbf{B}_2)_{jk}(\alpha_2^c)_k + 2(\alpha_2^c)_k^2(\mathbf{B}_2)_{jk}^2(\alpha_2^r)_j). \quad (95)$$

We set $\frac{\partial \mathcal{L}}{\partial (\alpha_2^r)_j} = 0$ to get the optimal solution for $(\alpha_2^r)_j$:

$$(\alpha_2^r)_j = \frac{\sum_k (\widetilde{\mathbf{W}}_2)_{jk}(\alpha_2^c)_k(\mathbf{B}_2)_{jk}}{\sum_k (\alpha_2^c)_k^2(\mathbf{B}_2)_{jk}^2}, \quad \text{where } j = 1, 2, \ldots, n. \quad (96)$$

We update $\alpha_2^c$ using the following formulas:

$$\frac{\partial \mathcal{L}}{\partial (\alpha_2^c)_k} = \sum_j (-2(\widetilde{\mathbf{W}}_2)_{jk}(\mathbf{B}_2)_{jk}(\alpha_2^r)_j + 2(\alpha_2^r)_j^2(\mathbf{B}_2)_{jk}^2(\alpha_2^c)_k). \quad (97)$$

We set $\frac{\partial \mathcal{L}}{\partial (\alpha_2^c)_k} = 0$ to get the optimal solution for $(\alpha_2^c)_k$:

$$(\alpha_2^c)_k = \frac{\sum_j (\widetilde{\mathbf{W}}_2)_{jk}(\alpha_2^r)_j(\mathbf{B}_2)_{jk}}{\sum_j (\alpha_2^r)_j^2(\mathbf{B}_2)_{jk}^2}, \quad \text{where } k = 1, 2, \ldots, m. \quad (98)$$

The objective of optimizing $\mathbf{B}_1$ and $\mathbf{B}_2$ is:

$$(\mathbf{B}_1)_{jk}, (\mathbf{B}_2)_{jk} = \underset{(\mathbf{B}_1)_{jk}, (\mathbf{B}_2)_{jk}}{\arg\min} ||\mathbf{W}_{jk} - ((\alpha_1^r)_j(\alpha_1^c)_k(\mathbf{B}_1)_{jk} + (\alpha_2^r)_j(\alpha_2^c)_k(\mathbf{B}_2)_{jk})||_{\ell_1}, \quad (99)$$

where $j = 1, 2, \ldots, n$, $k = 1, 2, \ldots, m$. The detailed pseudocode for second-order ARB-RC can be found in Algorithm 5.

---

**Algorithm 5** Second-Order **A**lternating **R**efined **B**inarization along **R**ow-**C**olumn Axes

func ARB-RC$^2$(**W**, **M**, **X** ,$T$)
**Input:** $\mathbf{W} \in \mathbb{R}^{n \times m}$ - full-precision weight
      $\mathbf{M} \in \mathbb{R}^{n \times m}$ - group mask
      $\mathbf{X} \in \mathbb{R}^{B \times L \times m}$ - calibration data
      $T$ - iteration rounds
**Output:** $\widehat{\mathbf{W}} \in \mathbb{R}^{n \times m}$

1: $\widehat{\mathbf{W}}_1, \alpha_1^r, \alpha_1^c, \mathbf{B}_1 \coloneqq \text{binary\_rc}(\mathbf{W}, \mathbf{M})$
2: $\widehat{\mathbf{W}}_2, \alpha_2^r, \alpha_2^c, \mathbf{B}_2 \coloneqq \text{binary\_rc}(\mathbf{W} - \widehat{\mathbf{W}}, \mathbf{M})$

3: $\widehat{\mathbf{W}} = \widehat{\mathbf{W}}_1 + \widehat{\mathbf{W}}_2$
4: $\mathbf{B}_1 \leftarrow \mathbf{B}_1 \odot \mathbf{M}$
5: $\mathbf{B}_2 \leftarrow \mathbf{B}_2 \odot \mathbf{M}$
6: **for** $iter = 1, 2, \ldots, T$ **do**
7:    **for** $j = 1, 2, \ldots, n; k = 1, 2, \ldots, m$ **do**
8:       $(\widetilde{\mathbf{W}}_1)_{jk} = \mathbf{W}_{jk} - (\alpha_2^r)_j (\alpha_2^c)_k (\mathbf{B}_2)_{jk}$

9:       $(\widetilde{\mathbf{W}}_2)_{jk} = \mathbf{W}_{jk} - (\alpha_1^r)_j (\alpha_1^c)_k (\mathbf{B}_1)_{jk}$
10:    **end for**
11:    $\alpha_1^r \leftarrow \text{refine\_}\alpha^r(\widetilde{\mathbf{W}}_1, \mathbf{B}_1, \alpha_1^c)$
12:    $\alpha_1^c \leftarrow \text{refine\_}\alpha^c(\widetilde{\mathbf{W}}_1, \mathbf{B}_1, \alpha_1^r)$
13:    $\alpha_2^r \leftarrow \text{refine\_}\alpha^r(\widetilde{\mathbf{W}}_2, \mathbf{B}_2, \alpha_2^c)$
14:    $\alpha_2^c \leftarrow \text{refine\_}\alpha^c(\widetilde{\mathbf{W}}_2, \mathbf{B}_2, \alpha_1^r)$
15:    $\mathbf{B}_1, \mathbf{B}_2 \leftarrow \text{refine\_B}(\mathbf{W}, \alpha_1^r, \alpha_2^r, \alpha_1^c, \alpha_2^c)$
16:    **for** $k = 1, 2, \ldots, n; l = 1, 2, \ldots, m$ **do**
17:       $\widehat{\mathbf{W}}_{kl} \leftarrow ((\alpha_1^r)_k (\alpha_1^c)_l (\mathbf{B}_1)_{kl} + (\alpha_2^r)_k (\alpha_2^c)_l (\mathbf{B}_2)_{kl}) \cdot \mathbf{M}_{kl}$
18:    **end for**
19: **end for**
20: **return** $\widehat{\mathbf{W}}$

func binary\_rc (**W**, **M**)
1: $\alpha^r \coloneqq \frac{1}{m} \sum_{j=1}^{m} |(\mathbf{W} \odot \mathbf{M})_{\cdot j}|$
2: $\alpha^c \coloneqq \frac{1}{n} \sum_{j=1}^{n} |\frac{(\mathbf{W} \odot \mathbf{M})_{j \cdot}}{\alpha_j^r}|$
3: $\mathbf{B} \coloneqq \text{sign}(\mathbf{W} \odot \mathbf{M})$
4: **for** $k = 1, 2, \ldots, n; l = 1, 2, \ldots, m$ **do**
5:    $\widehat{\mathbf{W}}_{kl} \coloneqq \alpha_k^r \alpha_l^c \mathbf{B}_{kl}$
6: **end for**
7: **return** $\widehat{\mathbf{W}}, \alpha^r, \alpha^c, \mathbf{B}$

func refine\_$\alpha^r$ (**W**, **B**, $\alpha^c$)
1: **for** $j = 1, 2, \ldots, n$ **do**

2:    $num \coloneqq \sum_k \mathbf{W}_{jk} \alpha_k^c \mathbf{B}_{jk}$
3:    $den \coloneqq \sum_k (\alpha_k^c)^2 \mathbf{B}_{jk}^2 + \epsilon$
4:    $\alpha_j^r \coloneqq \frac{num}{den}$
5: **end for**
6: **return** $\alpha^r$

func refine\_$\alpha^c$ (**W**, **B**, $\alpha^r$)
1: **for** $k = 1, 2, \ldots, m$ **do**
2:    $num \coloneqq \sum_j \mathbf{W}_{jk} \alpha_j^r \mathbf{B}_{jk}$
3:    $den \coloneqq \sum_j (\alpha_j^r)^2 \mathbf{B}_{jk}^2 + \epsilon$
4:    $\alpha_k^c \coloneqq \frac{num}{den}$
5: **end for**
6: **return** $\alpha^c$

func refine\_B (**W**, $\alpha_1^r, \alpha_2^r, \alpha_1^c, \alpha_2^c$)
1: **for** $i = 1, 2, \ldots, n; j = 1, 2, \ldots, m$ **do**
2:    $\mathbf{v} \coloneqq \begin{bmatrix} -(\alpha_1^r)_i (\alpha_1^c)_j - (\alpha_2^r)_i (\alpha_2^c)_j \\ -(\alpha_1^r)_i (\alpha_1^c)_j + (\alpha_2^r)_i (\alpha_2^c)_j \\ (\alpha_1^r)_i (\alpha_1^c)_j - (\alpha_2^r)_i (\alpha_2^c)_j \\ (\alpha_1^r)_i (\alpha_1^c)_j + (\alpha_2^r)_i (\alpha_2^c)_j \end{bmatrix}$
3:    $x \coloneqq \text{BST}(\mathbf{W}_{ij}, \mathbf{v})$
4:    **switch** $(x)$
5:    **case** $\mathbf{v}_1$:
6:       $(\mathbf{B}_1)_{ij} = -1, (\mathbf{B}_2)_{ij} = -1$
7:    **case** $\mathbf{v}_2$:
8:       $(\mathbf{B}_1)_{ij} = -1, (\mathbf{B}_2)_{ij} = +1$
9:    **case** $\mathbf{v}_3$:
10:      $(\mathbf{B}_1)_{ij} = +1, (\mathbf{B}_2)_{ij} = -1$
11:    **default**:
12:      $(\mathbf{B}_1)_{ij} = +1, (\mathbf{B}_2)_{ij} = +1$
13:    **end switch**
14: **end for**
15: **return** $\mathbf{B}_1, \mathbf{B}_2$

func BST $(w, \mathbf{v})$
1: $l \coloneqq \text{length}(\mathbf{v})$
2: **if** $l == 1$ **then**
3:    **return** $\mathbf{v}_1$
4: **else if** $w \geq (v_{m/2} + v_{m/2+1})/2$ **then**
5:    **return** $\text{BST}(w, \mathbf{v}_{m/2+1:m})$
6: **else**
7:    **return** $\text{BST}(w, \mathbf{v}_{1:m/2})$
8: **end if**

---

## C  PROOF OF THEOREM 1

Here we consider a row of the weight matrix $\mathbf{W} \in \mathbb{R}^{1 \times m}$ with single $\mu$ and $\alpha$. In the $(\tau+1)$-th iteration, we update $\mu$ by using the following formulas:

$$\mathbf{R}^\tau = \mathbf{W} - \alpha^\tau \mathbf{B}^\tau - \mu^\tau, \tag{100}$$

$$\delta_\mu^\tau = \overline{\mathbf{R}^\tau}, \tag{101}$$

$$\mu^{\tau+1} = \mu^\tau + \delta_\mu^\tau. \tag{102}$$

Then we can compute the specific value of the quantization error reduction after updating $\mu^\tau$:

$$||\mathbf{R}^\tau - \delta_\mu^\tau||^2 = ||\mathbf{R}^\tau||^2 - 2\langle \mathbf{R}^\tau, \delta_\mu^\tau \rangle + ||\delta_\mu^\tau||^2 \tag{103}$$

$$= ||\mathbf{R}^\tau||^2 - 2\delta_\mu^\tau \sum_{k=1}^m \mathbf{R}_k^\tau + m(\delta_\mu^\tau)^2 \tag{104}$$

$$= ||\mathbf{R}^\tau||^2 - m(\delta_\mu^\tau)^2 \tag{105}$$

$$\leq ||\mathbf{R}^\tau||^2. \tag{106}$$

Here we have $m\delta_\mu^{\tau 2} = ||\mathbf{R}^\tau||^2 - n\sigma^2$. We refine $\mathbf{B}$ using the following formula:

$$\mathbf{B}^{\tau+1} = \text{sign}(\mathbf{W} - \mu^\tau - \delta_\mu^\tau). \tag{107}$$

The following derivation provides the decrease in quantization error after updating $\mathbf{B}$:

$$||\mathbf{R}^\tau - \delta_\mu^\tau + \alpha^\tau(\mathbf{B}^\tau - \mathbf{B}^{\tau+1})||^2 \tag{108}$$

$$= ||\mathbf{R}^\tau - \delta_\mu^\tau||^2 + 2\langle(\mathbf{R}^\tau - \delta_\mu^\tau), \alpha^\tau(\mathbf{B}^\tau - \mathbf{B}^{\tau+1})\rangle + ||\alpha^\tau(\mathbf{B}^\tau - \mathbf{B}^{\tau+1})||^2 \tag{109}$$

$$= ||\mathbf{R}^\tau - \delta_\mu^\tau||^2 + \alpha^\tau \sum_k (\mathbf{B}_k^\tau - \mathbf{B}_k^{\tau+1})(2\mathbf{R}_k^\tau - 2\delta_\mu^\tau + \alpha^\tau \mathbf{B}_k^\tau - \alpha^\tau \mathbf{B}_k^{\tau+1}) \tag{110}$$

$$= ||\mathbf{R}^\tau||^2 - m\delta_\mu^{\tau 2} + \alpha^\tau \sum_k (\mathbf{B}_k^\tau - \mathbf{B}_k^{\tau+1})(2(\mathbf{W}_k - \mu^\tau - \delta_\mu^\tau) - \alpha^\tau(\mathbf{B}_k^\tau + \mathbf{B}_k^{\tau+1})) \tag{111}$$

$$= ||\mathbf{R}^\tau||^2 - m\delta_\mu^{\tau 2} + M, \tag{112}$$

$$\text{where } M = \alpha^\tau \sum_k (\mathbf{B}_k^\tau - \mathbf{B}_k^{\tau+1})(2(\mathbf{W}_k - \mu^\tau - \delta_\mu^\tau) - \alpha^\tau(\mathbf{B}_k^\tau + \mathbf{B}_k^{\tau+1})). \tag{113}$$

M only has the following conditions:
if $\mathbf{B}_k^\tau = \mathbf{B}_k^{\tau+1} \Rightarrow M = 0$,
if $\mathbf{B}_k^\tau = +1, \mathbf{B}_k^{\tau+1} = -1 \Rightarrow \mathbf{W} - \mu^\tau - \delta_\mu^\tau < 0 \Rightarrow M < 0$,
if $\mathbf{B}_k^\tau = -1, \mathbf{B}_k^{\tau+1} = +1 \Rightarrow \mathbf{W} - \mu^\tau - \delta_\mu^\tau > 0 \Rightarrow M < 0$,
$\Rightarrow M \leq 0$,
$\Rightarrow ||\mathbf{R}^\tau - \delta_\mu^\tau + \alpha^\tau(\mathbf{B}^\tau - \mathbf{B}^{\tau+1})||^2 \leq ||\mathbf{R}^\tau||^2 - m\delta_\mu^{\tau 2}$.
Specifically,

$$M = 2m(\alpha^\tau)^2 - 2\alpha^\tau \delta_\mu^\tau \sum_k \mathbf{B}_k^\tau - 2m\alpha^\tau \alpha^{\tau+1} \tag{114}$$

$$= 2m(\alpha^\tau)^2 + 2m\delta_\mu^\tau(\mu^{\tau+1} - \overline{\mathbf{W}}) - 2m\alpha^\tau \alpha^{\tau+1}. \tag{115}$$

Then we refine $\alpha$ and calculate the reduction of quantization error:

$$\mathbf{R}' = \mathbf{R}^\tau - \delta_\mu^\tau + \alpha^\tau(\mathbf{B}^\tau - \mathbf{B}^{\tau+1}), \tag{116}$$

$$||\mathbf{R}' + (\alpha^\tau - \alpha^{\tau+1})\mathbf{B}^{\tau+1}||^2 \tag{117}$$

$$= ||\mathbf{R}'||^2 + (\alpha^\tau - \alpha^{\tau+1}) \sum_k \mathbf{B}_k^{\tau+1}(2\mathbf{R}_k' + (\alpha^\tau - \alpha^{\tau+1})\mathbf{B}^{\tau+1}) \tag{118}$$

$$= ||\mathbf{R}'||^2 + (\alpha^\tau - \alpha^{\tau+1}) \sum_k \mathbf{B}_k^{\tau+1}(2(\mathbf{W}_k - \mu^{\tau+1}) - (\alpha^\tau + \alpha^{\tau+1})\mathbf{B}^{\tau+1}) \tag{119}$$

$$= ||\mathbf{R}'||^2 + 2m\alpha^{\tau+1}(\alpha^\tau - \alpha^{\tau+1}) - m(\alpha^\tau)^2 + m(\alpha^{\tau+1})^2 \tag{120}$$

$$= ||\mathbf{R}'||^2 - m(\alpha^\tau - \alpha^{\tau+1})^2 \tag{121}$$

$$\leq ||\mathbf{R}'||^2. \tag{122}$$

Combining the results from the above derivation, we can obtain the specific value of the decrease of quantization error after the $(\tau+1)$-th iteration:

$$||\mathbf{R}^{\tau+1}||^2 - ||\mathbf{R}^\tau||^2 \tag{123}$$

$$= -m(\delta_\mu^\tau)^2 + 2m(\alpha^\tau)^2 + 2m\delta_\mu^\tau(\mu^{\tau+1} - \mu^0) - 2m\alpha^\tau \alpha^{\tau+1} - m(\alpha^\tau - \alpha^{\tau+1})^2 \tag{124}$$

$$= m((\alpha^\tau)^2 - (\alpha^{\tau+1})^2 - (\delta_\mu^\tau)^2 + 2\delta_\mu^\tau(\mu^{\tau+1} - \mu^0)) \tag{125}$$

We can also derive the relationship between the quantization error after the $T$-th iteration $\mathbf{R}^T$ and the quantization error before iteration $\mathbf{R}^0$ as

$$||\mathbf{R}^T||^2 - ||\mathbf{R}^0||^2 = m((\alpha^0)^2 - (\alpha^T)^2 + (\mu^T - \mu^0)^2). \tag{126}$$

## D  PROOF OF THEOREM 2

$\mathbf{X} \in \mathbb{R}^{B \times L \times m}$ is the calibration data. $\mathbf{W}, \widehat{\mathbf{W}} \in \mathbb{R}^{n \times m}$ are the full-precision weight matrix and binarized weight matrix, $k$ is the block size, $T$ is theiteration rounds. The quantization error with calibration data $\mathcal{L}_1$ is shown as follows:

$$\mathcal{L}_1 = ||\mathbf{W}\mathbf{X} - \widehat{\mathbf{W}}\mathbf{X}||^2 \tag{127}$$

$$= ||\mathbf{X} \cdot (\mathbf{W} - \widehat{\mathbf{W}})^\top||^2. \tag{128}$$

We calculate the time complexity of $\mathbf{X} \cdot (\mathbf{W} - \widehat{\mathbf{W}})^\top$ as $\mathcal{O}(B \cdot k \cdot L \cdot n)$ since it involves $B$ matrix multiplications, each requiring $(L \cdot n \cdot k)$ multiplications and $(L \cdot n \cdot (k-1))$ additions. The time complexity of squaring all the elements of a matrix and then sums them up is $\mathcal{O}(L \cdot n)$. In each iteration, we need to perform $\frac{m}{k}$ calculations, and there are $T$ iterations in total. Therefore, the overall time complexity is $T \cdot \frac{m}{k} \cdot (\mathcal{O}(B \cdot k \cdot L \cdot n) + \mathcal{O}(L \cdot n))$, which simplifies to $\mathcal{O}(n \cdot B \cdot L \cdot m \cdot T)$.

$$\mathcal{L}_2 = \sum_i \sum_j (\mathbf{S} \odot \mathbf{R})_{ij}. \tag{129}$$

We calculate the time complexity of $\mathbf{S}$ as $\mathcal{O}(B \cdot k^2 \cdot L)$, since $\mathbf{S}$ contains $k^2$ elements, and calculating each element involves $(B \cdot L)$ multiplications and $(B \cdot L)$ additions. And we calculate the time complexity of $\mathbf{R}$ as $\mathcal{O}(n \cdot k^2)$, since $\mathbf{R}$ contains $k^2$ elements, and calculating each element involves $n$ multiplications and $(n-1)$ additions. The time complexity of the element-wise multiplication of $\mathbf{S}$ and $\mathbf{R}$, as well as the summation of the matrices, is $\mathcal{O}(k^2)$. In each iteration, we need to perform $\frac{m}{k}$ calculations, and there are $T$ iterations in total. It is important to note that the calculation of matrix $\mathbf{S}$ does not need to be performed in every iteration of the $T$ iterations; it only needs to be computed once. Therefore, the overall time complexity is $\frac{m}{k} \cdot (\mathcal{O}(B \cdot k^2 \cdot L) + T \cdot (\mathcal{O}(n \cdot k^2) + \mathcal{O}(k^2)))$, which simplifies to $\mathcal{O}(m \cdot k \cdot (B \cdot L + n \cdot T))$.

We define the acceleration ratio $\eta$ as the quotient of the time complexities of $\mathcal{L}_1$ and $\mathcal{L}_2$

$$\eta = \frac{\mathcal{O}(n \cdot B \cdot L \cdot m \cdot T)}{\mathcal{O}(m \cdot k \cdot (B \cdot L + n \cdot T))} \tag{130}$$

$$\propto \frac{1}{k \cdot (\frac{1}{n \cdot T} + \frac{1}{B \cdot L})}. \tag{131}$$

Typically, we set $n$ to 4096, $B$ to 128, $L$ to 2048, $T$ to 15, and $k$ to 128. Under these circumstances, $\eta$ is approximately equal to 389.

## E  MEMORY COMPUTATION

For $\mathbf{W} \in \mathbb{R}^{n \times m}$, block size $k$, the memory of $\hat{\mathbf{W}}$ after standard row-wise binarization is

$$\mathcal{M}^{1st} = \overbrace{n \times m}^{\mathbf{B}} + \overbrace{\lceil m/k \rceil}^{multiple\ blocks} \times \overbrace{2 \times n \times 16}^{row-wise\ \text{FP16}\ \alpha\ and\ \mu}. \tag{132}$$

Moreover, second-order row-wise binarization can be represented as

$$\mathcal{M}^{2nd} = \overbrace{2 \times n \times m}^{\mathbf{B_1}\ and\ \mathbf{B_2}} + \overbrace{\lceil m/k \rceil}^{multiple\ blocks} \times \overbrace{3 \times n \times 16}^{row-wise\ \text{FP16}\ \alpha_1,\ \alpha_2,\ and\ \mu}, \tag{133}$$

since row-wise $\mu_1$ and $\mu_2$ can be combined together as $\mu = \mu_1 + \mu_2$.

Thus, the memory required by BiLLM can be formulated as

$$\mathcal{M}_{\text{BiLLM}} = \overbrace{2 \times n \times c + \lceil m/k \rceil \times 3n \times 16}^{second-order\ binarization} + \overbrace{n \times (m - c) + \underbrace{\lceil m/k \rceil \times 2n \times 16 \times 2}_{2\ groups}}^{first-order\ binarization} \tag{134}$$

$$+ \overbrace{n \times m}^{group\ bitmap} + \overbrace{m}^{salient\ column\ bitmap}, \tag{135}$$

where $c$ is the number of salient columns for $\mathbf{W}$.

Similarly, we can formulate the memory occupation of first-order row-column-wise binarization and our ARB-RC as

$$\mathcal{M}^{1st}_{\text{row-column-wise}} = \overbrace{n \times m}^{\mathbf{B}} + \overbrace{(n + m) \times 16}^{\text{FP16}\ \alpha_r\ and\ \alpha_c}, \tag{136}$$

$$\mathcal{M}_{\text{ARB-RC}} = \overbrace{2 \times n \times c + (\lceil m/k \rceil \times 2n + 2c) \times 16}^{second-order\ binarization} \tag{137}$$

$$+ \overbrace{n \times (m - c) + (\lceil m/k \rceil \times n + (m - c)) \times 16 \times 2}^{first-order\ binarization} \tag{138}$$

$$\underbrace{\phantom{(\lceil m/k \rceil \times n + (m - c)) \times 16 \times 2}}_{2\ groups}$$

$$+ \overbrace{n \times m}^{group\ bitmap} + \overbrace{m}^{salient\ column\ bitmap}. \tag{139}$$

In addition, if added CGB, i.e. our refined strategy for the combination of salient column bitmap and group bitmap, the memory requirement slightly increases due to more scaling factors, but still less than BiLLM. The total memory of ARB-RC + CGB is

$$\mathcal{M}_{\text{ARB-RC + CGB}} = \overbrace{2 \times n \times c + \underbrace{(\lceil m/k \rceil \times 2n + 2c) \times 16 \times 2}_{2\ groups}}^{second-order\ binarization} \tag{140}$$

$$+ \overbrace{n \times (m - c) + \underbrace{(\lceil m/k \rceil \times n + (m - c)) \times 16 \times 2}_{2\ groups}}^{first-order\ binarization} \tag{141}$$

$$+ \overbrace{n \times m}^{group\ bitmap} + \overbrace{m}^{salient\ column\ bitmap}. \tag{142}$$

# F  VISUALIZATION DURING ALTERNATING REFINEMENT

Figure 1: The change of distribution shift (absolute difference between the mean of binarized and full-precision weights) during alternating refined binarization on LLaMA-7B. Each subfigure represents a block, with iteration 0 corresponding to the BiLLM method.

## F.1  DISTRIBUTION SHIFT

As shown in Figure 1, our Alternating Refined Binarization progressively reduces the distribution shift with fast convergence, where the initial distribution shift corresponds to BiLLM.

## F.2  COLUMN-WISE QUANTIZATION ERROR

As shown in Figure 2, We visualize the column-wise quantization error of a block in each layer of the LLM. The results indicate that our ARB-RC method can effectively reduce column-wise quantization error compared to previous row-wise binarization method.

## F.3  BINARIZATION PARAMETERS

As shown in Figure 3, we visualize the changes of alpha and mean during Alternating Refined Binarization. It is evident that all alpha values increase beyond their initial estimates, as supported by our analysis of quantization error in Equation (121). This suggests that alpha was underestimated by previous binarization methods.

# G  MORE EXPERIMENTAL RESULTS

**Comparison on PTB and C4.**  Due to the page limit, we provide the perplexity comparison on PTB dataset for LLaMA and OPT families in Table 1 and Table 3 respectively. Similarly, the comparisons on C4 dataset for LLaMA and OPT families are provided in Table 2 and Table 4 respectively.

**Comparison on 7 zero-shot QA datasets.**  We also provide the comparison of 7 zero-shot QA datasets on OPT family, as shown in Table 5.

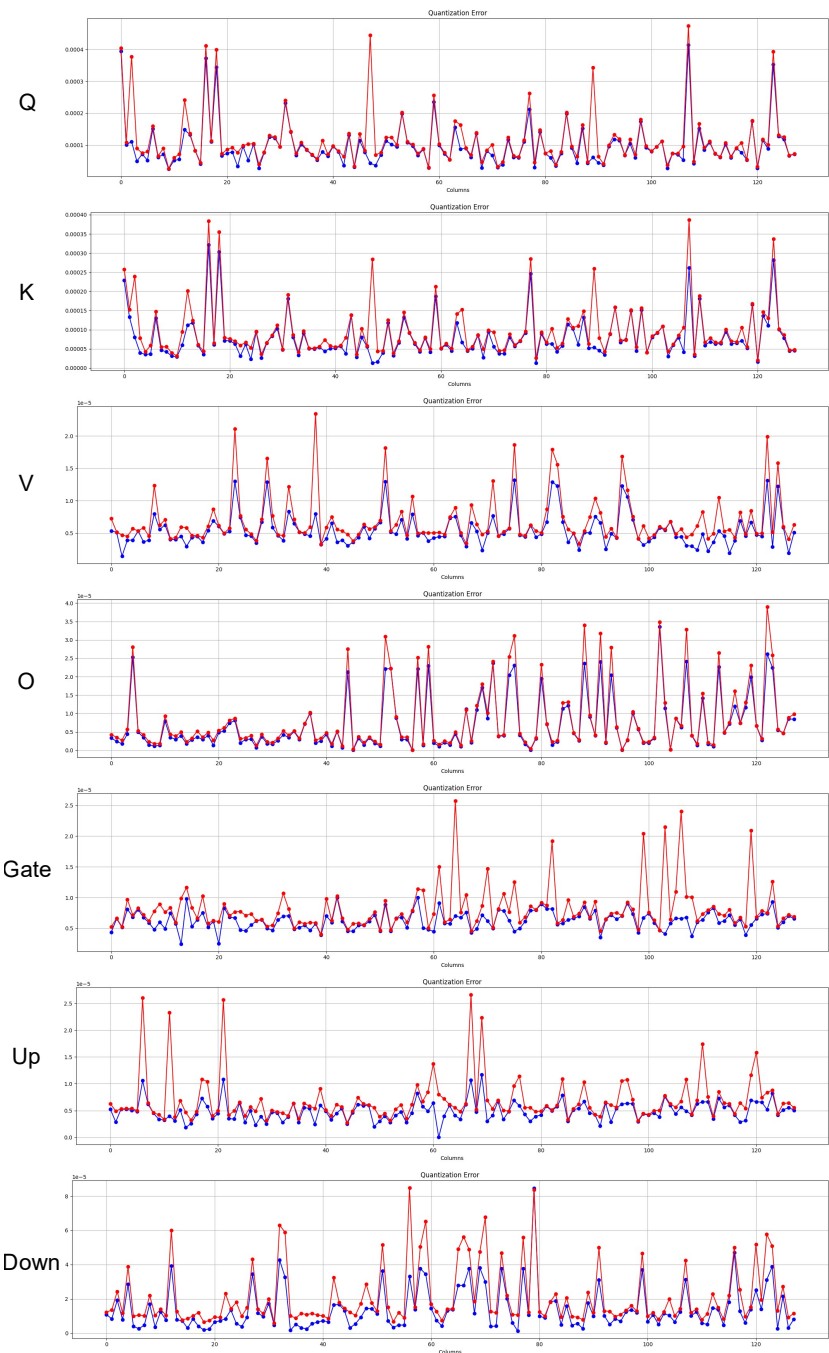

Figure 2: Quantization error comparison between row-wise binarization (red curve) and ARB-RC (blue curve) on LLaMA-7B. We display the error along columns, with each subfigure representing a block. The blue curve is notably lower than the red curve, with the difference being particularly pronounced in the *Gate Project*, *Up Project*, and *Down Project* layers.

**Evaluation with other metrics.** We conduct additional experiments on LLaMA-7B, measuring the F1 score on the SQuADv2 dataset and chrF on the WMT2014 (En-Fr) dataset. As shown in Table 6, our ARB-LLM significantly outperforms previous binarization methods, PB-LLM and BiLLM, in both F1 score and chrF metrics, further demonstrating the effectiveness of our proposed method.

**Evaluation on SQuADv2, SWAG, and MMLU College Mathematics datasets.** We conduct additional experiments on the long context dataset SQuADv2, math dataset MMLU College Mathematics,

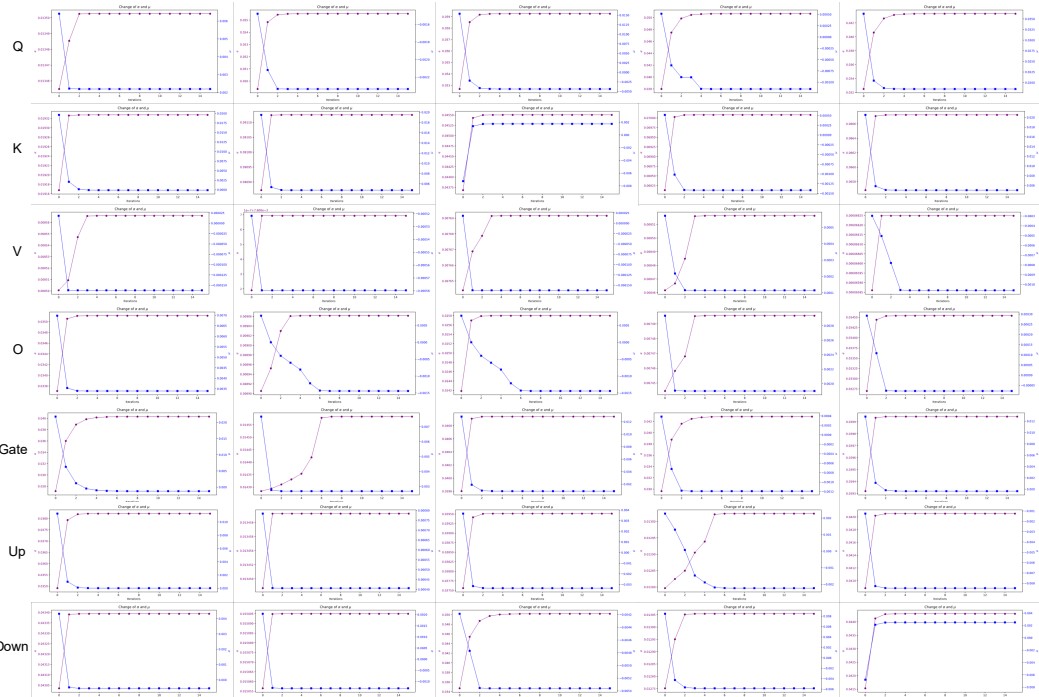

Figure 3: The changes of alpha and mean during alternating refinement on LLaMA-7B, with each subfigure representing a block. The red curve on the left represents the change of alpha, while the blue curve represents the change of mean. All alpha values exceed their initial estimates, indicating that alpha is underestimated in standard binarization.

and reasoning dataset SWAG. As shown in Table 7, our ARB-LLM also outperforms the previous binarization methods PB-LLM and BiLLM on these datasets, narrowing the performance gap with FP16, especially on the long-context SQuADv2 dataset.

**Comparison on Phi-3 models.** We conduct additional experiments by binarizing Phi-3-mini (3.8B) and Phi-3-medium (14B), and evaluate their perplexity (PPL) on WikiText2. As shown in Table 8, our ARB-LLM consistently outperforms previous binarization methods, PB-LLM and BiLLM. Moreover, the performance gap between binarized and FP16 models is reasonable. Compared to the binarization of OPT, the results for Phi-3-mini (3.8B) surpass those of OPT (2.7B), and the results for Phi-3-medium (14B) outperform OPT (13B).

**Comparison of runtime inference.** Evaluating runtime performance is crucial for demonstrating the practical feasibility of our proposed implementation. Unfortunately, previous works such as BiLLM and PB-LLM, did not report runtime performance due to the lack of a CUDA kernel for matrix multiplication between FP activation and 1-bit weights. We use the BitBLAS codebase to benchmark our method and comparable approaches, providing detailed runtime evaluations. We evaluate the runtime inference metrics by measuring the latency (ms) of various linear layers in LLaMA-7B and LLaMA-13B. The sequence length of input tensor X is 2048, and experiments are conducted on an NVIDIA A6000 GPU. As shown in Table 9, our method demonstrates significant improvements in inference speed compared to FP16 and PB-LLM. PB-LLM is slower due to the Int8-to-FP16 matrix multiplication. Moreover, both ARB-LLM-X and ARB-LLM-RC achieve a speed similar to BiLLM, while largely improving the performance.

**Pareto curve.** We present the Pareto curves of binarization methods PB-LLM, BiLLM, and our ARB-LLM (all with CSR compressed bitmap), as well as the low-bit quantization methods GPTQ in Figure 4. Among GPTQ models, the 4-bit version achieves the highest accuracy for a given memory budget compared to its 2-bit, 3-bit, and 8-bit counterparts. However, our ARB-LLM still outperforms 4-bit GPTQ on the Pareto curve. Moreover, low-bit quantization methods like GPTQ suffer significant accuracy degradation at extremely low bit levels (e.g., 2-bit). In contrast, our ARB-LLM excels in such scenarios, delivering superior performance while using less memory.

Table 1: Perplexity of RTN, GPTQ, PB-LLM, BiLLM, and our methods on LLaMA family. The columns represent the perplexity results on the **PTB** dataset with different model sizes. N/A: LLaMA2 lacks a 30B version, and LLaMA3 lacks both 13B and 30B versions. *: LLaMA has a 65B version, while both LLaMA2 and LLaMA3 have 70B versions.

| Model | Method | Block Size | Weight Bits | 7B/8B* | 13B | 30B | 65B/70B* |
|---|---|---|---|---|---|---|---|
| | Full Precision | - | 16.00 | 41.15 | 28.10 | 23.51 | 25.07 |
| | RTN | - | 3.00 | 329.78 | 64.53 | 80.46 | 81.57 |
| | GPTQ | 128 | 3.00 | 84.88 | 26.40 | 20.22 | 19.55 |
| | RTN | - | 2.00 | 126501.65 | 84172.61 | 32162.31 | 21743.58 |
| | GPTQ | 128 | 2.00 | 1421.47 | 224.45 | 69.46 | 47.70 |
| **LLaMA** | RTN | - | 1.00 | 155213.47 | 1960633.38 | 14821.51 | 68358.99 |
| | GPTQ | 128 | 1.00 | 121586.44 | 104769.39 | 10959.79 | 20192.53 |
| | PB-LLM | 128 | 1.70 | 603.57 | 237.22 | 114.35 | 119.19 |
| | BiLLM | 128 | 1.09 | 373.81 | 84.87 | 43.10 | 44.68 |
| | **ARB-LLM$_X$** | **128** | **1.09** | **281.70** | **81.50** | **38.07** | **36.08** |
| | **ARB-LLM$_{RC}$** | **128** | **1.09** | **195.94** | **54.38** | **34.65** | **32.20** |
| | Full Precision | - | 16.00 | 37.91 | 50.93 | N/A | 24.25 |
| | RTN | - | 3.00 | 1680.32 | 228.12 | N/A | 63.24 |
| | GPTQ | 128 | 3.00 | 4825.75 | 40.33 | N/A | 18.26 |
| | RTN | - | 2.00 | 24786.95 | 51250.84 | N/A | 29383.20 |
| | GPTQ | 128 | 2.00 | 5583.96 | 419.07 | N/A | 50.51 |
| **LLaMA2** | RTN | - | 1.00 | 99798.48 | 38487.07 | N/A | 110548.30 |
| | GPTQ | 128 | 1.00 | 66784.62 | 27741.64 | N/A | 14379.46 |
| | PB-LLM | 128 | 1.70 | 657.24 | 816.31 | N/A | NAN |
| | BiLLM | 128 | 1.08 | 5243.01 | 309.12 | N/A | 72.02 |
| | **ARB-LLM$_X$** | **128** | **1.08** | **681.24** | **182.10** | **N/A** | **49.18** |
| | **ARB-LLM$_{RC}$** | **128** | **1.08** | **389.59** | **198.17** | **N/A** | **32.79** |
| | Full Precision | - | 16.00 | 11.18 | N/A | N/A | 8.53 |
| | RTN | - | 3.00 | 1869.24 | N/A | N/A | 16180.72 |
| | GPTQ | 128 | 3.00 | 18.83 | N/A | N/A | 15.97 |
| | RTN | - | 2.00 | 633297.75 | N/A | N/A | 374834.19 |
| | GPTQ | 128 | 2.00 | 717.24 | N/A | N/A | 79.20 |
| **LLaMA3** | RTN | - | 1.00 | 764941.75 | N/A | N/A | 227967.19 |
| | GPTQ | 128 | 1.00 | 978209.31 | N/A | N/A | 118912.35 |
| | PB-LLM | 128 | 1.70 | 106.25 | N/A | N/A | 45.13 |
| | BiLLM | 128 | 1.06 | 87.25 | N/A | N/A | 97.13 |
| | **ARB-LLM$_X$** | **128** | **1.06** | **53.86** | **N/A** | **N/A** | **23.13** |
| | **ARB-LLM$_{RC}$** | **128** | **1.06** | **45.49** | **N/A** | **N/A** | **15.34** |

**Results of BiLLM.** We strictly follow the BiLLM codebase to reproduce the results. However, the experiments were conducted on a different GPU, and some package versions may differ. These slight variations in the experimental environment are likely the primary cause of any discrepancies. As shown in Table 10, for these two models, more than half of the reproduced results are better than those reported in the original paper. Whether compared against the original results or the reproduced ones, our ARB-LLM consistently outperforms BiLLM.

## H  DIALOG EXAMPLES

As shown in Figure 5, we provide some dialogue examples of PB-LLM, BiLLM, and our ARB-LLM$_{RC}$ on LLaMA-13B and Vicuna-13B models.

Table 2: Perplexity of RTN, GPTQ, PB-LLM, BiLLM, and our methods on LLaMA family. The columns represent the perplexity results on the **C4** dataset with different model sizes. N/A: LLaMA2 lacks a 30B version, and LLaMA3 lacks both 13B and 30B versions. *: LLaMA has a 65B version, while both LLaMA2 and LLaMA3 have 70B versions.

| Model | Method | Block Size | Weight Bits | 7B/8B* | 13B | 30B | 65B/70B* |
|---|---|---|---|---|---|---|---|
| | Full Precision | - | 16.00 | 7.34 | 6.80 | 6.13 | 5.81 |
| | RTN | - | 3.00 | 28.24 | 13.24 | 28.58 | 12.76 |
| | GPTQ | 128 | 3.00 | 9.95 | 7.16 | 6.51 | 6.03 |
| | RTN | - | 2.00 | 112668.16 | 58515.73 | 27979.50 | 22130.23 |
| | GPTQ | 128 | 2.00 | 79.06 | 18.97 | 14.86 | 10.23 |
| **LLaMA** | RTN | - | 1.00 | 194607.78 | 1288356.88 | 13556.87 | 135027.31 |
| | GPTQ | 128 | 1.00 | 186229.5 | 108958.73 | 9584.84 | 23965.75 |
| | PB-LLM | 128 | 1.70 | 76.63 | 40.64 | 25.16 | 15.30 |
| | BiLLM | 128 | 1.09 | 46.96 | 16.83 | 12.11 | 11.09 |
| | **ARB-LLM$_X$** | **128** | **1.09** | **22.73** | **13.86** | **10.93** | **9.64** |
| | **ARB-LLM$_{RC}$** | **128** | **1.09** | **17.92** | **12.48** | **10.09** | **8.91** |
| | Full Precision | - | 16.00 | 7.26 | 6.73 | N/A | 5.71 |
| | RTN | - | 3.00 | 384.02 | 12.50 | N/A | 10.03 |
| | GPTQ | 128 | 3.00 | 7.95 | 7.06 | N/A | 5.88 |
| | RTN | - | 2.00 | 30843.15 | 51690.40 | N/A | 27052.53 |
| | GPTQ | 128 | 2.00 | 35.27 | 19.66 | N/A | 9.55 |
| **LLaMA2** | RTN | - | 1.00 | 115058.76 | 46250.21 | N/A | 314504.09 |
| | GPTQ | 128 | 1.00 | 67954.04 | 19303.51 | N/A | 13036.32 |
| | PB-LLM | 128 | 1.70 | 80.69 | 184.67 | N/A | NAN |
| | BiLLM | 128 | 1.08 | 39.38 | 25.87 | N/A | 17.30 |
| | **ARB-LLM$_X$** | **128** | **1.08** | **28.02** | **19.82** | N/A | **11.85** |
| | **ARB-LLM$_{RC}$** | **128** | **1.08** | **20.12** | **14.29** | N/A | **8.65** |
| | Full Precision | - | 16.00 | 9.45 | N/A | N/A | 7.17 |
| | RTN | - | 3.00 | 566.43 | N/A | N/A | 12285.45 |
| | GPTQ | 128 | 3.00 | 17.68 | N/A | N/A | 10.04 |
| | RTN | - | 2.00 | 777230.94 | N/A | N/A | 447601.09 |
| | GPTQ | 128 | 2.00 | 394.74 | N/A | N/A | 122.55 |
| **LLaMA3** | RTN | - | 1.00 | 1422473.38 | N/A | N/A | 188916.13 |
| | GPTQ | 128 | 1.00 | 1118313.13 | N/A | N/A | 126439.66 |
| | PB-LLM | 128 | 1.70 | 104.15 | N/A | N/A | 40.69 |
| | BiLLM | 128 | 1.06 | 61.04 | N/A | N/A | 198.86 |
| | **ARB-LLM$_X$** | **128** | **1.06** | **41.86** | N/A | N/A | **21.67** |
| | **ARB-LLM$_{RC}$** | **128** | **1.06** | **35.70** | N/A | N/A | **15.44** |

Table 3: Perplexity of RTN, GPTQ, PB-LLM, BiLLM, and our methods on **OPT** family. The columns represent the perplexity results on **PTB** datasets with different model sizes.

| Method | Block Size | Weight Bits | 1.3B | 2.7B | 6.7B | 13B | 30B | 66B |
|---|---|---|---|---|---|---|---|---|
| Full Precision | - | 16.00 | 20.29 | 17.97 | 15.77 | 14.52 | 14.04 | 13.36 |
| RTN | - | 3.00 | 8987.17 | 9054.89 | 4661.77 | 2474.14 | 1043.13 | 3647.87 |
| GPTQ | 128 | 3.00 | 17.54 | 15.15 | 12.86 | 11.93 | 11.28 | 11.42 |
| RTN | - | 2.00 | 8030.18 | 5969.35 | 17222.70 | 72388.19 | 105760.72 | 462581.28 |
| GPTQ | 128 | 2.00 | 110.93 | 58.38 | 22.73 | 17.81 | 14.19 | 62.04 |
| RTN | - | 1.00 | 11062.04 | 28183.08 | 11981.09 | 32157360.00 | 5435.99 | 147668.78 |
| GPTQ | 128 | 1.00 | 6524.99 | 8405.25 | 5198.99 | 3444847.25 | 7158.62 | 5737.15 |
| PB-LLM | 128 | 1.70 | 324.62 | 183.97 | 169.49 | 101.00 | 41.87 | 45.32 |
| BiLLM | 128 | 1.11 | 115.94 | 88.52 | 69.41 | 27.16 | 21.41 | 18.51 |
| **ARB-LLM$_X$** | **128** | **1.11** | **71.96** | **54.28** | **31.23** | **23.46** | **19.28** | **17.64** |
| **ARB-LLM$_{RC}$** | **128** | **1.11** | **43.34** | **31.77** | **22.31** | **18.81** | **16.88** | **15.66** |

Table 4: Perplexity of RTN, GPTQ, PB-LLM, BiLLM, and our methods on **OPT** family. The columns represent the perplexity results on **C4** datasets with different model sizes.

| Method | Block Size | Weight Bits | 1.3B | 2.7B | 6.7B | 13B | 30B | 66B |
|---|---|---|---|---|---|---|---|---|
| Full Precision | - | 16.00 | 16.07 | 14.34 | 12.71 | 12.06 | 11.45 | 10.99 |
| RTN | - | 3.00 | 5039.85 | 11165.54 | 5022.57 | 2550.72 | 1030.62 | 3394.97 |
| GPTQ | 128 | 3.00 | 16.11 | 14.17 | 12.29 | 11.54 | 10.91 | 11.05 |
| RTN | - | 2.00 | 7431.04 | 7387.40 | 13192.40 | 89517.66 | 61213.64 | 823566.00 |
| GPTQ | 128 | 2.00 | 63.06 | 35.81 | 18.60 | 16.29 | 12.92 | 33.03 |
| RTN | - | 1.00 | 9999.56 | 23492.89 | 9617.07 | 23436088.00 | 5041.77 | 113236.92 |
| GPTQ | 128 | 1.00 | 6364.65 | 6703.36 | 5576.82 | 1799217.88 | 7971.37 | 7791.47 |
| PB-LLM | 128 | 1.70 | 168.12 | 222.15 | 104.78 | 57.84 | 27.67 | 27.73 |
| BiLLM | 128 | 1.11 | 64.14 | 44.77 | 42.13 | 19.83 | 16.17 | 14.16 |
| **ARB-LLM$_\mathbf{X}$** | **128** | **1.11** | **47.60** | **34.97** | **22.54** | **17.71** | **14.71** | **13.32** |
| **ARB-LLM$_\mathbf{RC}$** | **128** | **1.11** | **28.19** | **21.46** | **16.97** | **15.01** | **13.34** | **12.43** |

Table 5: Accuracy of 7 QA datasets on **OPT** family. We compare the results among GPTQ, PB-LLM, BiLLM, **ARB-LLM$_\mathbf{X}$**, and **ARB-LLM$_\mathbf{RC}$** to validate the quantization effect.

| Model | Method | Weight Bits | PIQA ↑ | BoolQ ↑ | OBQA ↑ | Winogrande ↑ | ARC-e ↑ | ARC-c ↑ | Hellaswag ↑ | Average ↑ |
|---|---|---|---|---|---|---|---|---|---|---|
| | GPTQ | 2.00 | 59.47 | 42.66 | 15.80 | 50.04 | 37.21 | 21.42 | 30.92 | 36.79 |
| | PB-LLM | 1.70 | 54.57 | 61.77 | 13.00 | 50.99 | 28.79 | 20.56 | 26.55 | 36.60 |
| OPT-1.3B | BiLLM | 1.09 | 59.52 | 61.74 | 14.80 | 52.17 | 36.53 | 17.83 | 29.64 | 38.89 |
| | **ARB-LLM$_\mathbf{X}$** | **1.09** | **62.84** | **61.99** | **13.40** | **52.17** | **43.43** | **18.94** | **30.86** | **40.52** |
| | **ARB-LLM$_\mathbf{RC}$** | **1.09** | **65.45** | **60.31** | **15.40** | **53.04** | **48.27** | **19.37** | **33.44** | **42.18** |
| | GPTQ | 2.00 | 61.81 | 54.43 | 15.40 | 52.33 | 40.15 | 20.56 | 32.55 | 39.60 |
| | PB-LLM | 1.70 | 56.42 | 62.23 | 12.80 | 50.12 | 31.61 | 18.60 | 27.61 | 37.06 |
| OPT-2.7B | BiLLM | 1.09 | 62.57 | 62.20 | 15.40 | 52.57 | 39.65 | 19.80 | 30.88 | 40.44 |
| | **ARB-LLM$_\mathbf{X}$** | **1.09** | **65.61** | **62.08** | **14.80** | **53.59** | **47.22** | **19.62** | **32.57** | **42.21** |
| | **ARB-LLM$_\mathbf{RC}$** | **1.09** | **68.50** | **61.99** | **21.60** | **58.33** | **52.82** | **22.27** | **37.50** | **46.14** |
| | GPTQ | 2.00 | 69.37 | 55.05 | 21.20 | 55.80 | 56.06 | 23.38 | 41.29 | 46.02 |
| | PB-LLM | 1.70 | 56.47 | 55.57 | 13.20 | 50.28 | 29.97 | 18.69 | 27.50 | 35.95 |
| OPT-6.7B | BiLLM | 1.09 | 58.60 | 62.14 | 13.20 | 53.12 | 33.75 | 18.26 | 28.83 | 38.27 |
| | **ARB-LLM$_\mathbf{X}$** | **1.09** | **69.75** | **62.20** | **17.80** | **58.64** | **55.47** | **24.32** | **37.78** | **46.57** |
| | **ARB-LLM$_\mathbf{RC}$** | **1.09** | **72.47** | **62.87** | **22.20** | **60.62** | **59.09** | **26.79** | **42.08** | **49.45** |
| | GPTQ | 2.00 | 66.54 | 56.51 | 18.60 | 59.12 | 48.53 | 24.06 | 41.34 | 44.96 |
| | PB-LLM | 1.70 | 57.29 | 62.17 | 12.80 | 51.22 | 30.93 | 20.56 | 26.83 | 37.40 |
| OPT-13B | BiLLM | 1.09 | 68.72 | 62.32 | 18.00 | 59.91 | 54.71 | 26.37 | 39.02 | 47.00 |
| | **ARB-LLM$_\mathbf{X}$** | **1.09** | **71.98** | **62.57** | **21.20** | **61.40** | **59.72** | **26.02** | **41.45** | **49.19** |
| | **ARB-LLM$_\mathbf{RC}$** | **1.09** | **73.56** | **65.93** | **24.20** | **64.25** | **62.54** | **29.52** | **45.14** | **52.16** |
| | GPTQ | 2.00 | 73.88 | 63.94 | 24.20 | 62.19 | 60.77 | 28.24 | 47.88 | 51.59 |
| | PB-LLM | 1.70 | 66.76 | 62.29 | 17.40 | 51.07 | 49.33 | 22.53 | 36.53 | 43.70 |
| OPT-30B | BiLLM | 1.09 | 72.74 | 62.35 | 21.00 | 60.14 | 60.69 | 27.56 | 42.81 | 49.61 |
| | **ARB-LLM$_\mathbf{X}$** | **1.09** | **74.27** | **62.39** | **23.60** | **64.25** | **63.51** | **28.33** | **46.04** | **51.77** |
| | **ARB-LLM$_\mathbf{RC}$** | **1.09** | **75.08** | **65.78** | **26.40** | **65.43** | **64.81** | **29.69** | **48.59** | **53.68** |
| | GPTQ | 2.00 | 57.62 | 57.13 | 13.20 | 51.85 | 36.11 | 21.67 | 34.01 | 38.80 |
| | PB-LLM | 1.70 | 72.74 | 62.54 | 24.20 | 63.46 | 60.10 | 30.20 | 43.13 | 50.91 |
| OPT-66B | BiLLM | 1.09 | 75.08 | 65.26 | 25.60 | 65.43 | 65.66 | 31.40 | 47.54 | 53.71 |
| | **ARB-LLM$_\mathbf{X}$** | **1.09** | **75.79** | **66.27** | **27.00** | **66.77** | **67.51** | **32.76** | **49.33** | **55.06** |
| | **ARB-LLM$_\mathbf{RC}$** | **1.09** | **76.88** | **70.89** | **28.60** | **66.22** | **69.28** | **33.62** | **51.23** | **56.67** |

## I  LIMITATIONS

**Combination of ARB-X and ARB-RC.**  We find that it is hard to incorporate the calibration data into the update of column scaling factors. After initializing the row and column scaling factors, we take the derivative of quantization error $\mathcal{L}_2$ with respect to $\alpha^c$ and set it to zero:

$$\frac{\partial \mathcal{L}}{\partial \alpha_t^c} = \sum_k \mathbf{S}_{kt} \sum_j (-\alpha_j^r \mathbf{B}_{jt} \mathbf{W}_{jk} + (\alpha_j^r)^2 \alpha_k^c \mathbf{B}_{jt} \mathbf{B}_{jk}) = 0, \quad \text{where } t = 1, 2, \ldots, m. \quad (143)$$

Table 6: Comparison on SQuADv2 (F1 score) and WMT2014 En-Fr (chrF).

| Method | SQuADv2 (F1 score↑) | WMT2014 En-Fr (chrF↑) |
|---|---|---|
| FP16 | 19.45 | 28.89 |
| PB-LLM | 2.78 | 14.27 |
| BiLLM | 3.55 | 17.45 |
| **ARB-LLM$_X$** | **8.23** | **23.90** |
| **ARB-LLM$_{RC}$** | **12.24** | **19.22** |

Table 7: Comparison on SQuADv2, SWAG, and MMLU College Mathematics datasets.

| Method | SQuADv2 (F1↑) | SWAG (Acc↑) | MMLU College Mathematics (Acc↑) |
|---|---|---|---|
| FP16 | 19.45 | 0.57 | 0.36 |
| PB-LLM | 2.78 | 0.31 | 0.20 |
| BiLLM | 3.55 | 0.36 | 0.21 |
| **ARB-LLM$_X$** | **8.23** | **0.41** | **0.22** |
| **ARB-LLM$_{RC}$** | **12.24** | **0.44** | **0.23** |

Table 8: Perplexity of WikiText2 on Phi-3 models.

| Model | Phi-3-mini (3.8B) | Phi-3-medium (14B) |
|---|---|---|
| FP16 | 5.82 | 4.02 |
| PB-LLM | 377.98 | 754.27 |
| BiLLM | 21.03 | 10.33 |
| **ARB-LLM$_X$** | **18.32** | **9.31** |
| **ARB-LLM$_{RC}$** | **17.32** | **8.97** |

Table 9: Comparison of runtime inference (ms) on LLaMA-1/2-7B and LLaMA-1/2-13B.

| Model | LLaMA-1/2-7B | | | LLaMA-1/2-13B | | |
|---|---|---|---|---|---|---|
| **Weight Size** | 4096×4096 | 4096×11008 | 11008×4096 | 5120×5120 | 5120×13824 | 13824×5120 |
| FP16 | 0.76595 | 1.63532 | 1.76949 | 0.91443 | 2.68492 | 2.71254 |
| PB-LLM | 0.73363 | 1.44076 | 1.69881 | 0.83148 | 2.17292 | 2.19443 |
| BiLLM | 0.34201 | 0.36777 | 0.37689 | 0.35948 | 0.48947 | 0.49406 |
| **ARB-LLM$_{RC}$** | 0.35974 | 0.37218 | 0.37981 | 0.36312 | 0.49801 | 0.50038 |
| **ARB-LLM$_X$** | 0.33180 | 0.35539 | 0.36792 | 0.35505 | 0.47788 | 0.48275 |

Table 10: Perplexity of WikiText2 on LLaMA-1 and LLaMA-2. ‡We reproduce BiLLM based on their codebase.

| Model | LLaMA-1 | | | | LLaMA-2 | | |
|---|---|---|---|---|---|---|---|
| | 7B | 13B | 30B | 65B | 7B | 13B | 70B |
| BiLLM | 35.04 | 15.14 | 10.52 | 8.49 | 32.48 | 16.77 | 8.41 |
| BiLLM‡ | 49.79 | 14.58 | 9.90 | 8.37 | 32.31 | 21.35 | 13.32 |
| **ARB-LLM$_X$** | **21.81** | **11.20** | **8.66** | **7.27** | **21.61** | **14.86** | **7.88** |
| **ARB-LLM$_{RC}$** | **14.03** | **10.18** | **7.75** | **6.56** | **16.44** | **11.85** | **6.16** |

We observe that during the process of updating $\alpha_t^c$, the derivative of the quantization error with respect to $\alpha_t^c$ includes terms involving other $\alpha_l^c$. This indicates that introducing a calibration set results in coupling between $\alpha^c$ values, complicating their updates. Incorporating calibration data into ARB-RC presents a promising direction for future work.

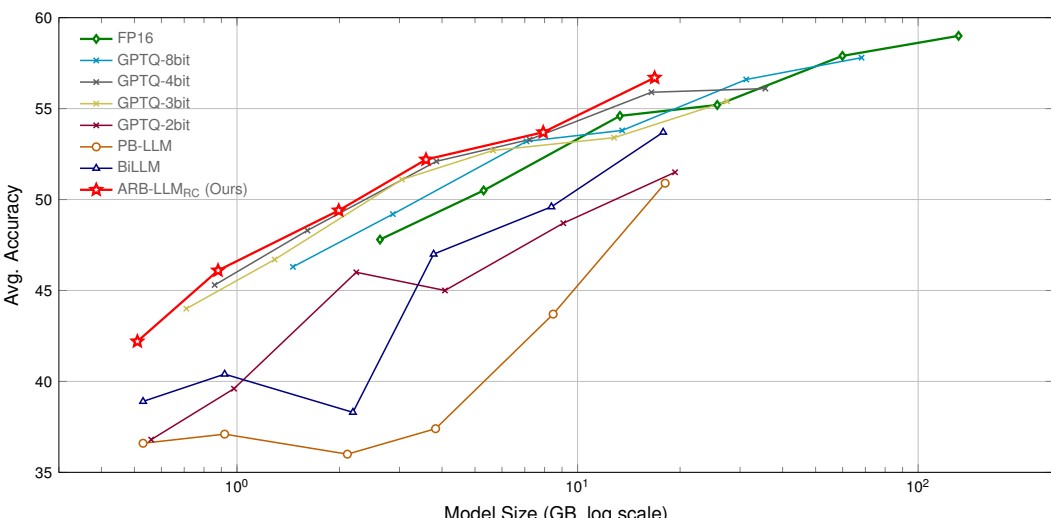

Figure 4: The Pareto curves of binarization and low-bit quantization methods demonstrate that our ARB-LLM outperforms all other approaches within the same memory constraints. Low-bit methods like GPTQ suffer significant performance degradation at extremely low bit levels (e.g., 2-bit), whereas our method maintains strong performance.

LLaMA-13B

| | |
|---|---|
| Prompt : | Florida is a state of ___ |
| **PB-LLM-1.7bit :** | Florida is a atate of is a is that is a is that is a is that is that is |
| **BiLLM-1.1bit :** | Florida is a state of the art business in the works. |
| **ARB-LLM_RC-1.1bit :** | Florida is a state of the United States and covers an area of 57,680 hectares. The Florida surface comprises mostly of salt beaches. |
| Prompt : | Amazon River locate in ___ |
| **PB-LLM-1.7bit :** | Amazon River locate in all of this way. You can see me, and you can hear me. |
| **BiLLM-1.1bit :** | Amazon River locate in front of the road, in the rear, in the lane, in the lane. |
| **ARB-LLM_RC-1.1bit :** | Amazon River locate in the Amazon rainforest. Amazon River is a river with the longest length in the world. |
| Prompt : | Having hiked through the steep mountain trail for hours, he was exhausted, his clothes soaked with sweat, and his legs trembling from the effort. Seeing his condition, I suggested he ___ |
| **PB-LLM-1.7bit :** | I suggested he could was a piss, with his hiked piss wisp. He was sopped. He had a bust. |
| **BiLLM-1.1bit :** | I suggested he to come close. And he was glad. I could be a good thing by a good action. |
| **ARB-LLM_RC-1.1bit :** | I suggested he went up to mountain for 8km again but when he couldn't even run for ten minutes, he turned his head. He went back to the bottom and went to the mountain for another time. |

Vicuna-13B

| | |
|---|---|
| Prompt : | I failed my physics exam, and I am unsure what steps to take next. What should I do? |
| **PB-LLM-1.7bit :** | Ne Ne and it looked like much o Ne' ne because sheâNe ne |
| **BiLLM-1.1bit :** | Please assist me. |
| **ARB-LLM_RC-1.1bit :** | Well, it could be best first to speak with your teacher about how exactly you can try to improve your performance. If you are sure that you failed the exam, get a retake. |
| Prompt : | My friend is studying in Rome, and I want to visit. What's the easiest way to get there from Madrid? |
| **PB-LLM-1.7bit :** | Mr.” I’ leads she_ home inside of no is often. I didn. I needed outside off. I didn. I needed outsideoff. I was. I looked inside. I am! Unmann. I needs. I. I would. I. L. I would. I. It.I. I would have. I had. |
| **BiLLM-1.1bit :** | Can you guide me through the process of buying tickets from Madrid? Can you help me travel from Madrid to a specific destination in Japan where I could go to the place because I don't want to do it. I don't have the ability to do it, but if you could tell me. |
| **ARB-LLM_RC-1.1bit :** | The easiest way to get from Madrid to Rome is to take a flight from Madrid to Rome. The flight will take approximately 3-5 hours, depending on the specific airline and the time it takes to get to Rome. The flight from Madrid to Rome is usually relatively cheap and can be done by any Spanish airline, like Vuelta, Iberia and many other. |
| Prompt : | After running a marathon, I was completely exhausted and out of breath. What should I do to recover? |
| **PB-LLM-1.7bit :** | I. I was too and she said.I. I didn. It would give. It was. I was. It. It is. It’ It was. |
| **BiLLM-1.1bit :** | I already asked my doctor friend. He suggested what to do now but what to do, how to do it, or to do, how to do, how to do, how to do... |
| **ARB-LLM_RC-1.1bit :** | During recovery, it's important to relax, rest, and replenish your body. You may need to rest for at least a week to recover, or you might be a little better. Some people might feel better after a day after running a marathon. In order to recover, I recommend that you rest on Sunday and Monday, and then be ready and full of energy by Tuesday. |

Figure 5: Conversation examples on LLaMA-13B (language supplementary) and Vicuna-13B (Q&A). We compare our best method ARB-LLM_RC with PB-LLM and BiLLM. Inappropiate and reasonable responses are shown in corresponding colors.