# OpenReview forum: "ARB-LLM: Alternating Refined Binarizations for Large Language Models"
_ICLR.cc/2025/Conference — ICLR 2025 Poster_

### Official Review · Reviewer_muCn · 2024-11-01

**Soundness:** 3
**Presentation:** 3
**Contribution:** 3
**Rating:** 8
**Confidence:** 3

**Summary:**

Authors propose to ALTERNATING REFINED BINARIZATION (ARB) which is based on BiLLM paper. They introduce correction term δµ to the original mean µ, effectively mitigating the distribution shift. As a result α, B and µ have to be refined.

They further extend ARB to ARB-X and ARB-RC, where
ARB-X is extension of ARB with calibration data and loss function which is based on papers Optimal Brain Compression and GPTQ.

ARB-RC introduces a column-wise scaling factor αc to better handle parameter variations across columns, while eliminating the redistribution parameter µ to enhance compression in LLM.

In addition they combine ARB-X and ARB-RC with weight partition strategy based on columngroup bitmap (CGB).
And call final versions as ARB-LLMX and ARB-LLMRC respectively, which outperform state-of-the-art (SOTA) binarization methods for LLMs.

ARB-LLMRC outperforms FP16 models of the same size (in GB).

**Strengths:**

Authors proposed several improvements of BiLLM method and show that ARB-LLMRC outperforms FP16 models of the same size (in GB) and outperforms BiLLM (it is SOTA of LLM post training binarization)

Extensive ablation study of the proposed methods showed effectiveness of Effectiveness of ARB, ARB-LLMX, ARB-LLMRC its combination with CGB and calibration data size, iteration number and group number.

Conducted extensive experiments on the LLaMA, LLaMA-2, and LLaMA3 families.

They did a thorough analysis of time and memory consumption for proposed methods.

They will release all the code and models.

**Weaknesses:**

Q: On Figure 1, authors show Pareto curve and demonstrate that  ARB-LLM-RC outperforms the same-size FP16 models and previously published binarized models. So authors show an improvements over previous models binarization, but it does not show the whole picture in terms of pareto curve: they only compare binarized models vs fp16 (on Figure 1). But it will be informative to show also 4bits and 8bits models. For example paper "The case for 4-bit precision: k-bit Inference Scaling Laws" shows that in terms of pareto curve, for the same model size, 4bits has the best accuracy. In addition to 4bits and 8bits it would be great to plot 3bits too on pareto curve (given that 3bits is already presented in multiple Tables in this paper).

**Questions:**

Q1: Is it ARB second-order binarization is called in the paper " higher-bit representation, i.e., second-order binarizatio".  Is effectively 2bits quantization?

Q2: Please explain why BiLLM accuracy in Table 2 does not match the results in the original BiLLM paper (shown in Table 3 LLaMA and LLaMA2)

---

> ### Author Response · Authors · 2024-11-21
> **Response to Reviewer muCn (denoted as R3)**
>
> `Q3-1:` On Figure 1, authors show Pareto curve and demonstrate that ARB-LLM-RC outperforms the same-size FP16 models and previously published binarized models. So authors show an improvements over previous models binarization, but it does not show the whole picture in terms of pareto curve: they only compare binarized models vs fp16 (on Figure 1). But it will be informative to show also 4bits and 8bits models. For example paper "The case for 4-bit precision: k-bit Inference Scaling Laws" shows that in terms of pareto curve, for the same model size, 4bits has the best accuracy. In addition to 4bits and 8bits it would be great to plot 3bits too on pareto curve (given that 3bits is already presented in multiple Tables in this paper).
>
> `A3-1:` Thank you for your valuable suggestion. We add the 2-bit, 3-bit, 4-bit, and 8-bit models to the Pareto curve. Due to the page limit, we put this big Pareto curve **in Figure 4 of the refined supplementary file**.
> 1. Among GPTQ models, the 4-bit version achieves the highest accuracy for a given memory budget compared to its 2-bit, 3-bit, and 8-bit counterparts. However, our ARB-LLM still outperforms 4-bit GPTQ on the Pareto curve.
> 2. Low-bit quantization methods like GPTQ suffer significant accuracy degradation at extremely low bit levels (e.g., 2-bit). In contrast, our ARB-LLM excels in such scenarios, delivering superior performance while using less memory.
>
> `Q3-2:` Is it ARB second-order binarization is called in the paper " higher-bit representation, i.e., second-order binarizatio". Is effectively 2bits quantization?
>
> `A3-2:` Thank you for your question. We provide additional explanation below.
> 1. "ARB second-order binarization" is equal to "higher-bit representation, i.e., second-order binarization" in our paper.
> 2. **It is different from 2-bit quantization.**
>     - **Second-order binarization** employs **two 1-bit matrices** (-1, +1) with scaling and shifting to approximate the full-precision matrix, whereas **2-bit quantization** uses a **single 2-bit matrix** (0, 1, 2, 3) along with scaling and shifting to approximate FP16.
>     - As claimed in DB-LLM [ref1], second-order binarization might be more informative compared to standard 2-bit quantization. Additionally, leveraging second-order binarization to approximate salient weights enhances consistency within a binarization framework.
>
> [ref1] DB-LLM: Accurate Dual-Binarization for Efficient LLMs, ACL, 2024.
>
> `Q3-3:` Please explain why BiLLM accuracy in Table 2 does not match the results in the original BiLLM paper (shown in Table 3 LLaMA and LLaMA2)
>
> `A3-3:` Thank you for your question. We provide the explanation below.
> 1. We strictly follow the BiLLM codebase to reproduce the results. However, the experiments were conducted on a different GPU, and some package versions may differ. These slight variations in the experimental environment are likely the primary cause of any discrepancies.
> 2. For these two models, more than half of the reproduced results are better than those reported in the original paper. **Whether compared against the original results or the reproduced ones, our ARB-LLM consistently outperforms BiLLM.** We also add this table to **Table 10 of the supplementary file**.
>
> | **Model**        | **LLaMA-7B** | **LLaMA-13B** | **LLaMA-30B** | **LLaMA-65B** |
> |--------------|------------------|---------------|--------------------|----------------|
> | BiLLM (paper)       | 35.04         | 15.14        | 10.52    |    8.49           |
> | BiLLM (reproduced)  | 49.79        | 14.58        | 9.90    |  8.37        |
> | ARB-LLM-X| 21.81           | 11.20        | 8.66                     |   7.27       |
> |ARB-LLM-RC| 14.03          | 10.18        | 7.75                      |   6.56      |
>
> | **Model**        | **LLaMA2-7B** | **LLaMA2-13B** | **LLaMA2-70B** |
> |--------------|------------------|---------------|----------------|
> | BiLLM (paper)       | 32.48         | 16.77        |    8.41           |
> | BiLLM (reproduced)  | 32.31        | 21.35        |  13.32        |
> | ARB-LLM-X| 21.61           | 14.86        |   7.88       |
> |ARB-LLM-RC| 16.44          | 11.85        |   6.16      |

---

### Official Review · Reviewer_H9m3 · 2024-11-04

**Soundness:** 3
**Presentation:** 3
**Contribution:** 3
**Rating:** 5
**Confidence:** 4

**Summary:**

The paper describes an Alternating Refined Binarization (ARB) framework for performing post-training quantization (PTQ) of 1-bit for large language models (LLM). The framework is based on the observation that there is a distribution gap between the binarized weights and the floating-point weights. Through the iterative refinement process of ARB, the quantization error can be reduced. The authors also provide two variants that use calibration data and address column deviation. They conduct experiments on OPT and LLAMA models, achieving better results compared to other binary PTQ methods such as BiLLM.

**Strengths:**

The paper identifies and addresses the distribution shift between floating-point weights and binary weights through an iterative refinement method. The study shows that through the ARB process, the quantization error decreases.

The paper further extends the method by adding a calibration dataset and column-wise scales, resulting in better performance compared to BiLLM models.

The paper offers a comprehensive ablation study and time/memory analysis, showing the trade-offs of each component.

**Weaknesses:**

The authors claim that the weight bit is 1.11 bits, but based on the memory comparison, the model takes approximately 3GB of memory on the Llama-7B model, which is similar in size to GPTQ-3bit of the same model, with inferior performance. Can the authors clarify this?

It seems the approach works better on the older OPT model than the Llama family, with Llama3's performance, in particular, being much worse than floating-point. Can the authors showcase the performance on more recent models like Phi/Llama3.2 to address this concern?

**Questions:**

1. Can the authors provide any runtime inference metrics to showcase the practical feasibility of the proposed implementation?

2. Could the authors introduce the meaning of each term in equation (6) so that the reader can understand without referring to prior papers?

3. Can the authors clarify the actual memory consumption and compare it to PTQ methods under the same memory budget?

---

> ### Author Response · Authors · 2024-11-21
> **Response to Reviewer H9m3 (denoted as R2) part 1**
>
> `Q2-1:` The authors claim that the weight bit is 1.11 bits, but based on the memory comparison, the model takes approximately 3GB of memory on the Llama-7B model, which is similar in size to GPTQ-3bit of the same model, with inferior performance. Can the authors clarify this?
>
> `A2-1:` Thank you for your question. For simplicity and consistency, we did not include storage optimization when calculating memory. This ensures a fair comparison among the binarization methods since all of them utilize bitmaps. In practice, as shown in prior works like PB-LLM and BiLLM, these methods employ the sparse matrix format CSR for bitmap storage. Therefore, when comparing to low-bit quantization methods that do not use bitmaps, it would be fair to include the CSR compression for bitmaps. Below, we provide additional results incorporating the compressed bitmap on LLaMA-7B.
>
> | Method | WikiText2 (PPL$\downarrow$) | Memory |
> | ----- | --------------------------- | ------ |
> | FP16  |    5.68 | 13.48 GB |
> | GPTQ-3bit  | 8.63 | 3.16 GB |
> | GPTQ-2bit  | 129.19 | 2.35 GB |
> | PB-LLM     | 82.76 | 2.21 GB |
> | BiLLM      | 49.79 | 2.19 GB |
> | ARB-LLM-X  | 21.81 | 2.49 GB |
> | ARB-LLM-RC | 14.03 | 2.09 GB |
>
> 1. The optimized memory usage of ARB-LLM and other binarized PTQ methods is comparable to GPTQ-2bit, while achieving significantly improved performance.
>
> 2. Our ARB-LLM-RC significantly **outperforms GPTQ-2bit and other binarized PTQ methods, while using less memory**.
> We also clarify this in **section 4.4 of the refined version**.
>
> `Q2-2:` It seems the approach works better on the older OPT model than the Llama family, with Llama3's performance, in particular, being much worse than floating-point. Can the authors showcase the performance on more recent models like Phi/Llama3.2 to address this concern?
>
> `A2-2:` Thank you for your valuable suggestion. We conduct additional experiments by binarizing **Phi-3-mini (3.8B), and Phi-3-medium (14B)**, and evaluate their perplexity (PPL) on WikiText2.
>
> | **Model**          | **Phi-3-mini (3.8B)** | **Phi-3-medium (14B)** |
> |----------------|------------|--------------|
> | FP16           | 5.82   | 4.02     |
> | PB-LLM         | 377.98 | 754.27   |
> | BiLLM          | 21.03  | 10.33    |
> | ARB-LLM-X  | 18.32  | 9.31      |
> | ARB-LLM-RC | 17.32  | 8.97     |
>
> 1. In the recent Phi models, our ARB-LLM consistently outperforms previous binarization methods, PB-LLM and BiLLM.
>
> 2. The performance gap between binarized and FP16 models is reasonable. Compared to the binarization of OPT, the results for Phi-3-mini (3.8B) surpass those of OPT (2.7B), and the results for Phi-3-medium (14B) outperform OPT (13B).
>
>
> `Q2-3:` Can the authors provide any runtime inference metrics to showcase the practical feasibility of the proposed implementation?
>
> `A2-3:` Thank you for raising the question about runtime inference metrics. We acknowledge that evaluating runtime performance is crucial for demonstrating the practical feasibility of our proposed implementation. Unfortunately, previous works such as BiLLM and PB-LLM, did not report runtime performance due to the lack of a CUDA kernel for matrix multiplication between FP activation and 1-bit weights. We use the BitBLAS [ref1] codebase to benchmark our method and comparable approaches, providing detailed runtime evaluations. We evaluate the runtime inference metrics by measuring the **latency (ms) of various linear layers in LLaMA-7B and LLaMA-13B**. The sequence length of input tensor X is 2048, and experiments are conducted on an NVIDIA A6000 GPU.
> | **Model**       | | **LLaMA-1/2-7B** | |
> | - | - | - | - |
> | Weight Size  | 4096$\times$4096 | 4096$\times$11008 | 11008$\times$4096 |
> | FP16         | 0.76595   | 1.63532    | 1.76949    |
> | PB-LLM       | 0.73363   | 1.44076    | 1.69881    |
> | BiLLM        | 0.34201   | 0.36777    | 0.37689    |
> | ARB-LLM-RC       | 0.35974   | 0.37218    | 0.37981    |
> | ARB-LLM-X        | 0.33180    | 0.35539    | 0.36792    |
>
> | **Model**     | | **LLaMA-1/2-13B** | |
> | - | - | - | - |
> | Weight Size | 5120$\times$5120 | 5120$\times$13824 | 13824$\times$5120 |
> | FP16        | 0.91443   | 2.68492    | 2.71254    |
> | PB-LLM      | 0.83148   | 2.17292    | 2.19443    |
> | BiLLM       | 0.35948   | 0.48947    | 0.49406    |
> | ARB-LLM-RC      | 0.36312   | 0.49801    | 0.50038    |
> | ARB-LLM-X       | 0.35505   | 0.47788    | 0.48275    |
>
> 1. Our method demonstrates **significant improvements** in inference speed **compared to FP16 and PB-LLM**. PB-LLM is slower due to the Int8-to-FP16 matrix multiplication.
>
> 2. Both ARB-LLM-X and ARB-LLM-RC achieve a **speed similar to BiLLM, while largely improving the performance**.
>
> [ref1] https://github.com/microsoft/BitBLAS

---

> > ### Comment · Reviewer_H9m3 · 2024-11-25
> >
> > Thanks author for sharing the per-layer latency, showing from 2x to 4x speed increase compared to floating point. I recognize the technical challenges involved. However I hope the author can demonstrate the end-to-end performance for both the Prefill and Decode stages to illustrate the practical application of Binary LLMs. For instance, a 4-bit kernel can easily achieve more than a 2x speed increase compared to floating point.[1]
> > I would like to maintain my current score.
> >
> > [1] QUIK: TOWARDS END-TO-END 4-BIT INFERENCE ON GENERATIVE LARGE LANGUAGE MODELS

---

> > > ### Author Response · Authors · 2024-11-26
> > >
> > > Thank you for clarifying the question. We use [ref1] to estimate the end-to-end acceleration for the Prefill and Decode stages. The table below compares the throughput improvements of the Prefill and Decode passes (single batch with 2048 tokens) for quantized models, relative to their FP16 counterparts. The base model used is LLaMA-3-8B, with inferences conducted on a **AMD EPYC 7763 CPU** ([ref1] will soon support NPU and GPU).
> > >
> > > | **Method** | **Prefill Throughput** | **Decode Throughput** |
> > > | ---------- | ----------- | ---------- |
> > > | FP16       | 1.44 tokens/s | 1.50 tokens/s |
> > > | ARB-LLM-X  | 8.97 tokens/s | 7.19 tokens/s |
> > > | ARB-LLM-RC | 8.89 tokens/s | 7.14 tokens/s |
> > >
> > > 1. Our binarization method achieves a **6x speedup in the Prefill phase** and a nearly **5x speedup in the Decode phase** compared to the full-precision method, demonstrating the practical utility of Binary LLMs. Although [ref1] currently supports only CPU, which limits the absolute throughput, the observed speedup clearly demonstrates the superiority of Binary LLMs.
> > > 2. Additionally, QUIK achieves a 2x speedup by **quantizing both weights and activations to 4-bit**, enabling acceleration through INT4 matrix multiplication. In contrast, ARB-LLM **only quantizes weights to 1-bit**, consistent with prior research such as PB-LLM and BiLLM.
> > > 3. As demonstrated in [ref1], the **matrix multiplication for Binary LLMs can be converted into addition operations**, as the weights are 1-bit. This enables Binary LLMs to offer significant acceleration potential compared to low-bit quantization.
> > >
> > > We hope that these results and explanations can address your concerns regarding the application of Binary LLMs.
> > >
> > > [ref1] https://github.com/microsoft/BitNet

---

> ### Author Response · Authors · 2024-11-21
> **Response to Reviewer H9m3 (denoted as R2) part 2**
>
> `Q2-4:` Could the authors introduce the meaning of each term in equation (6) so that the reader can understand without referring to prior papers?
>
> `A2-4:` Thank you for your suggestion. We add the explanation of each item **in the refined version (highlighted in red color)**.
>
> `Q2-5:` Can the authors clarify the actual memory consumption and compare it to PTQ methods under the same memory budget?
>
> `A2-5:` Thanks for your valuable feedback.
> 1. Compared to low-bit quantization, our ARB-LLM-RC **outperforms 2-bit GPTQ while using less memory**.
>
> 2. Compared to binarization methods, our ARB-LLM-RC largely **outperforms PB-LLM and BiLLM with less memory**.
>
> **We have responded to another similar question, `Q2-1`. Please refer to `A2-1` for more details.**

---

> ### Author Response · Authors · 2024-11-30
> **Further discussion with Reviewer H9m3**
>
> Dear Reviewer H9m3,
>
> We sincerely appreciate your valuable time and insightful comments. Regarding your remaining concern about the end-to-end speedup of Binary LLMs, we have taken your suggestions into account and conducted additional experiments to further demonstrate the practical benefits of our method.
>
> The results show that our approach achieves a **6x speedup in the Prefill phase** and a nearly **5x speedup in the Decode phase**, compared to FP16. These improvements are notably significant, especially when compared to the **2x speedup** achieved by the 4-bit quantization method QUIK.
>
> We believe our responses have covered your concerns, and we would appreciate your feedback on whether they adequately resolve the issue. Please let us know if there are any remaining aspects of our work that require further clarification.
>
> Thank you once again for your time and constructive feedback.
>
> Best,
>
> Authors

---

> ### Author Response · Authors · 2024-12-02
> **Second call for discussion with Reviewer H9m3**
>
> Dear Reviewer H9m3,
>
> We sincerely appreciate the time and effort you dedicated to reviewing our work, as well as your insightful comments. We have referred to your suggestions and conducted additional experiments to validate the end-to-end speedup of our method, which we believe have covered your concerns.
>
> Due to the approaching deadline for rebuttal and reviewer mmbQ responding that our response can address the concerns, we also hope to discuss further with you whether or not your concerns have been addressed. Please let us know if you still have any unclear parts of our work.
>
> Best,
>
> Authors

---

> ### Author Response · Authors · 2024-12-03
> **Third call for discussion with Reviewer H9m3**
>
> Dear Reviewer H9m3,
>
> We sincerely appreciate the time and effort you have invested in reviewing our work, as well as your insightful comments. In response to your suggestions, we have conducted additional experiments to validate the end-to-end speedup of our method, which we believe address your concerns.
>
> With the rebuttal deadline approaching and Reviewer mmbQ indicating that our response adequately addresses the raised issues, we would greatly appreciate any further feedback from you to ensure that all your concerns have been fully resolved. Please let us know if there are still any aspects of our work that remain unclear.
>
> Best，
>
> Authors

---

### Official Review · Reviewer_mmbQ · 2024-11-04

**Soundness:** 4
**Presentation:** 4
**Contribution:** 4
**Rating:** 8
**Confidence:** 2

**Summary:**

This paper proposes a novel post-training quantization method alternating Refined Binarization for quantizing LLM using binary weight. The existing methods suffering the mis-alignment from the distribution of the binarized weight with the full precision counterparts. The existing method also didn't address the column-wise deviation. The proposed framework progressively refines binarization parameters to minimize the quantization error. The proposed framework has 4 components. 1. alternating refined binarization which iteractively update the binarized weight. 2. The proposed method is also improved by calibration data. 3. A more accurate binarization can be achieved by column-wise scaling factor. 4. A column-group bitmap strategy was applied in the paper to improve quantization by maximizing bitmap utilization through salient and non-salient division. The experiment results show impressive results.

**Strengths:**

1. The proposed ARB-LLM method is innovative, which also addresses a critical problem in binary quantization.
2. The experiments on various LLM families demonstrate the superiority of the proposed methods.
3. The paper also provides thorough ablation studies the effectiveness of the individual components of ARB-LLM.

**Weaknesses:**

1. The main weakness of the paper is the evaluation seems limited to perplexity and average accuracy. The paper would be more interesting if there were trade-offs on more tasks.

**Questions:**

Does the proposed method perform consistently on 7 zero-shot QA datasets?

Is the binarized model able to get reasonable performance on long context, math and reasoning dataset?

---

> ### Author Response · Authors · 2024-11-21
> **Response to Reviewer mmbQ (denoted as R1)**
>
> `Q1-1:` The main weakness of the paper is the evaluation seems limited to perplexity and average accuracy. The paper would be more interesting if there were trade-offs on more tasks.
>
> `A1-1:` Thank you for your valuable feedback. In line with previous work, we primarily evaluate using PPL and accuracy. Yet, the evaluation can be extended to other metrics.
> We conduct additional experiments on LLaMA-7B, measuring the **F1 score on the SQuADv2 dataset** and **chrF on the WMT2014 (En-Fr) dataset**.
>
> | **Method** | **SQuADv2 (F1 score$\uparrow$)** | **WMT2014 En-Fr (chrF$\uparrow$)** |
> | ----- | ----- | ----- |
> | FP16 | 19.45 | 28.89 |
> | PB-LLM | 2.78 | 14.27 |
> | BiLLM | 3.55 | 17.45 |
> | ARB-LLM-X | 8.23 | 23.90 |
> | ARB-LLM-RC | 12.24 | 19.22 |
>
> Our ARB-LLM significantly outperforms previous binarization methods, PB-LLM and BiLLM, in both F1 score and chrF metrics, further demonstrating the effectiveness of our proposed method.
>
> `Q1-2:` Does the proposed method perform consistently on 7 zero-shot QA datasets?
>
> `A1-2:` Thank you for your question. We have presented a detailed accuracy comparison on seven zero-shot QA datasets in **Table 5 of the supplementary file**. Please refer to it for comprehensive details.
> 1. Compared to the SOTA method BiLLM, ARB-LLM demonstrates superior performance on most datasets, while achieving comparable results on one or two remaining datasets.
> 2. In the field of LLM quantization, it is a standard practice to evaluate performance across multiple datasets and emphasize average accuracy to mitigate the effects of randomness. For example, there is a fluctuation in accuracy between PB-LLM and BiLLM on individual datasets—PB-LLM outperforms BiLLM on some, while BiLLM leads on others. However, the average accuracy clearly shows that BiLLM significantly outperforms PB-LLM overall. Similarly, ARB-LLM surpasses BiLLM and even 2-bit GPTQ by a considerable margin in terms of average accuracy.
>
> `Q1-3:` Is the binarized model able to get reasonable performance on long context, math and reasoning dataset?
>
> `A1-3:` Thank you for your question. We conduct additional experiments on the **long context dataset SQuADv2, math dataset MMLU College Mathematics, and reasoning dataset SWAG**.
>
> | **Method**        | **SQuADv2 (F1 Score $\uparrow$)** | **SWAG (Accuracy $\uparrow$)** | **MMLU College Mathematics (Accuracy $\uparrow$)** |
> |--------------|------------------|---------------|------------------------------------|
> | FP16         | 19.45          | 0.57         | 0.36                               |
> | PB-LLM       | 2.78           | 0.31         | 0.20                                |
> | BiLLM        | 3.55           | 0.36        | 0.21                               |
> | ARB-LLM-X| 8.23           | 0.41        | 0.22                               |
> |ARB-LLM-RC| 12.24          | 0.44        | 0.23                               |
>
> Our ARB-LLM also outperforms the previous binarization methods PB-LLM and BiLLM on these datasets, narrowing the performance gap with FP16, especially on the long-context SQuADv2 dataset.

---

> > ### Comment · Reviewer_mmbQ · 2024-11-26
> >
> > Thanks authors for the additional experiments to address my questions and comments. They look solid. So I will keep my previous rating.

---

> > > ### Author Response · Authors · 2024-11-26
> > >
> > > Dear Reviewer mmbQ,
> > >
> > > Thank you for your response. We are delighted to see that our answers were able to address your concerns.
> > >
> > > Best,
> > >
> > > Authors

---

### Author Response · Authors · 2024-11-21
**Response to all reviewers and area chairs for a brief summary**

Dear reviewers and area chairs,

We sincerely thank all reviewers and area chairs for their valuable time and insightful comments.

We are pleased to note that:
1. Reviewer mmbQ acknowledges the innovation of our method, and both Reviewer mmbQ and H9m3 recognize that it effectively addresses critical challenges in binary quantization.
2. Reviewers H9m3 and muCn appreciate the comprehensive time and memory analysis provided in our work.
3. All reviewers commend the extensive nature of our experiments, particularly the ablation study, and recognize the superiority of our proposed method, which outperforms the previous SOTA method, BiLLM.

We have responded to each reviewer individually and would like to summarize our responses here:
1. We conduct experiments on **additional metrics, including F1 score and chrF**， further demonstrating the effectiveness of our proposed method.
2. We perform experiments on **various types of datasets (long context, math, and reasoning)** to showcase the versatility and effectiveness of our method.
3. We clarify the **memory usage of our method**, highlighting that it achieves **better performance than GPTQ-2bit while using less memory**.
4. We evaluate our method on newly introduced LLMs, **Phi-3-mini and Phi-3-medium**, demonstrating that it also **performs well on recent models**.
5. We provide a **runtime inference comparison**, showing that our method can **accelerate inference compared to FP16 and PB-LLM models**.
6. We extend the **Pareto curve analysis**, showing that **our method delivers the best performance under the same memory constraints**.
7. We clarify details in the paper, including the **accuracy of QA datasets**, an explanation of **second-order binarization**, and the **results of BiLLM**.

We extend our gratitude again to all reviewers and area chairs!

Best regards,

Authors

---

### Meta-Review · Area_Chair_LeVD · 2024-12-19

**Metareview:**

This paper proposes a new Alternating Refined Binarization (ARB) framework for performing post-training quantization (PTQ) of 1-bit for LLM. After rebuttal, it received scores of 588. On one hand, reviewers acknowledge that the proposed method is innovative, which effectively addresses critical challenges in binary quantization. The experiments are comprehensive, with thorough ablation studies. On the other hand, the reviewer who gave a score of 5 still shows concerns regarding runtime and performance. The authors have tried to address the reviewer's concerns during the rebuttal. Overall, the AC thinks that the merits of the paper outweigh the flaws, therefore, decided to recommend acceptance by the end.

**Additional Comments On Reviewer Discussion:**

This paper received mixed scores of 588.

1. During the rebuttal, the authors had some discussions with the reviewer who gave a score of 5, and tried to address the reviewer's concerns via providing more detailed results on memory comparison, results on additional Phi-3 models, and providing runtime inference metrics. The reviewer still showed concerns regarding runtime and performance, and the practical application of Binary LLMs.

2. Other reviewers also showed the concern that the evaluation seems limited to perplexity and average accuracy. During rebuttal, the authors provided additional results.

---

### Decision · Program_Chairs · 2025-01-22

Accept (Poster)